# SPHERE: Mitigating the Loss of Spectral Plasticity in Mixture-of-Experts for Deep Reinforcement Learning

**Lirui Luo** [1 2]  **Guoxi Zhang** [2]  **Hongming Xu** [2]  **Cong Fang**[† 1 3]  **Qing Li**[† 2]

## Abstract

In deep reinforcement learning (DRL), an agent is trained from a stream of experience. In a continual learning setting, such agents can suffer from *plasticity loss*: their ability to learn new skills from new experiences diminishes over training. Recently, Mixture-of-Experts (MoE) networks have been reported to enable scaling laws and facilitate the learning of diverse skills. However, in continual reinforcement learning settings, their performance can degenerate as learning proceeds, indicating a loss of plasticity. To address this, building on Neural Tangent Kernel (NTK) theory, we formalize the plasticity loss in MoE policies as a loss of *spectral plasticity*. We then derive a tractable proxy for spectral plasticity, one expressible in terms of individual expert feature matrices. Leveraging this proxy, we introduce *SPHERE*, a practical Parseval penalty tailored for MoE-based policies that alleviates the loss of spectral plasticity. On MetaWorld and HumanoidBench, SPHERE improves average success under continual RL by 133% and 50% over an unregularized MoE baseline, while maintaining higher spectral plasticity throughout training.

## 1. Introduction

In neuroscience and cognitive science, *plasticity loss* refers to a gradual decline in a learner's ability to incorporate new experience (Livingston, 1966; Mateos-Aparicio & Rodríguez-Moreno, 2019). Recent work has shown that deep reinforcement learning (DRL) agents can exhibit a

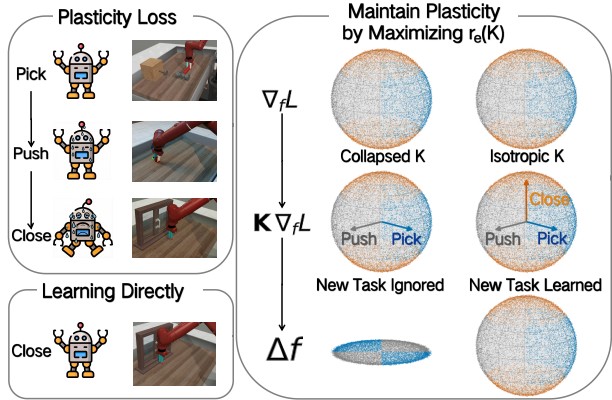

*Figure 1.* **Plasticity loss as loss of spectral plasticity. Left:** Continual RL can fail on later tasks despite isolated learnability. **Right:** By Eq. (6), $\Delta f = -\eta \, \mathbf{K} \, \nabla_f L$; low effective rank of the empirical Neural Tangent Kernel (eNTK) restricts updates to a few directions (collapsed spectrum), while high eNTK effective rank enables diverse directions (isotropic spectrum).

similar effect. As training proceeds, it becomes harder for the policy to improve from new experience (Kumar et al., 2021; Nikishin et al., 2022; Lyle et al., 2022; 2023; Abbas et al., 2023; Dohare et al., 2024; Sutton, 2018). This plasticity loss limits the ability of DRL agents to perform in continual learning settings.

Meanwhile, Mixture-of-Experts (MoE) architectures provide a way to scale policy capacity in DRL and have been deployed in multi-task control for manipulators (Huang et al., 2025b), humanoids (Wang et al., 2025), and quadrupeds (Huang et al., 2025a), as well as in large-scale online RL systems (Obando-Ceron et al., 2024). Yet empirical studies report that, in continual learning settings, MoE performance can degrade as training proceeds (Willi et al., 2024), suggesting that MoE itself can suffer severe plasticity loss.

Plasticity loss in DRL has been increasingly studied. However, its manifestation in MoE policies remains poorly characterized. Motivated by this gap, we focus on mitigating plasticity loss in MoE policies. Addressing this problem is important because MoE policies are increasingly used in modern RL, yet plasticity loss can cause systematic failures on later tasks even when those tasks are learnable in

---

[1]State Key Lab of General AI, School of Intelligence Science and Technology, Peking University [2]State Key Laboratory of General Artificial Intelligence, BIGAI [3]Institute for Artificial Intelligence, Peking University. Correspondence to: Cong Fang <fangcong@pku.edu.cn>, Qing Li <dylan.liqing@gmail.com>. † Corresponding authors. Project page: https://sphere-rl.github.io/.

*Proceedings of the 43rd International Conference on Machine Learning*, Seoul, South Korea. PMLR 306, 2026. Copyright 2026 by the author(s).

isolation.

In this paper, we formalize plasticity loss through *spectral plasticity*: the diversity of functional update directions available to the policy, quantified by the spectral-entropy effective rank of the empirical Neural Tangent Kernel (eNTK). Under this lens, plasticity loss corresponds to a decline in this effective rank, which in turn confines functional updates to fewer directions. Since explicitly forming the eNTK matrix is intractable, we derive its layerwise block decomposition. This decomposition leads to a tractable effective-rank proxy controlled by a *weighted expert feature matrix* available from forward passes, yielding a principled, practical regularization target.

Guided by this view, we introduce *SPHERE*, Spectral Plasticity via Hyperspherical Expert REgularization, which applies a practical Parseval penalty to this weighted feature matrix. On MetaWorld and HumanoidBench, SPHERE improves average success by up to 133% and 50% over an unregularized MoE baseline, while maintaining higher spectral plasticity throughout training.

Concretely, we make the following contributions:

- We formalize plasticity loss as the loss of spectral plasticity and derive a tractable proxy building on NTK theory.

- We introduce SPHERE, a principled and practical regularizer to counter the loss of spectral plasticity.

- We evaluate SPHERE on MetaWorld and Humanoid-Bench, showing consistent gains over MoE baselines and prior plasticity methods, and sustained spectral plasticity throughout training.

**Conflict of Interest Disclosure.** The authors declare no financial conflicts of interest related to this work.

## 2. Related Work

**Plasticity Loss in DRL.** Plasticity loss, also described as capacity loss (Lyle et al., 2022) and primacy bias (Nikishin et al., 2022), refers to a progressive decline in an agent's ability to learn from new interaction data as training proceeds (Lyle et al., 2023; Nikishin et al., 2022). Empirically, it manifests as increased dormant neurons (Sokar et al., 2023), effective-rank and spectral collapse in representations (Chung et al., 2024), and related collapse in Hessian spectra (He et al., 2025). It is often attributed to the non-stationarity of RL training, where early experience can dominate and hinder learning from later samples (Igl et al., 2021; Kumar et al., 2021). In continual RL, these effects can lead to substantial performance degradation (Dohare et al., 2024). Recent work links plasticity loss to eNTK spectral

collapse and proposes mitigation strategies in dense DRL architectures (Lyle et al., 2025; Tang et al., 2025). We build on this line of work but focus specifically on MoE policies.

**Spectral Regularization.** Regularization based on spectral quantities has also been used to mitigate plasticity loss. Spectral normalization and related regularizers stabilize training by constraining the largest singular value of weight matrices (Miyato et al., 2018; Bjorck et al., 2021; Lewandowski et al., 2025). A related line studies Hessian geometry, including Hessian spectral collapse, as a mechanism of plasticity loss (Lewandowski et al., 2023; He et al., 2025). SPHERE focuses on the empirical NTK spectrum of MoE policies, aiming to preserve spectrum-wide uniformity rather than only controlling the largest component of a weight spectrum.

**MoE in Online DRL.** MoE architectures have been adopted in online DRL as a means of scaling policy capacity via sparse expert routing (Obando-Ceron et al., 2024), and have been deployed in multi-task control for manipulators (Huang et al., 2025b), humanoids (Wang et al., 2025), and quadrupeds (Huang et al., 2025a). Willi et al. (2024) further evaluate MoE in continual RL settings and report performance degradation over task sequences, indicating that MoE policies can suffer plasticity loss. Despite the broad use and study of MoE in RL, work on mitigating plasticity loss in MoE policies remains lacking.

## 3. Preliminaries

**Markov Decision Processes.** Consider a Markov Decision Process (MDP) defined by a tuple $(\mathcal{S}, \mathcal{A}, \mathcal{P}, \mathcal{R}, \gamma, \rho_0, T)$, with state space $\mathcal{S}$, action space $\mathcal{A}$, transition kernel $\mathcal{P} : \mathcal{S} \times \mathcal{A} \to \mathcal{P}(\mathcal{S})$, reward function $\mathcal{R} : \mathcal{S} \times \mathcal{A} \to \mathbb{R}$, discount factor $\gamma \in [0, 1)$, initial-state distribution $\rho_0$, and horizon $T$ (Sutton, 2018). The agent interacts with the MDP via a policy $a_t \sim \pi(\cdot \mid s_t)$ that maps states to action distributions. The objective is to optimize the policy to maximize the expected discounted return $J(\pi) = \mathbb{E}_\pi[\sum_{t=0}^{T} \gamma^t r_t]$, where $s_0 \sim \rho_0$, $s_{t+1} \sim \mathcal{P}(\cdot \mid s_t, a_t)$ and $r_t = \mathcal{R}(s_t, a_t)$. The state–action value function is $q^\pi(s, a) = \mathbb{E}_\pi[\sum_{t=0}^{T} \gamma^t r_t \mid s_0 = s, a_0 = a]$ for $(s, a) \in \mathcal{S} \times \mathcal{A}$.

**Top-$K$ Mixture-of-Experts Policies.** In a dense MoE (Shazeer et al., 2017), a set of experts $\{m_e(\cdot; \theta_e)\}_{e=1}^{E}$ is combined by a gate that produces mixture weights. Given an input $x$, each expert produces an output $m_e(x; \theta_e)$, and the gate computes logits $s(x; \psi) \in \mathbb{R}^E$ and normalized weights $h(x; \psi) \in \mathbb{R}^E$ (e.g., via softmax). The MoE output is the weighted sum

$$y(x; \Theta) = \sum_{e=1}^{E} m_e(x; \theta_e) \, h_e(x; \psi), \tag{1}$$

where $\Theta = (\psi, \{\theta_e\}_{e=1}^{E})$ denotes all learnable parameters.

Top-$K$ MoE introduces sparsity by routing each input to only a subset of experts (Shazeer et al., 2017). Given gate logits $s(x; \psi) \in \mathbb{R}^E$, define the selected expert index set $\mathcal{S}(x) \subseteq \{1, \ldots, E\}$ as $\mathcal{S}(x) = \arg \operatorname{top}_K(s(x; \psi))$, i.e., the indices of the $K$ largest components of $s(x; \psi)$, with $|\mathcal{S}(x)| = K$. The corresponding subset softmax is

$$h_e^{(K)}(x) = \begin{cases} \dfrac{\exp\left(s_e(x)\right)}{\sum_{j \in \mathcal{S}(x)} \exp\left(s_j(x)\right)}, & \text{if } e \in \mathcal{S}(x), \\ 0, & \text{otherwise,} \end{cases} \quad (2)$$

and the MoE output is

$$y(x; \Theta) = \sum_{e \in \mathcal{S}(x)} m_e(x; \theta_e) \, h_e^{(K)}(x; \psi), \quad (3)$$

In this paper, we also consider DS-MoE (Dai et al., 2024), which splits experts more finely and includes always-active shared experts.

**Empirical Neural Tangent Kernel.** Consider a parameterized model $f(x; \theta) \in \mathbb{R}$ with parameters $\theta \in \mathbb{R}^P$. Given inputs $X = \{x_n\}_{n=1}^N$, let $f(X; \theta) \in \mathbb{R}^N$ stack the outputs on $X$, and let $\mathbf{J} \in \mathbb{R}^{N \times P}$ denote its Jacobian with respect to $\theta$, whose $n$-th row is $(\nabla_\theta f(x_n; \theta))^\top$. Then the empirical neural tangent kernel (eNTK) matrix (Lyle et al., 2025) is

$$\mathbf{K} = \mathbf{J}\mathbf{J}^\top \in \mathbb{R}^{N \times N}, \quad (4)$$

which captures gradient correlations between samples. Here $\mathbf{K}$ is symmetric positive semidefinite (SPSD): $(\mathbf{J}\mathbf{J}^\top)^\top = \mathbf{J}\mathbf{J}^\top$, and for any $v \in \mathbb{R}^N$, $v^\top \mathbf{K} v = v^\top \mathbf{J}\mathbf{J}^\top v = \|\mathbf{J}^\top v\|_2^2 \geq 0$. The eNTK also governs gradient descent dynamics in output space. Consider the loss $L(\theta) = \sum_{n=1}^N \ell(f(x_n; \theta))$ on the minibatch $X$, and denote $f_i(X) = f(X; \theta_i) \in \mathbb{R}^N$. Then discrete-time gradient descent induces the functional iteration (Lee et al., 2019)

$$f_{i+1}(X) = f_i(X) - \eta \, \mathbf{K}_i \, \nabla_{f_i(X)} L, \quad (5)$$

where $\mathbf{K}_i$ denotes the eNTK matrix on $X$ at iteration $i$. Dropping the iteration index, let $\Delta f(X) = f_{i+1}(X) - f_i(X)$ denote the change in outputs across two consecutive gradient steps, and write $\nabla_f L = \nabla_{f(X)} L$. Then Eq. (5) gives

$$\Delta f = -\eta \, \mathbf{K} \, \nabla_f L, \quad (6)$$

**Continual RL and Plasticity Loss.** In continual RL, an agent encounters a sequence of tasks $\mathcal{T}_1, \mathcal{T}_2, \ldots, \mathcal{T}_k$, each modeled as an MDP. We assume these MDPs share the same state and action spaces but may differ in transition dynamics and reward functions. Recent work in DRL links plasticity loss to spectral collapse in the eNTK spectrum. Lyle et al. (2025) and Tang et al. (2025) demonstrate that the spectrum of the eNTK of dense MLP networks collapses to a low-rank structure as plasticity degrades.

**Spectral-Entropy Effective Rank.** Since rank is a discrete quantity, it is not amenable to gradient-based optimization and is highly sensitive to small perturbations. We therefore use the spectral-entropy effective rank (Roy & Vetterli, 2007) as a smooth proxy. For a matrix $M$ with singular values $\{\sigma_i(M)\}$ and normalized weights $p_i = \sigma_i(M)/\sum_j \sigma_j(M)$, this spectral-entropy effective rank is

$$r_e(M) = \exp\Big( - \sum_{\{i:\, \sigma_i(M) > 0\}} p_i \log p_i \Big), \quad (7)$$

where $r_e(M) \leq \operatorname{rank}(M) \leq \min\{m, n\}$. The first inequality holds with equality if and only if the nonzero singular values of $M$ are equal; the second holds with equality if and only if $M$ is full rank. For SPSD matrices (e.g., $\mathbf{K}$), singular values coincide with eigenvalues.

# 4. Method

In this section, we introduce *spectral plasticity* and use it to characterize plasticity loss in Top-$K$ MoE policies. Section 4.1 formalizes spectral plasticity and relates its decay to plasticity loss. To obtain a tractable objective without explicitly constructing the eNTK matrix $\mathbf{K}$, Section 4.2 analyzes spectral plasticity in parameter space and decomposes the resulting matrix structure into smaller matrices. Building on this decomposition, Section 4.3 derives a tractable lower bound that serves as a surrogate objective for preserving spectral plasticity. Finally, Section 4.4 instantiates and optimizes this lower bound, yielding the SPHERE penalty.

## 4.1. Formulating Spectral Plasticity

**A Spectral View of Plasticity.** We interpret plasticity loss through the spectrum of the eNTK matrix $\mathbf{K}$. Since $\mathbf{K}$ is SPSD, it admits the eigendecomposition $\mathbf{K} = \sum_{j=1}^N \lambda_j u_j u_j^\top$, where $\{u_j\}_{j=1}^N$ are orthonormal eigenvectors and $\lambda_j \geq 0$ are the corresponding eigenvalues. Substituting into Eq. (6) yields

$$\Delta f = -\eta \sum_{j=1}^N \lambda_j \langle u_j, \nabla_f L \rangle u_j, \quad (8)$$

where $\langle u_j, \nabla_f L \rangle$ is the projection of the loss gradient onto direction $u_j$. $\{u_j\}$ are the update directions and $\{\lambda_j\}$ are their gains. If $\lambda_j = 0$, then the component of $\nabla_f L$ along $u_j$ is completely filtered out and cannot change the outputs, implying a loss of learning capacity in that direction. More generally, when the spectrum is highly non-uniform, $\mathbf{K}$ biases updates toward a small set of directions, implicitly encoding a prior over gradient structure and attenuating components that do not align with the dominant eigenspace. Accordingly, high plasticity corresponds to an isotropic spectrum, with eigenvalues as equal as possible. This motivates the following definition.

**Definition 4.1** (Spectral Plasticity). Given inputs $X = \{x_i\}_{i=1}^N$, let $\mathbf{K}(X; \theta)$ denote the eNTK matrix (Eq. (4)). We define spectral plasticity as $\mathrm{SP}(X; \theta) \triangleq r_e(\mathbf{K}(X; \theta))$.

Directly working with the sample-space eNTK $\mathbf{K} = \mathbf{J}\mathbf{J}^\top$ is expensive to form and analyze: given $\mathbf{J} \in \mathbb{R}^{N \times P}$, forming $\mathbf{J}\mathbf{J}^\top$ explicitly costs $O(N^2 P)$ time and requires $O(NP + N^2)$ memory to store $\mathbf{J}$ and $\mathbf{K}$. Therefore, we now introduce a tractable parameter-space surrogate for spectral plasticity.

### 4.2. Decomposing Spectral Plasticity

Since the hierarchical structure of the parameter space, we shift the analysis from the sample space to the parameter space to decompose it into smaller matrices, thereby simplifying the computation. First, we introduce the spectral plasticity surrogate for the parameter space as follows:

**Definition 4.2** (Gauss–Newton surrogate). Let $\mathbf{J}$ be as defined in Eq. (4). The Gauss-Newton matrix (Botev et al., 2017; Zhao et al., 2024) is defined as

$$G^{\mathrm{GN}} \triangleq \frac{1}{N} \mathbf{J}^\top \mathbf{J} \in \mathbb{R}^{P \times P}. \qquad (9)$$

Since $\mathbf{K} = \mathbf{J}\mathbf{J}^\top$ and $\mathbf{J}^\top \mathbf{J}$ share the same nonzero eigenvalues (Horn & Johnson, 2012; Golub & Van Loan, 2013) and $r_e$ is invariant under positive scalar rescaling (Roy & Vetterli, 2007), we have

$$r_e(\mathbf{K}) = r_e(\mathbf{J}^\top \mathbf{J}) = r_e((1/N)\mathbf{J}^\top \mathbf{J}) = r_e(G^{\mathrm{GN}}). \qquad (10)$$

For the Top-$K$ MoE it is convenient to partition the parameter Jacobian $\mathbf{J} \in \mathbb{R}^{N \times P}$ according to gate and expert parameters:

$$\mathbf{J} = \begin{bmatrix} \mathbf{J}^{\mathrm{g}} \mid \mathbf{J}^{\mathrm{exp}} \end{bmatrix}, \qquad (11)$$

where $\mathbf{J}^{\mathrm{g}} \in \mathbb{R}^{N \times P^{\mathrm{g}}}$ contains the columns corresponding to gate parameters $\psi$ and $\mathbf{J}^{\mathrm{exp}} \in \mathbb{R}^{N \times P^{\mathrm{exp}}}$ contains the columns corresponding to expert parameters $\{\theta_e\}_{e=1}^E$. Within the gate, we further decompose

$$\mathbf{J}^{\mathrm{g}} = \begin{bmatrix} \mathbf{J}_1^{\mathrm{g}} \cdots \mathbf{J}_{L^{\mathrm{g}}}^{\mathrm{g}} \end{bmatrix}, \qquad (12)$$

where $L^{\mathrm{g}}$ denotes the number of gate layers and $\mathbf{J}_\ell^{\mathrm{g}} \in \mathbb{R}^{N \times p_\ell^{\mathrm{g}}}$ stacks the vectorized gradients with respect to the parameters of gate layer $\ell$. For the experts, we similarly write

$$\mathbf{J}^{\mathrm{exp}} = \begin{bmatrix} \mathbf{J}_1^{\mathrm{exp}} \cdots \mathbf{J}_{L^{\mathrm{exp}}}^{\mathrm{exp}} \end{bmatrix}, \qquad \mathbf{J}_\ell^{\mathrm{exp}} = \begin{bmatrix} \mathbf{J}_{1,\ell}^{\mathrm{exp}} \cdots \mathbf{J}_{E,\ell}^{\mathrm{exp}} \end{bmatrix}, \qquad (13)$$

where $L^{\mathrm{exp}}$ denotes the number of expert layers and $\mathbf{J}_{e,\ell}^{\mathrm{exp}} \in \mathbb{R}^{N \times p_{e,\ell}^{\mathrm{exp}}}$ stacks the vectorized gradients with respect to the parameters of expert $e$ at layer $\ell$.

Under the gate/expert partition above, the GN matrix $G^{\mathrm{GN}} = (1/N)\mathbf{J}^\top \mathbf{J}$ has the block form

$$\begin{aligned} G^{\mathrm{GN}} &= \frac{1}{N} \begin{bmatrix} (\mathbf{J}^{\mathrm{g}})^\top \mathbf{J}^{\mathrm{g}} & (\mathbf{J}^{\mathrm{g}})^\top \mathbf{J}^{\mathrm{exp}} \\ (\mathbf{J}^{\mathrm{exp}})^\top \mathbf{J}^{\mathrm{g}} & (\mathbf{J}^{\mathrm{exp}})^\top \mathbf{J}^{\mathrm{exp}} \end{bmatrix} \\ &= \begin{bmatrix} \mathbf{G}^{\mathrm{GN,g}} & \mathbf{G}^{\mathrm{GN,g,exp}} \\ \mathbf{G}^{\mathrm{GN,exp,g}} & \mathbf{G}^{\mathrm{GN,exp}} \end{bmatrix}, \end{aligned} \qquad (14)$$

Here $\mathbf{G}^{\mathrm{GN,g}}$ and $\mathbf{G}^{\mathrm{GN,exp}}$ denote the gate and expert blocks, and $\mathbf{G}^{\mathrm{GN,g,exp}}$ and $\mathbf{G}^{\mathrm{GN,exp,g}}$ denote the cross blocks. $(\mathbf{J}^{\mathrm{g}})^\top \mathbf{J}^{\mathrm{g}}$ is an $L^{\mathrm{g}} \times L^{\mathrm{g}}$ block matrix whose $(\ell, \ell')$ block is $(\mathbf{J}_\ell^{\mathrm{g}})^\top \mathbf{J}_{\ell'}^{\mathrm{g}}$, and $(\mathbf{J}^{\mathrm{exp}})^\top \mathbf{J}^{\mathrm{exp}}$ is an $L^{\mathrm{exp}} \times L^{\mathrm{exp}}$ block matrix whose $(\ell, \ell')$ block is $(\mathbf{J}_\ell^{\mathrm{exp}})^\top \mathbf{J}_{\ell'}^{\mathrm{exp}}$. Since $\mathbf{J}_\ell^{\mathrm{exp}} = \begin{bmatrix} \mathbf{J}_{1,\ell}^{\mathrm{exp}} \cdots \mathbf{J}_{E,\ell}^{\mathrm{exp}} \end{bmatrix}$, each same-layer expert block $(\mathbf{J}_\ell^{\mathrm{exp}})^\top \mathbf{J}_\ell^{\mathrm{exp}}$ is itself an $E \times E$ block matrix whose $(e, e')$ block is $(\mathbf{J}_{e,\ell}^{\mathrm{exp}})^\top \mathbf{J}_{e',\ell}^{\mathrm{exp}}$. We next derive the corresponding layerwise gate and expert blocks.

We now express the eNTK effective rank in terms of the layerwise gate and expert GN blocks, yielding a decomposition of $r_e(\mathbf{K})$ across layers.

**Assumption 4.1** (Block-diagonal approximation). We adopt a block-diagonal approximation (Botev et al., 2017):

$$G^{\mathrm{GN}} \approx \Big( \bigoplus_{\ell=1}^{L^{\mathrm{g}}} \mathbf{G}_\ell^{\mathrm{GN,g}} \Big) \bigoplus \Big( \bigoplus_{\ell=1}^{L^{\mathrm{exp}}} \mathbf{G}_\ell^{\mathrm{GN,exp}} \Big), \qquad (15)$$

where $\mathbf{G}_\ell^{\mathrm{GN,g}}$ and $\mathbf{G}_\ell^{\mathrm{GN,exp}}$ denote the gate- and expert-layer blocks and $\bigoplus$ denotes the block direct sum.

A useful property of a block-diagonal matrix is that its effective rank admits a block-wise decomposition, in which the blocks correspond to layers in our application. Lemma 4.1 formalizes this decomposition, and Proposition 4.1 applies it to decompose $r_e(\mathbf{K})$ across layer blocks.

**Lemma 4.1** (Block-diagonal effective rank). *Let $M = \bigoplus_{b=1}^B M_b$ be block-diagonal. Set $s_b = \|M_b\|_*$ and assume $\sum_b s_b > 0$. Define mixture weights $\alpha_b = s_b / \sum_m s_m$. Then*

$$r_e(M) = \exp\Big( H(\alpha) + \sum_{\{b:\, s_b > 0\}} \alpha_b \log r_e(M_b) \Big), \qquad (16)$$

*where $H(\alpha) = -\sum_b \alpha_b \log \alpha_b$.*

*Proof.* Proof in Appendix E. □

**Proposition 4.1** (Layerwise mixture form of $r_e(\mathbf{K})$). *Under (15), define nuclear norms $s_\ell^{\mathrm{g}} = \|\mathbf{G}_\ell^{\mathrm{GN,g}}\|_*$ and $s_\ell^{\mathrm{exp}} = \|\mathbf{G}_\ell^{\mathrm{GN,exp}}\|_*$. Let $S = \sum_m s_m^{\mathrm{g}} + \sum_m s_m^{\mathrm{exp}} > 0$ and mixture weights $\alpha_\ell^{\mathrm{g}} = s_\ell^{\mathrm{g}}/S$ and $\alpha_\ell^{\mathrm{exp}} = s_\ell^{\mathrm{exp}}/S$. Then the spectral-*

entropy effective rank of the eNTK satisfies

$$r_e(\mathbf{K}) \approx \exp\Big( H(\alpha) + \sum_{\{\ell:\, s_\ell^{\mathrm{g}} > 0\}} \alpha_\ell^{\mathrm{g}} \log r_e\big(\mathbf{G}_\ell^{\mathrm{GN,g}}\big) + \sum_{\{\ell:\, s_\ell^{\mathrm{exp}} > 0\}} \alpha_\ell^{\mathrm{exp}} \log r_e\big(\mathbf{G}_\ell^{\mathrm{GN,exp}}\big)\Big),$$

$$(17)$$

where $H(\alpha) = -\sum_{\ell:\, s_\ell^{\mathrm{g}} > 0} \alpha_\ell^{\mathrm{g}} \log \alpha_\ell^{\mathrm{g}} - \sum_{\ell:\, s_\ell^{\mathrm{exp}} > 0} \alpha_\ell^{\mathrm{exp}} \log \alpha_\ell^{\mathrm{exp}}$ is the Shannon entropy of the combined gate and expert mixture weights.

*Proof.* Combine (10) and the block-diagonal approximation (15) with Lemma 4.1. □

Moreover, since $r_e(M) \leq \mathrm{rank}(M) \leq P(M)$, the maximal attainable effective rank of the gate block is limited by its parameter count $P^{\mathrm{g}}$, which in typical Top-$K$ MoE policies is much smaller than the expert parameter count $P^{\mathrm{exp}}$. Thus the gate contribution to the mixture in Proposition 4.1 is limited. In the sequel, we treat the gate terms as fixed and focus on the expert blocks $\{\mathbf{G}_\ell^{\mathrm{GN,exp}}\}$.

## 4.3. A Lower Bound on Spectral Plasticity

We now derive a lower bound for $r_e(\mathbf{G}_\ell^{\mathrm{GN,exp}})$. Each expert-layer Gauss–Newton block $\mathbf{G}_\ell^{\mathrm{GN,exp}}$ can be expressed in terms of two components: (i) the concatenation of expert feature terms weighted by their corresponding Top-$K$ gating, and (ii) pre-activation backprop-gradient terms. Then $r_e(\mathbf{G}_\ell^{\mathrm{GN,exp}})$ is lower-bounded by the inverse of the condition number of a tractable Kronecker proxy $\tilde{\mathbf{G}}_\ell^{\mathrm{GN,exp}}$.

**Proposition 4.2** (Surrogate for $r_e(\mathbf{G}_\ell^{\mathrm{GN,exp}})$). *Consider an expert layer $\ell$ of the Top-$K$ MoE policy (3). Define the concatenated weighted expert feature vector*

$$a_{\ell-1}(x_i) = \big[\, h_{i,1}^{(K)} a_{1,\ell-1}^{\mathrm{exp}}(x_i)^\top \mid \cdots \mid h_{i,E}^{(K)} a_{E,\ell-1}^{\mathrm{exp}}(x_i)^\top \,\big]^\top,$$

$$(18)$$

*where $a_{e,\ell-1}^{\mathrm{exp}}(x_i) \in \mathbb{R}^{d_{e,\ell-1}^{\mathrm{exp}}}$ is the input to expert $e$ on sample $x_i$ and $h_{i,e}^{(K)}$ is the Top-$K$ gating weight from (2). Stack $a_{\ell-1}(x_i)$ over the batch into $\Phi_{\ell-1}$ and define the feature Gram*

$$\mathbf{A}_{\ell-1}^{\mathrm{exp}} = (1/N)\, \Phi_{\ell-1}^\top \Phi_{\ell-1}. \qquad (19)$$

*Similarly, define the concatenated expert backprop vector $g_\ell(x_i) = \big[\, g_{1,\ell}^{\mathrm{exp}}(x_i)^\top \mid \cdots \mid g_{E,\ell}^{\mathrm{exp}}(x_i)^\top \,\big]^\top$, stack it over the batch into $\Psi_\ell$, and define the gradient Gram*

$$\mathbf{G}_\ell^{\mathrm{exp}} = (1/N)\, \Psi_\ell^\top \Psi_\ell, \qquad (20)$$

*where $g_{e,\ell}^{\mathrm{exp}}(x_i) \in \mathbb{R}^{d_{e,\ell}^{\mathrm{exp}}}$ is the corresponding pre-activation backprop vector. Under a within-layer independence approximation (Martens & Grosse, 2015), let*

$\tilde{\mathbf{G}}_\ell^{\mathrm{GN,exp}} = \mathbf{A}_{\ell-1}^{\mathrm{exp}} \otimes \mathbf{G}_\ell^{\mathrm{exp}}$ *be the Kronecker proxy for the expert-layer block $\mathbf{G}_\ell^{\mathrm{GN,exp}}$. Let $k_\ell = \dim(\mathbf{G}_\ell^{\mathrm{GN,exp}})$. Then*

$$r_e(\mathbf{G}_\ell^{\mathrm{GN,exp}}) \geq \frac{k_\ell}{\kappa(\tilde{\mathbf{G}}_\ell^{\mathrm{GN,exp}})} = \mathrm{LB}_\ell^{\mathrm{GN,exp}}, \qquad (21)$$

*where $\mathrm{LB}_\ell^{\mathrm{GN,exp}}$ denotes the corresponding proxy-to-target lower bound and $\kappa(A) = \lambda_{\max}(A)/\lambda_{\min}(A)$ is the spectral condition number. Moreover, equality holds if and only if $\kappa(\tilde{\mathbf{G}}_\ell^{\mathrm{GN,exp}}) = 1$.*

*Proof.* Proof in Appendix D. □

To increase this lower bound, we introduce a spectral contraction operator that provably increases it. Definition 4.3 formalizes the update, Lemma 4.3 establishes monotone decrease of $\kappa$, Lemma 4.4 propagates monotone $\kappa$ through the Kronecker proxy, and Proposition 4.5 yields a monotone increase of the bound in Proposition 4.2.

**Definition 4.3** (Spectral contraction). *Let $R(M)$ be a differentiable regularizer on nonzero SPSD matrices $M \in \mathbb{R}^{m \times m}$. We say that $R$ is spectrally contractive if there exists $\eta_{\max} > 0$ such that, for all nonzero SPSD $M_t$ and all $0 \leq \eta \leq \eta_{\max}$, the update induced by a gradient step of size $\eta$ monotonically contracts every eigenvalue toward the mean eigenvalue. In other words, let $\bar{\lambda}_t = \mathrm{Tr}(M_t)/m$ denote the mean eigenvalue of $M_t$, and write $M_t = U\, \mathrm{diag}(\lambda_{i,t})\, U^\top$. Then there exists $\beta \in [0,1]$ such that*

$$M_{t+1} = U\, \mathrm{diag}\big((1-\beta)\lambda_{i,t} + \beta\bar{\lambda}_t\big)\, U^\top. \qquad (22)$$

**Lemma 4.3** (Monotonicity of spectral contraction). *If $M_{t+1}$ is obtained from $M_t$ by a spectral contraction update in the sense of Definition 4.3, then*

$$\kappa(M_{t+1}) \leq \kappa(M_t). \qquad (23)$$

*Proof.* Proof in Appendix F.3. □

**Lemma 4.4** (Kronecker preservation of monotone $\kappa$). *Let $A_t, A_{t+1} \in \mathbb{R}^{m \times m}$ and $B_t \in \mathbb{R}^{n \times n}$ be nonzero SPSD. If $\kappa(A_{t+1}) \leq \kappa(A_t)$, then for $K_t = A_t \otimes B_t$ and $K_{t+1} = A_{t+1} \otimes B_t$,*

$$\kappa(K_{t+1}) \leq \kappa(K_t). \qquad (24)$$

*Proof.* Proof in Appendix F.5. □

**Proposition 4.5** (Spectral contraction on $\mathbf{A}_{\ell-1}^{\mathrm{exp}}$ increases the lower bound). *If $\mathbf{A}_{\ell-1,t+1}^{\mathrm{exp}}$ is obtained from $\mathbf{A}_{\ell-1,t}^{\mathrm{exp}}$ by a spectrally contractive update (Definition 4.3), with $\mathbf{G}_\ell^{\mathrm{exp}}$ fixed, then*

$$\mathrm{LB}_{\ell,t+1}^{\mathrm{GN,exp}} \geq \mathrm{LB}_{\ell,t}^{\mathrm{GN,exp}}. \qquad (25)$$

*Proof.* Proof in Appendix F.6. □

In summary, we derive an explicit lower bound on $r_e(\mathbf{G}_\ell^{\mathrm{GN,exp}})$ via a Kronecker proxy and identify an operator that monotonically improves this lower bound. Via Proposition 4.1, improving the expert-block lower bounds provides a tractable surrogate objective for increasing spectral plasticity $r_e(\mathbf{K})$. The proxy depends only on the low-dimensional Gram factors $\mathbf{A}_{\ell-1}^{\mathrm{exp}}$ and $\mathbf{G}_\ell^{\mathrm{exp}}$, avoiding explicit construction and manipulation of the full expert-layer block $\mathbf{G}_\ell^{\mathrm{GN,exp}}$ and thereby substantially reducing computation and memory.

### 4.4. Algorithm

In this section, we instantiate the spectral contraction operator.

**Objective.** Motivated by the tractable proxy in Section 4.3, we act on the feature factor to improve spectral plasticity. We regularize only the MoE *actor* to isolate effects on policy plasticity, and target its last hidden expert layer: empirically, deeper layers tend to exhibit the lowest effective-rank embeddings (Huh et al., 2023). Denote the corresponding weighted expert feature Gram by $\mathbf{A}_{\mathrm{last}}^{\mathrm{exp}} \in \mathbb{R}^{m \times m}$, where $m = \sum_{e=1}^{E} d_{e,\mathrm{last}}^{\mathrm{exp}}$ is the concatenated feature dimension. SPHERE instantiates the spectral contraction desideratum (Definition 4.3) with a quadratic penalty on the *anisotropic* component of $\mathbf{A}_{\mathrm{last}}^{\mathrm{exp}}$.

$$\begin{aligned} \mathcal{L}_{\mathrm{SPHERE}}\big(\mathbf{A}_{\mathrm{last}}^{\mathrm{exp}}\big) &= \Big\| \mathbf{A}_{\mathrm{last}}^{\mathrm{exp}} - \tfrac{\mathrm{Tr}(\mathbf{A}_{\mathrm{last}}^{\mathrm{exp}})}{m}\mathbf{I}_m \Big\|_F^2 \\ &= \big\| \mathbf{A}_{\mathrm{last}}^{\mathrm{exp}} \big\|_F^2 - \tfrac{\mathrm{Tr}(\mathbf{A}_{\mathrm{last}}^{\mathrm{exp}})^2}{m}. \end{aligned} \quad (26)$$

See Proposition F.1 for the detailed derivation of the second equality.

**Proposition 4.6** (SPHERE is spectrally contractive)**.** *The SPHERE penalty $\mathcal{L}_{\mathrm{SPHERE}}(\cdot)$ (Eq. (26)) is spectrally contractive (Definition 4.3) with $\eta_{\max} = \frac{1}{2}$.*

*Proof.* Proof in Appendix F.4. □

**Theorem 4.7** (Optimizing Eq. (26) increases $r_e(\mathbf{K})$)**.** *If SPHERE is applied to $\mathbf{A}_{\mathrm{last},t}^{\mathrm{exp}}$ with $0 \leq \eta \leq \frac{1}{2}$, then it increases $r_e(\mathbf{K})$.*

*Proof.* Proof in Appendix F.7. □

**Total Loss.** Let $\mathcal{L}_{\mathrm{PPO}}$ denote the PPO objective. At each gradient update, we compute $\mathbf{A}_{\mathrm{last}}^{\mathrm{exp}}$ and add the SPHERE penalty $\mathcal{L}_{\mathrm{SPHERE}}(\mathbf{A}_{\mathrm{last}}^{\mathrm{exp}})$ (Eq. (26)) to the actor.

$$\mathcal{L} = \mathcal{L}_{\mathrm{PPO}} + \lambda^{\mathrm{e}} \mathcal{L}_{\mathrm{SPHERE}}\big(\mathbf{A}_{\mathrm{last}}^{\mathrm{exp}}\big). \quad (27)$$

$\lambda^{\mathrm{e}}$ is a regularization coefficient. See Algorithm 1 for a summary of the training procedure.

## 5. Experiments

We first provide evidence of the loss of spectral plasticity in both dense PPO and MoE policies. We then evaluate SPHERE using task success and $r_e(\mathbf{K})$, ablate key design choices, and provide a qualitative analysis.

### 5.1. Experimental Setup

**Training Protocols.** We consider two training protocols over a shared task list: *RL*, where each task is trained independently from scratch, and *CRL*, where a single agent is trained sequentially through the list. In CRL, training on each new task resumes from the final parameters of the previous task.

**Tasks and Training.** We evaluate on tasks from two robotics benchmarks: HumanoidBench (Sferrazza et al., 2024) and MetaWorld (Yu et al., 2019). For HumanoidBench, we use the five H1 tasks, as they share the same observation and action spaces. For MetaWorld, we adopt the CW10 subset from Continual World (Wołczyk et al., 2021), a set of 10 manipulation tasks over shared observation and action spaces. Following benchmark protocols (Wołczyk et al., 2021; Sferrazza et al., 2024), we train for $T_{\mathrm{MW}} = 10^6$ environment steps per MetaWorld task and $T_{\mathrm{HB}} = 10^7$ steps per HumanoidBench task.

**Metrics.** We report the mean and standard deviation of final success rates over five random seeds. To quantify spectral plasticity, we track $r_e(\mathbf{K})$; higher is better.[1]

**Baselines.** We compare dense PPO, PPO(10x), with an actor that has $10\times$ the PPO actor parameter count, Top-$K$ MoE actors, Dense-MoE actors, and DS-MoE (Dai et al., 2024). For Top-$K$ MoE, we use $E = 10$ experts and route $K = 2$ experts per input. As mitigation baselines on the same Top-$K$ MoE actor as SPHERE, we evaluate Layer-Norm (LN), which has been shown empirically to mitigate plasticity loss in DRL (Juliani & Ash, 2024; Ma et al., 2024), C-CHAIN (Tang et al., 2025), which reduces churn to preserve plasticity, CBP (Dohare et al., 2024), which continuously reinitializes a subset of neurons, and PW (Chung et al., 2024), which regularizes weight matrices toward approximate orthogonality. All methods share the same PPO training pipeline.

### 5.2. Results

**Evidence of the Loss of Spectral Plasticity in DRL.** We first show that the loss of spectral plasticity arises broadly

---

[1]We evaluate $r_e(\mathbf{K})$ on a fixed held-out batch of states from rollouts of a pre-trained Stair policy (excluded from the CRL sequence) and reuse it across evaluation checkpoints and methods. This targets plasticity on *new* experience rather than on previously seen training states.

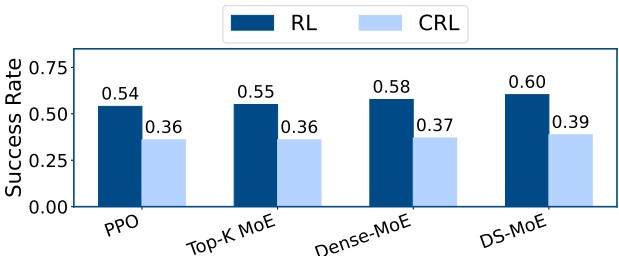

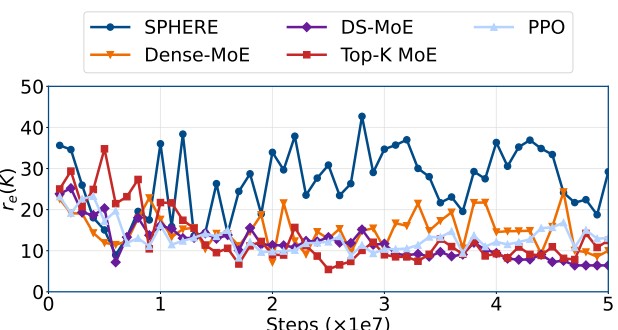

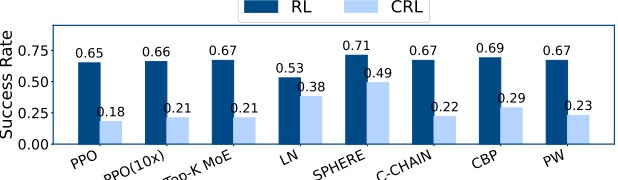

Figure 2. **All methods show degraded performance under CRL relative to RL.** We report average final success rates on Humanoid-Bench under RL and CRL for PPO, Top-$K$ MoE, Dense-MoE, and DS-MoE.

Figure 4. **Under CRL, SPHERE improves average success by 133% over Top-$K$ MoE and reduces the RL–CRL gap by 52%.** We report average final success rates on MetaWorld across methods under RL and CRL. SPHERE outperforms other mitigation baselines.

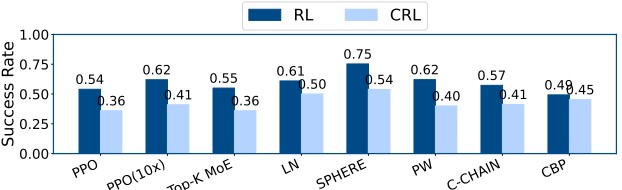

Figure 3. **SPHERE maintains higher $r_e(\mathbf{K})$ while baselines decay.** We plot $r_e(\mathbf{K})$ over HumanoidBench CRL training for PPO, Top-$K$ MoE, Dense-MoE, DS-MoE, and SPHERE. The baseline decay indicates the loss of spectral plasticity.

Figure 5. **Relative to Top-$K$ MoE, SPHERE improves average success by 36% under RL and 50% under CRL.** We report average final success rates on HumanoidBench across methods under RL and CRL. SPHERE outperforms other mitigation baselines.

in DRL across policy architectures. Since CRL learns each new task by resuming from the previous task checkpoint, a direct probe of plasticity is to compare it against training from a random initialization. We therefore compare RL with CRL for PPO, Top-$K$ MoE, Dense-MoE, and DS-MoE, and report both final success rates and $r_e(\mathbf{K})$.

As shown in Figure 2, simply switching from RL to CRL reduces success for both dense PPO and MoE policies. Consistent with this, Figure 3 shows that MoE variants also exhibit rank collapse of $\mathbf{K}$, indicating the loss of spectral plasticity across MoE types. Together, these results show that the loss of spectral plasticity is a common phenomenon across policy architectures.

**SPHERE Mitigates the Loss of Spectral Plasticity.** We next evaluate whether SPHERE mitigates the loss of spectral plasticity, using both final success rates and $r_e(\mathbf{K})$.

Figures 4 and 5 summarize the average final success rates under RL and CRL on MetaWorld and HumanoidBench.

*MetaWorld.* On MetaWorld, SPHERE improves performance under both RL and CRL relative to the Top-$K$ MoE baseline. Under CRL, SPHERE improves average success by +133% relative to Top-$K$ MoE and substantially narrows

the RL–CRL gap by 52%. The gain is larger under CRL, where sustained adaptation across tasks is required and preserving spectral plasticity helps maintain learning on later tasks.

*HumanoidBench.* On HumanoidBench, relative to Top-$K$ MoE, SPHERE improves average success by 36% under RL and 50% under CRL. The large gain under RL shows that spectral plasticity can decay even within a single task. PPO continually updates from newly collected on-policy rollouts, so the effective training distribution drifts over time. HumanoidBench uses $10\times$ more environment steps per task than MetaWorld, which makes this within-task decay more pronounced and amplifies the benefit of SPHERE under RL. Consistent with this, Figure 3 shows that SPHERE maintains higher $r_e(\mathbf{K})$ during CRL training.

**Key Design Choices for SPHERE.** We ablate key design choices of SPHERE on HumanoidBench under CRL with five seeds, as summarized in Table 1.

*Layerwise Placement.* SPHERE regularizes the last hidden expert layer of the actor. Applying the same penalty to all hidden expert layers underperforms regularizing only the last hidden expert layer, suggesting that regularizing earlier layers can overconstrain representation learning.

*Cross-Expert Concatenation.* SPHERE regularizes a Gram matrix formed from the routing-weighted *concatenation* of expert features. This concatenated Gram includes off-diagonal blocks that capture cross-expert correlations, so

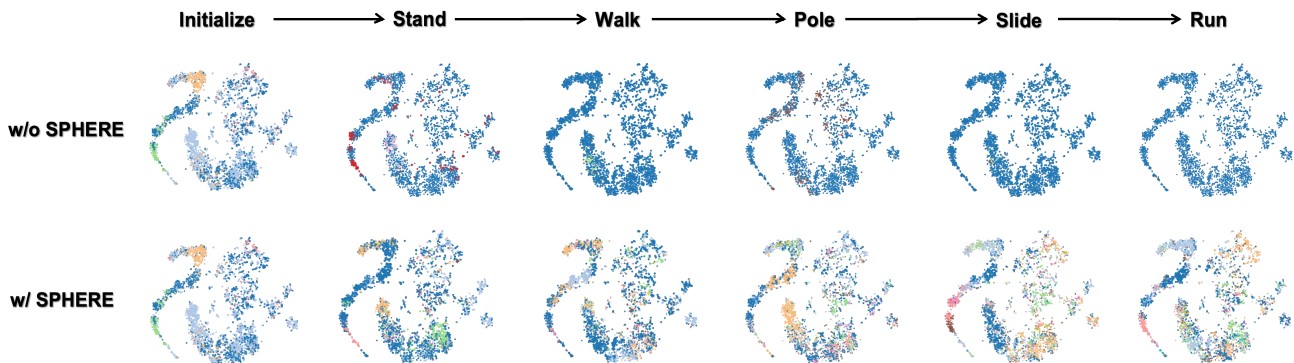

*Figure 6.* **SPHERE avoids spectral collapse to a single component.** We visualize held-out states with t-SNE and color each state by the singular direction of the weighted expert feature matrix with the largest absolute projection. Columns show snapshots along the HumanoidBench CRL sequence. Top: w/o SPHERE. Bottom: w/ SPHERE.

*Table 1.* **Layerwise placement and the $\mathbf{A}_{\text{last}}^{\text{exp}}$ instantiation of Equation (26), as well as cross-expert concatenation (Equation (18)), are all useful.** On HumanoidBench under CRL, we compare SPHERE to variants that (i) remove the regularizer (w/o SPHERE), (ii) apply Equation (26) to $\mathbf{A}_{\ell}^{\text{exp}}$ at all hidden expert layers $\ell$, (iii) regularize each expert independently and sum the losses (no cross-expert concatenation), and (iv) regularize $\mathbf{G}_{\text{last}}^{\text{exp}}$ instead of $\mathbf{A}_{\text{last}}^{\text{exp}}$. None of these variants matches SPHERE consistently.

| Variant | Average success |
| --- | --- |
| w/o SPHERE | $0.36 \pm 0.08$ |
| w/ SPHERE | $\mathbf{0.54 \pm 0.12}$ |
| All hidden expert layers | $0.42 \pm 0.07$ |
| Per-expert loss sum | $0.40 \pm 0.08$ |
| $\mathbf{G}_{\text{last}}^{\text{exp}}$ Regularization | $0.43 \pm 0.09$ |

Eq. (26) directly suppresses cross-expert correlation and encourages a shared, decorrelated representation geometry. To test whether these cross-expert terms matter, we keep the same layerwise placement as SPHERE but regularize each expert independently and sum the losses, which removes cross-expert correlation penalties. This *Per-expert loss sum* variant yields $0.40 \pm 0.08$, only slightly above the baseline and below SPHERE, suggesting that suppressing cross-expert correlations via concatenation is important for the gains.

*Gradient-Gram Regularization.* By the Kronecker proxy in Proposition 4.2, the surrogate for spectral plasticity depends on a feature Gram and a gradient Gram. As a complementary ablation, we apply the SPHERE penalty to $\mathbf{G}_{\text{last}}^{\text{exp}}$ while leaving $\mathbf{A}_{\text{last}}^{\text{exp}}$ unregularized. This improves over the baseline but remains below SPHERE, suggesting that the gains primarily come from shaping $\mathbf{A}_{\text{last}}^{\text{exp}}$. Since $\mathbf{G}_{\text{last}}^{\text{exp}}$ requires backpropagation to extract, we adopt feature-Gram regularization on $\mathbf{A}_{\text{last}}^{\text{exp}}$ as the main SPHERE instantiation.

**Qualitative Analysis.** Figure 6 shows the evolution of effective-rank collapse in the weighted expert representations and the associated loss of spectral plasticity over the HumanoidBench CRL sequence. Using a fixed held-out state batch from Section 5.1, we visualize last-layer weighted expert features with t-SNE at initialization and after each task. We compute the singular directions of the weighted expert feature matrix on the held-out batch, and color each state by the direction that best aligns with its feature vector, i.e., the one with the largest absolute projection.

In the baseline, colors quickly become nearly uniform, indicating that most states concentrate on a single dominant feature direction. Consistent with our proxy analysis in Proposition 4.2, this loss of feature diversity manifests as a collapsed weighted feature Gram $\mathbf{A}_{\text{last}}^{\text{exp}}$, which shrinks the expert-block surrogate and, via Proposition 4.1, contributes to a lower $r_e(\mathbf{K})$. With SPHERE, multiple components remain active throughout the sequence, consistent with preserved spectral plasticity.

As a quantitative complement, Figure 7 shows that $r_e(\mathbf{A}_{\text{last}}^{\text{exp}})$ positively co-varies with $r_e(\mathbf{K})$ across rollout batches at a representative checkpoint. The Pearson correlation is $r=0.846$, supporting expert-feature isotropy as a practical proxy signal for spectral plasticity.

# 6. Conclusion

In this paper, we formalize the loss of spectral plasticity in MoE policies for deep reinforcement learning and propose a regularizer to mitigate it. Building on NTK theory, we formulate MoE plasticity loss as a decline in spectral plasticity. To make this quantity tractable, we further decompose it and derive a tractable surrogate objective. We then instantiate it as SPHERE, a principled and practical regularizer applied to weighted expert features to counter the loss of spectral plasticity. On MetaWorld and HumanoidBench, SPHERE improves average success by up to 133% and 50% over an unregularized MoE baseline, while maintaining higher spectral plasticity throughout training.

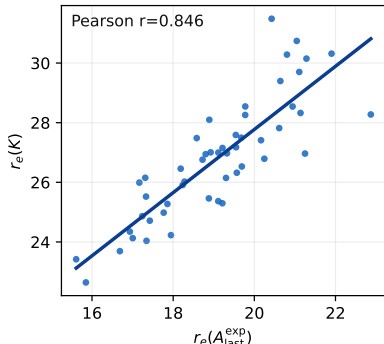

*Figure 7.* **Expert feature isotropy tracks spectral plasticity.** We visualize a scatter plot of $r_e(\mathbf{A}_{\text{last}}^{\text{exp}})$ versus $r_e(\mathbf{K})$. The Pearson correlation is $r=0.846$. This supports using the SPHERE penalty on $\mathbf{A}_{\text{last}}^{\text{exp}}$ as a practical proxy for spectral plasticity.

**Limitations.** Our derivation relies on theoretical approximations. While we empirically validate these assumptions in the appendix, removing these approximations is a direction for future theoretical work. We focus on MoE policies for continuous-control RL on MetaWorld and HumanoidBench; extending SPHERE to MoE architectures in large-scale pretrained foundation models remains future work.

# Acknowledgements

C. Fang was supported by the National Science and Technology Major Project (No. 2022ZD0114902) and NSF China (No. 92470117).

# Impact Statement

This paper presents work whose goal is to advance the field of machine learning. There are many potential societal consequences of our work, none of which we feel must be specifically highlighted here.

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

# Appendix

## Appendix Roadmap

**Implementation.** Appendix A contains implementation details. It includes the training procedure in Algorithm 1, minimal JAX code for the SPHERE penalty, adaptive regularization and hyperparameters, and implementation details for estimating $r_e(\mathbf{K})$.

**Core theory derivations.** Appendices B to F provide supporting derivations for the main text. They cover matrix-calculus preliminaries, Top-$K$ MoE gradients and Jacobian-Gram structure, the Tracy–Singh/Kronecker proxy construction, effective-rank identities, and SPHERE-as-spectral-contraction proofs.

**Interpretation and extensions.** Appendices G to I provide interpretation and extensions. They include the $E{=}1$ dense specialization, an eNTK spectral perspective on load balancing and specialization, and perturbation and stability bounds.

**Extra experiments.** Appendix J reports additional experimental results. It includes per-task success tables, additional ablations and robustness analyses, runtime overhead at large expert counts, and supervised continual-learning experiments in computer vision and natural language processing, including Transformer MoE backbones.

# Appendix Contents

# A. Implementation Details

## A.1. Training Procedure

Algorithm 1 summarizes PPO with an MoE actor and SPHERE regularization.

---

**Algorithm 1** PPO with Actor MoE SPHERE regularization

---

1: Initialize actor $\pi_\theta$ with MoE layers and Top-$K$ gating, and critic $V_\phi$ with dense layers.
2: **for** iteration $j = 1$ to $J$ **do**
3:     Collect trajectories with $\pi_\theta$ and compute advantages and PPO targets.
4:     **for** each PPO minibatch $\mathcal{B}$ of $N$ samples **do**
5:         Run the MoE actor forward pass on $\mathcal{B}$ and compute $\mathbf{A}_{\text{last}}^{\text{exp}}$ at the last hidden expert layer.
6:         Compute $\mathcal{L}_{\text{PPO}}(\theta, \phi; \mathcal{B})$ and the SPHERE loss $\mathcal{L}_{\text{SPHERE}}\big(\mathbf{A}_{\text{last}}^{\text{exp}}\big)$ via Eq. (26), then set the total loss $\mathcal{L}$ via Eq. (27).
7:         Update $\{\theta, \phi\}$ with a gradient step on $\mathcal{L}$.
8:     **end for**
9: **end for**

---

## A.2. Minimal JAX Implementation of the SPHERE Penalty

The SPHERE penalty (Eq. (26)) depends only on the Frobenius norm and trace of the normalized Gram matrix induced by the weighted expert features (Eq. (18)). It can be implemented in a few lines without SVD; to avoid forming a large Gram, compute it in feature space or sample space depending on which dimension is smaller.

```
# JAX implementation (phi: [T,D])
import jax.numpy as jnp

# Dimensions:
#   T: minibatch size (#samples/tokens)
#   E: number of experts
#   H: per-expert feature dimension
#   D: concatenated feature dimension (= E*H)
#
# Construct phi by concatenating gating-weighted expert features:
#   probs: [T,E], expert_features: [T,E,H]  ->  phi: [T, E*H]
weighted = probs[..., None] * expert_features
phi = weighted.reshape((weighted.shape[0], -1))

def sphere_loss(phi):
    T, D = phi.shape
    G = (phi.T @ phi)/T if D <= T else (phi @ phi.T)/T
    fro2 = jnp.sum(G * G); tr = jnp.trace(G)
    return fro2 - (tr * tr) / D
```

## A.3. Adaptive Regularization Coefficient

In Eq. (27), $\lambda^{\text{e}}$ trades off adherence to the SPHERE feature constraint against PPO optimization. We set it adaptively per minibatch. Unless stated, we use gradient-norm scaling,

$$\lambda^{\text{e}} \;=\; \rho \cdot \text{stopgrad}\Big(\frac{\|\nabla_\theta \mathcal{L}_{\text{actor}}\|_2}{\|\nabla_\theta \mathcal{L}_{\text{SPHERE}}\|_2 + \varepsilon}\Big),$$

where $\rho$ is a tunable ratio and $\varepsilon > 0$ is a small constant.

## A.4. Training Hyperparameters

*Table 2.* **Training hyperparameters for Top-$K$ MoE PPO with SPHERE.** Values summarize the default MetaWorld and Humanoid-Bench training setups.

| | MetaWorld | HumanoidBench |
|---|---|---|
| Env steps per task | 1,000,000 | 10,000,000 |
| Vectorized envs ($n_{\text{env}}$) | 8 | 128 |
| Observation normalization | none | VecNormalize |
| PPO rollout steps ($n_{\text{steps}}$) | 2048 | 16 |
| PPO batch size | 256 | 64 |
| PPO epochs per update | 10 | 10 |
| Discount $\gamma$ | 0.99 | 0.99 |
| GAE $\lambda$ | 0.95 | 0.95 |
| Clip range | 0.20 | 0.20 |
| Max grad norm | 0.50 | 0.50 |
| Entropy coefficient | 0.00 | 0.00 |
| Value coefficient | 0.50 | 0.50 |
| Learning rate | linear 3e−4 → 1e−4 | linear 3e−4 → 1e−4 over first half |
| Activation | ReLU | ReLU |
| MLP width | 256 | 256 |
| MLP depth | 3 | 3 |
| Number of experts $E$ | 10 | 10 |
| Top-$K$ | 2 | 2 |
| Gating softmax temperature $\tau$ | 1.0 | 1.0 |
| SPHERE ratio $\rho$ | 1e−1 | 1e−3 |

## A.5. Implementation Details for Computing $r_e(\mathbf{K})$

**Output definition.** To measure spectral plasticity, we track the spectral-entropy effective rank $r_e(\mathbf{K})$ of the actor empirical NTK. For the diagonal-Gaussian actor used by PPO, we take $f_\theta(s) := \mu_\theta(s)$, where $\mu_\theta(s)$ is the mean vector of $\pi_\theta(a \mid s) = \mathcal{N}(\mu_\theta(s), \text{diag}(\sigma^2))$, and exclude the log-standard-deviation parameters. In our implementation, $\log \sigma$ is learned and state-independent.

**SLQ computation of $r_e(\mathbf{K})$.** We estimate $r_e(\mathbf{K})$ without explicitly constructing the sample-space Gram. Using Eq. (7), let $\{\lambda_k\}$ denote the eigenvalues of $\mathbf{K}$ and define $p_k := \lambda_k/\text{Tr}(\mathbf{K})$ (assuming $\text{Tr}(\mathbf{K}) > 0$). Define $f(x) := x \log x$ with the convention $f(0) := 0$. Writing an eigendecomposition $\mathbf{K}=U\Lambda U^\top$ with $\Lambda = \text{diag}(\lambda_1, \ldots, \lambda_N)$, we have $\text{Tr}(f(\mathbf{K})) = \text{Tr}(f(\Lambda)) = \sum_k f(\lambda_k) = \sum_k \lambda_k \log \lambda_k$. Then

$$-\sum_k p_k \log p_k = -\sum_k \frac{\lambda_k}{\text{Tr}(\mathbf{K})} \log \frac{\lambda_k}{\text{Tr}(\mathbf{K})} = -\frac{1}{\text{Tr}(\mathbf{K})} \sum_k \lambda_k \big(\log \lambda_k - \log \text{Tr}(\mathbf{K})\big)$$

$$= \log \text{Tr}(\mathbf{K}) - \frac{1}{\text{Tr}(\mathbf{K})} \sum_k \lambda_k \log \lambda_k = \log \text{Tr}(\mathbf{K}) - \frac{\text{Tr}(f(\mathbf{K}))}{\text{Tr}(\mathbf{K})}.$$

Substituting into $r_e(\mathbf{K}) = \exp(-\sum_k p_k \log p_k)$ gives

$$r_e(\mathbf{K}) = \exp\Big(\log \text{Tr}(\mathbf{K}) - \frac{\text{Tr}(f(\mathbf{K}))}{\text{Tr}(\mathbf{K})}\Big), \tag{28}$$

Directly constructing $\mathbf{K}$ (or materializing $\mathbf{J} \in \mathbb{R}^{N \times P}$) is typically GPU-memory prohibitive for modern actor networks. We therefore estimate the traces in (28) with stochastic Lanczos quadrature (SLQ). For Gaussian probes $z \sim \mathcal{N}(0, I)$, Hutchinson's identity gives $\text{Tr}(f(\mathbf{K})) = \mathbb{E}_z[z^\top f(\mathbf{K})z]$. Given a probe $z$, run $M$ steps of the Lanczos algorithm on $\mathbf{K}$ with $q_1 = z/\|z\|_2$, using only matrix–vector products with $\mathbf{K}$, to obtain an orthonormal basis $Q_M = [q_1, \ldots, q_M]$ and tridiagonal $T_M := Q_M^\top \mathbf{K} Q_M$. Gauss quadrature then yields the approximation

$$z^\top f(\mathbf{K})z \approx \|z\|_2^2 \, e_1^\top f(T_M)e_1, \tag{29}$$

where $e_1$ is the first canonical basis vector. Averaging (29) over probes gives an estimator of $\text{Tr}(f(\mathbf{K}))$ (and similarly for $\text{Tr}(\mathbf{K})$ by taking $f(x) = x$), which we plug into (28). Finally, each Lanczos matrix–vector product can be computed

as $\mathbf{K}v = \mathbf{J}(\mathbf{J}^\top v)$ via reverse-/forward-mode automatic differentiation, avoiding materialization of $\mathbf{K}$. For vector-valued outputs $f_\theta(x_i) \in \mathbb{R}^d$, we follow the SLQ estimator and reduce to the scalar-output form in Eq. (4) by defining a per-sample scalar projection $g_\theta(x_i) := \mathbf{1}_d^\top f_\theta(x_i)$. Let $\mathbf{J}$ in Eq. (4) be the Jacobian of $g_\theta$ over the batch, i.e., its $i$-th row is $(\nabla_\theta g_\theta(x_i))^\top$.

## A.6. Code Availability

Code for reproducing our experiments is available through the project page.

# B. Technical Preliminaries for Derivations

For completeness, we collect here a few without-loss-of-generality conventions and matrix-calculus identities used in Section 4.

## B.1. Without-Loss-of-Generality Conventions

We record two reductions used throughout the theory. Neither changes the implemented algorithm.

### B.1.1. BIAS ABSORPTION FOR AFFINE LAYERS

Throughout Section 4 we work with bias-free layers without loss of generality. Formally, let a generic gate or expert layer be affine, of the form $z_\ell = \mathbf{W}_\ell a_{\ell-1} + b_\ell$ with $\mathbf{W}_\ell \in \mathbb{R}^{d_\ell \times d_{\ell-1}}$, $a_{\ell-1} \in \mathbb{R}^{d_{\ell-1}}$, $b_\ell \in \mathbb{R}^{d_\ell}$, and $z_\ell \in \mathbb{R}^{d_\ell}$. By augmenting the activation with a constant

$$\tilde{a}_{\ell-1} = \begin{bmatrix} a_{\ell-1} \\ 1 \end{bmatrix}, \qquad \tilde{\mathbf{W}}_\ell = [\, \mathbf{W}_\ell \;\; b_\ell \,],$$

we can rewrite the layer as a purely linear map

$$z_\ell = \tilde{\mathbf{W}}_\ell \tilde{a}_{\ell-1}.$$

Thus, after this reparameterization, each layer can be treated as bias-free by working with $\tilde{\mathbf{W}}_\ell$ and $\tilde{a}_{\ell-1}$. In Section 4 we therefore use the notation $\mathbf{W}_\ell$ and $a_{\ell-1}$ for this bias-absorbed form, and all layer widths $\{d_\ell\}$ are understood for this representation.

### B.1.2. NUMERICAL CONVENTION FOR SPSD MATRICES

All Gram matrices we analyze (e.g., $\mathbf{K}$, $\mathbf{A}_\ell^{\mathrm{exp}}$, and $\mathbf{G}_\ell^{\mathrm{exp}}$) are symmetric positive semidefinite (SPSD) by construction. We do not explicitly perturb these matrices in our experiments.

**Lemma B.1** (Ridge shift turns SPSD into SPD). *Let $M \in \mathbb{R}^{n \times n}$ be SPSD. For any $\varepsilon > 0$, the ridge-shifted matrix $M_\varepsilon := M + \varepsilon \mathbf{I}_n$ is SPD and satisfies $\|M_\varepsilon - M\|_2 = \varepsilon$.*

*Proof.* Since both $M$ and $\mathbf{I}_n$ are symmetric, $M_\varepsilon$ is symmetric. For any nonzero $x \in \mathbb{R}^n$,

$$x^\top M_\varepsilon x = x^\top M x + \varepsilon \|x\|_2^2 \geq \varepsilon \|x\|_2^2 > 0,$$

so $M_\varepsilon \succ 0$. Finally, $M_\varepsilon - M = \varepsilon \mathbf{I}_n$ and $\|\mathbf{I}_n\|_2 = 1$, hence $\|M_\varepsilon - M\|_2 = \varepsilon$. $\qquad\square$

Therefore, whenever a strictly positive spectrum is required (e.g., in $\kappa(\cdot)$ or matrix logarithms), all statements can be read as applied to $M_\varepsilon$ for an arbitrarily small $\varepsilon$, with $\varepsilon$ suppressed in the notation. In this sense, throughout the appendix we freely treat SPSD matrices as SPD.

**Gate and Expert Parameterization in Top-$K$ MoE.** Specializing to the Top-$K$ MoE policy of Section 4, we write the full parameter vector as $\theta = (\psi, \{\theta_e\}_{e=1}^E)$, where $\psi$ denotes the gate parameters and $\theta_e$ denotes the parameters of expert $e$. Let $L^{\mathrm{g}}$ and $L^{\mathrm{exp}}$ denote the numbers of gate and expert layers, respectively. After absorbing biases into the weights as above we take

$$\psi = \{\mathbf{W}_\ell^{\mathrm{g}}\}_{\ell=1}^{L^{\mathrm{g}}}, \qquad \theta_e = \{\mathbf{W}_{e,\ell}^{\mathrm{exp}}\}_{\ell=1}^{L^{\mathrm{exp}}},$$

where $\mathbf{W}_\ell^{\mathrm{g}} \in \mathbb{R}^{d_\ell^{\mathrm{g}} \times d_{\ell-1}^{\mathrm{g}}}$ are the gate weight matrices and $\mathbf{W}_{e,\ell}^{\mathrm{exp}} \in \mathbb{R}^{d_{e,\ell}^{\mathrm{exp}} \times d_{e,\ell-1}^{\mathrm{exp}}}$ are the expert weight matrices for expert $e$ at layer $\ell$. Thus each gate or expert layer is treated as bias-free in the Jacobian Gram analysis, with parameters grouped into gate and expert blocks via $\{\mathbf{W}_\ell^{\mathrm{g}}\}$ and $\{\mathbf{W}_{e,\ell}^{\mathrm{exp}}\}$.

**Lemma B.2** (Linear-layer gradient). *Let $z_\ell = W_\ell a_{\ell-1}$ and $g_\ell = \partial f / \partial z_\ell$ for a scalar-valued function $f$. Then*

$$\frac{\partial f}{\partial W_\ell} = g_\ell a_{\ell-1}^\top.$$

*Proof.* We use differential notation and consider the partial derivative of $f$ with respect to $W_\ell$, treating $a_{\ell-1}$ as independent of $W_\ell$ (i.e., $\partial a_{\ell-1}/\partial W_\ell = 0$). Then $f$ depends on $W_\ell$ only through $z_\ell = W_\ell a_{\ell-1}$. Since $f$ is scalar-valued in $z_\ell$, its differential satisfies $df = (\partial f / \partial z_\ell)^\top dz_\ell = g_\ell^\top dz_\ell$. By the chain rule,

$$df = g_\ell^\top dz_\ell = g_\ell^\top d(W_\ell a_{\ell-1}) = g_\ell^\top dW_\ell a_{\ell-1}.$$

Using the scalar–trace identity $v^\top X u = \mathrm{Tr}(u v^\top X)$ for $v \in \mathbb{R}^m$, $X \in \mathbb{R}^{m \times n}$, and $u \in \mathbb{R}^n$ (see, e.g., Petersen et al., 2008), this can be written as

$$df = \mathrm{Tr}(a_{\ell-1} g_\ell^\top dW_\ell) = \mathrm{Tr}\big((g_\ell a_{\ell-1}^\top)^\top dW_\ell\big),$$

where the second equality uses cyclic invariance of trace, $\mathrm{Tr}(ABC) = \mathrm{Tr}(CAB)$. By the definition of the matrix derivative of a scalar function,

$$df = \mathrm{Tr}\big((\partial f / \partial W_\ell)^\top dW_\ell\big) \quad \text{for all } dW_\ell,$$

which identifies $\partial f / \partial W_\ell = g_\ell a_{\ell-1}^\top$. $\qquad\square$

## C. Gradients of Top-$K$ MoE

**Overview.** This appendix derives the expert-layer block Khatri–Rao structure used in Section 4 and in Proposition 4.2. Along the way, it derives the same-layer gate and expert Jacobian Grams (Eqs. (35) and (40)), which are also used by later structural results (e.g., Appendix D).

### C.1. Setup for Per-Sample Gradients

We first record the Top-$K$ MoE per-sample gradients that will be used throughout the derivations below.

**Top-$K$ MoE Per-Sample Gradients.** For each sample $x_i$, the scalar output can be written as

$$f(x_i) = \sum_{e' \in \mathcal{S}(x_i)} h_{i,e'}^{(K)} m_{e'}(x_i), \tag{30}$$

where $m_{e'}(x_i)$ is the contribution of expert $e'$. Let $\vartheta$ denote any one of the parameter blocks in $\theta$; for example, $\vartheta$ may be the gate parameters $\psi$, an expert block $\theta_e$, or a specific weight matrix such as $\mathbf{W}_\ell^{\mathrm{g}}$ or $\mathbf{W}_{e,\ell}^{\mathrm{exp}}$. Differentiating $f(x_i)$ with respect to such a $\vartheta$ gives

$$\begin{aligned}
\frac{\partial f(x_i)}{\partial \vartheta} &= \sum_{e' \in \mathcal{S}(x_i)} \frac{\partial}{\partial \vartheta}\big(h_{i,e'}^{(K)} m_{e'}(x_i)\big) \\
&= \sum_{e' \in \mathcal{S}(x_i)} \Big(\frac{\partial h_{i,e'}^{(K)}}{\partial \vartheta} m_{e'}(x_i) + h_{i,e'}^{(K)} \frac{\partial m_{e'}(x_i)}{\partial \vartheta}\Big).
\end{aligned} \tag{31}$$

This is the *full* Top-$K$ MoE gradient in block form.

### C.2. Derivation of Same-Layer Jacobian Grams

In this section we derive the same-layer gate and expert Jacobian Grams starting from the per-sample Top-$K$ MoE gradients in Eq. (31).

**Gate Gradients.** For the gate parameters $\psi$, the expert outputs $m_{e'}(x_i)$ do not depend on $\psi$, so the second term in Eq. (31) vanishes for all $e'$, giving

$$\frac{\partial f(x_i)}{\partial \psi} = \sum_{e' \in \mathcal{S}(x_i)} m_{e'}(x_i) \frac{\partial h_{i,e'}^{(K)}}{\partial \psi}.$$

At a specific gate layer $\ell$, only the dependence of the gating weights $h_{i,e'}^{(K)}$ on the layer weights $\mathbf{W}_\ell^{\mathrm{g}}$ contributes. Differentiating $f(x_i)$ with respect to $\mathbf{W}_\ell^{\mathrm{g}}$ and using the product rule gives

$$\frac{\partial f(x_i)}{\partial \mathbf{W}_\ell^{\mathrm{g}}} = \sum_{e' \in \mathcal{S}(x_i)} m_{e'}(x_i) \frac{\partial h_{i,e'}^{(K)}}{\partial W_\ell^{\mathrm{g}}} .$$

Writing the gate as a multilayer network with logits $s(x_i; \psi)$, pre-activations $z_\ell^{\mathrm{g}}(x_i)$, and hidden activations $a_{\ell-1}^{\mathrm{g}}(x_i)$, $h_{i,e'}^{(K)}$ depends on $\mathbf{W}_\ell^{\mathrm{g}}$ only through $z_\ell^{\mathrm{g}}(x_i)$. A second application of the chain rule gives

$$\frac{\partial h_{i,e'}^{(K)}}{\partial \mathbf{W}_\ell^{\mathrm{g}}} = \frac{\partial h_{i,e'}^{(K)}}{\partial z_\ell^{\mathrm{g}}(x_i)} \frac{\partial z_\ell^{\mathrm{g}}(x_i)}{\partial \mathbf{W}_\ell^{\mathrm{g}}}.$$

For the linear map $z_\ell^{\mathrm{g}}(x_i) = \mathbf{W}_\ell^{\mathrm{g}} a_{\ell-1}^{\mathrm{g}}(x_i)$, applying Lemma B.2 with $f(x_i) = h_{i,e'}^{(K)}(x_i)$ and $g_\ell(x_i) = \partial h_{i,e'}^{(K)} / \partial z_\ell^{\mathrm{g}}(x_i)$ gives

$$\frac{\partial h_{i,e'}^{(K)}}{\partial \mathbf{W}_\ell^{\mathrm{g}}} = \frac{\partial h_{i,e'}^{(K)}}{\partial z_\ell^{\mathrm{g}}(x_i)} \left(a_{\ell-1}^{\mathrm{g}}(x_i)\right)^\top,$$

so substituting into the previous expression for $\partial f(x_i)/\partial \mathbf{W}_\ell^{\mathrm{g}}$ yields

$$\frac{\partial f(x_i)}{\partial \mathbf{W}_\ell^{\mathrm{g}}} = \sum_{e' \in \mathcal{S}(x_i)} m_{e'}(x_i) \frac{\partial h_{i,e'}^{(K)}}{\partial z_\ell^{\mathrm{g}}(x_i)} \left(a_{\ell-1}^{\mathrm{g}}(x_i)\right)^\top. \tag{32}$$

Thus gate gradients are explicitly weighted by the expert outputs. Differentiating Eq. (30) with respect to the gate pre-activation $z_\ell^{\mathrm{g}}(x_i)$ and using that $m_{e'}(x_i)$ does not depend on gate activations gives

$$\frac{\partial f(x_i)}{\partial z_\ell^{\mathrm{g}}(x_i)} = \sum_{e' \in \mathcal{S}(x_i)} m_{e'}(x_i) \frac{\partial h_{i,e'}^{(K)}}{\partial z_\ell^{\mathrm{g}}(x_i)} .$$

We therefore identify the gate pre-activation gradient at layer $\ell$ as

$$g_\ell^{\mathrm{g}}(x_i) := \frac{\partial f(x_i)}{\partial z_\ell^{\mathrm{g}}(x_i)} = \sum_{e' \in \mathcal{S}(x_i)} m_{e'}(x_i) \frac{\partial h_{i,e'}^{(K)}}{\partial z_\ell^{\mathrm{g}}(x_i)} \in \mathbb{R}^{d_\ell^{\mathrm{g}}}. \tag{33}$$

and Eq. (32) can be written compactly as

$$\frac{\partial f(x_i)}{\partial \mathbf{W}_\ell^{\mathrm{g}}} = g_\ell^{\mathrm{g}}(x_i) \left(a_{\ell-1}^{\mathrm{g}}(x_i)\right)^\top, \tag{34}$$

where $g_\ell^{\mathrm{g}}(x_i)$ is just a shorthand for the gate pre-activation gradient $\partial f(x_i)/\partial z_\ell^{\mathrm{g}}(x_i)$ and still explicitly contains the expert-output weights $m_{e'}(x_i)$. Thus, like a dense layer, the gate layer admits a per-sample gradient in backprop–activation outer-product form. We define the gate-layer Jacobian block $\mathbf{J}_\ell^{\mathrm{g}}$ to stack the vectorized per-sample gradients with respect to $\mathbf{W}_\ell^{\mathrm{g}}$, so the $i$-th row is

$$j_{i,\ell}^{\mathrm{g}}{}^\top := \left(\mathrm{vec}(\tfrac{\partial f(x_i)}{\partial W_\ell^{\mathrm{g}}})\right)^\top.$$

Using the vec–Kronecker identity $\mathrm{vec}(uv^\top) = v \otimes u$ (see, e.g., Petersen et al., 2008; Horn & Johnson, 2012; Golub & Van Loan, 2013) on $\partial f(x_i)/\partial W_\ell^{\mathrm{g}} = g_\ell^{\mathrm{g}}(x_i)\left(a_{\ell-1}^{\mathrm{g}}(x_i)\right)^\top$, we obtain

$$j_{i,\ell}^{\mathrm{g}}{}^\top = \left(a_{\ell-1}^{\mathrm{g}}(x_i) \otimes g_\ell^{\mathrm{g}}(x_i)\right)^\top.$$

Stacking these rows over $i=1,\ldots,N$ gives the gate-layer Jacobian block $\mathbf{J}_\ell^{\mathrm{g}} \in \mathbb{R}^{N \times p_\ell^{\mathrm{g}}}$ with rows $j_{i,\ell}^{\mathrm{g}}{}^\top$ (and $p_\ell^{\mathrm{g}}{=}d_\ell^{\mathrm{g}} d_{\ell-1}^{\mathrm{g}}$). Its same-layer Gram expands as

$$
\begin{aligned}
\left(\mathbf{J}_\ell^{\mathrm{g}}\right)^\top \mathbf{J}_\ell^{\mathrm{g}} &= \sum_{i=1}^{N} j_{i,\ell}^{\mathrm{g}}\left(j_{i,\ell}^{\mathrm{g}}\right)^\top \\
&= \sum_{i=1}^{N} \left(a_{\ell-1}^{\mathrm{g}}(x_i) \otimes g_\ell^{\mathrm{g}}(x_i)\right) \left(a_{\ell-1}^{\mathrm{g}}(x_i) \otimes g_\ell^{\mathrm{g}}(x_i)\right)^\top \\
&= \sum_{i=1}^{N} \left(a_{\ell-1}^{\mathrm{g}}(x_i) \otimes g_\ell^{\mathrm{g}}(x_i)\right) \left(a_{\ell-1}^{\mathrm{g}}(x_i)^\top \otimes g_\ell^{\mathrm{g}}(x_i)^\top\right) \\
&= \sum_{i=1}^{N} \left(a_{\ell-1}^{\mathrm{g}}(x_i) a_{\ell-1}^{\mathrm{g}}(x_i)^\top\right) \otimes \left(g_\ell^{\mathrm{g}}(x_i) g_\ell^{\mathrm{g}}(x_i)^\top\right),
\end{aligned}
\tag{35}
$$

where the last line uses the Kronecker product rule $(A \otimes B)(C \otimes D){=}(AC) \otimes (BD)$ with $A{=}a_{\ell-1}^{\mathrm{g}}(x_i)$, $B{=}g_\ell^{\mathrm{g}}(x_i)$, $C{=}a_{\ell-1}^{\mathrm{g}}(x_i)^\top$, and $D{=}g_\ell^{\mathrm{g}}(x_i)^\top$.

**Expert Gradients.** For expert parameters $\vartheta \subset \theta_e$, differentiating Eq. (30) with respect to $\vartheta$ gives

$$
\frac{\partial f(x_i)}{\partial \vartheta} = \sum_{e' \in \mathcal{S}(x_i)} h_{i,e'}^{(K)} \frac{\partial m_{e'}(x_i)}{\partial \vartheta}.
$$

Here the gate outputs $h_{i,e'}^{(K)}$ do not depend on $\vartheta$, and $m_{e'}(x_i)$ depends on $\vartheta$ only when $e'{=}e$, so the sum reduces to

$$
\frac{\partial f(x_i)}{\partial \vartheta} = h_{i,e}^{(K)} \frac{\partial m_e(x_i)}{\partial \vartheta}.
$$

Specializing to the layer weights $\vartheta{=}\mathbf{W}_{e,\ell}^{\mathrm{exp}}$ yields

$$
\frac{\partial f(x_i)}{\partial \mathbf{W}_{e,\ell}^{\mathrm{exp}}} = h_{i,e}^{(K)} \frac{\partial m_e(x_i)}{\partial \mathbf{W}_{e,\ell}^{\mathrm{exp}}}.
\tag{36}
$$

Within expert $e$, we write the layer-$\ell$ pre-activation on sample $x_i$ as

$$
z_{e,\ell}(x_i) = \mathbf{W}_{e,\ell}^{\mathrm{exp}} a_{e,\ell-1}^{\mathrm{exp}}(x_i), \qquad a_{e,\ell-1}^{\mathrm{exp}}(x_i) \in \mathbb{R}^{d_{e,\ell-1}^{\mathrm{exp}}},
$$

and define the corresponding backprop signal

$$
g_{e,\ell}^{\mathrm{exp}}(x_i) := \frac{\partial m_e(x_i)}{\partial z_{e,\ell}(x_i)} \in \mathbb{R}^{d_{e,\ell}^{\mathrm{exp}}}.
$$

By the chain rule and the linear-layer gradient Lemma B.2 applied to the scalar function $m_e(x_i)$ and the map $z_{e,\ell}(x_i){=}\mathbf{W}_{e,\ell}^{\mathrm{exp}} a_{e,\ell-1}^{\mathrm{exp}}(x_i)$,

$$
\frac{\partial m_e(x_i)}{\partial \mathbf{W}_{e,\ell}^{\mathrm{exp}}} = g_{e,\ell}^{\mathrm{exp}}(x_i) \, a_{e,\ell-1}^{\mathrm{exp}}(x_i)^\top.
\tag{37}
$$

Substituting into the previous expression yields

$$
\frac{\partial f(x_i)}{\partial \mathbf{W}_{e,\ell}^{\mathrm{exp}}} = h_{i,e}^{(K)} g_{e,\ell}^{\mathrm{exp}}(x_i) a_{e,\ell-1}^{\mathrm{exp}}(x_i)^\top = g_{e,\ell}^{\mathrm{exp}}(x_i) \left(h_{i,e}^{(K)} a_{e,\ell-1}^{\mathrm{exp}}(x_i)\right)^\top.
\tag{38}
$$

Vectorizing and transposing gives the gradient row

$$
j_{i,e,\ell}^\top := \left(\mathrm{vec}\big(g_{e,\ell}^{\mathrm{exp}}(x_i)\big(h_{i,e}^{(K)} a_{e,\ell-1}^{\mathrm{exp}}(x_i)\big)^\top\big)\right)^\top = \big(h_{i,e}^{(K)} a_{e,\ell-1}^{\mathrm{exp}}(x_i) \otimes g_{e,\ell}^{\mathrm{exp}}(x_i)\big)^\top = h_{i,e}^{(K)} \big(a_{e,\ell-1}^{\mathrm{exp}}(x_i) \otimes g_{e,\ell}^{\mathrm{exp}}(x_i)\big)^\top,
\tag{39}
$$

where we used the same vec–Kronecker identity $\mathrm{vec}(uv^\top)=v \otimes u$ and the fact that the scalar $h_{i,e}^{(K)}$ can be absorbed into either factor of the Kronecker product.

Let $\mathbf{J}_{e,\ell}^{\exp} \in \mathbb{R}^{N \times p_{e,\ell}^{\exp}}$ collect the Jacobian rows for expert $e$ at layer $\ell$, so its $i$-th row is $j_{i,e,\ell}^\top$, and stack experts within layer $\ell$ as $\mathbf{J}_\ell^{\exp}=[\mathbf{J}_{1,\ell}^{\exp},\ldots,\mathbf{J}_{E,\ell}^{\exp}] \in \mathbb{R}^{N \times p_\ell^{\exp}}$ with $p_\ell^{\exp}=\sum_e p_{e,\ell}^{\exp}$ and rows $j_{i,\ell}^\top$. The exact same-layer Gram $(\mathbf{J}_\ell^{\exp})^\top \mathbf{J}_\ell^{\exp}$ is then an $E \times E$ block matrix

$$(\mathbf{J}_\ell^{\exp})^\top \mathbf{J}_\ell^{\exp} = \begin{bmatrix} (\mathbf{J}_{1,\ell}^{\exp})^\top \mathbf{J}_{1,\ell}^{\exp} & \cdots & (\mathbf{J}_{1,\ell}^{\exp})^\top \mathbf{J}_{E,\ell}^{\exp} \\ \vdots & \ddots & \vdots \\ (\mathbf{J}_{E,\ell}^{\exp})^\top \mathbf{J}_{1,\ell}^{\exp} & \cdots & (\mathbf{J}_{E,\ell}^{\exp})^\top \mathbf{J}_{E,\ell}^{\exp} \end{bmatrix},$$

whose $(e,e')$ block is

$$(\mathbf{J}_{e,\ell}^{\exp})^\top \mathbf{J}_{e',\ell}^{\exp} = \sum_{i=1}^N j_{i,e,\ell}\, j_{i,e',\ell}^\top.$$

Substituting the per-sample form $j_{i,e,\ell}=h_{i,e}^{(K)} \mathrm{vec}(g_{e,\ell}^{\exp}(x_i)a_{e,\ell-1}^{\exp}(x_i)^\top)$ and using $\mathrm{vec}(uv^\top)=v \otimes u$ gives $j_{i,e,\ell}=h_{i,e}^{(K)}\,(a_{e,\ell-1}^{\exp}(x_i) \otimes g_{e,\ell}^{\exp}(x_i))$, so

$$\begin{aligned} (\mathbf{J}_{e,\ell}^{\exp})^\top \mathbf{J}_{e',\ell}^{\exp} &= \sum_{i=1}^N j_{i,e,\ell}\, j_{i,e',\ell}^\top \\ &= \sum_{i=1}^N h_{i,e}^{(K)} h_{i,e'}^{(K)} \left( a_{e,\ell-1}^{\exp}(x_i) \otimes g_{e,\ell}^{\exp}(x_i) \right) \left( a_{e',\ell-1}^{\exp}(x_i) \otimes g_{e',\ell}^{\exp}(x_i) \right)^\top \\ &= \sum_{i=1}^N h_{i,e}^{(K)} h_{i,e'}^{(K)} \left( a_{e,\ell-1}^{\exp}(x_i) \otimes g_{e,\ell}^{\exp}(x_i) \right) \left( a_{e',\ell-1}^{\exp}(x_i)^\top \otimes g_{e',\ell}^{\exp}(x_i)^\top \right) \\ &= \sum_{i=1}^N h_{i,e}^{(K)} h_{i,e'}^{(K)} \left( a_{e,\ell-1}^{\exp}(x_i) a_{e',\ell-1}^{\exp}(x_i)^\top \right) \otimes \left( g_{e,\ell}^{\exp}(x_i) g_{e',\ell}^{\exp}(x_i)^\top \right), \end{aligned} \tag{40}$$

where the last line uses the Kronecker rule $(A \otimes B)(C \otimes D)=(AC) \otimes (BD)$. The within-layer independence approximation step and the resulting block Khatri–Rao form are collected in Appendix C.3.

## C.3. Within-Layer Independence and Block Khatri–Rao Form

We summarize how the exact expert-layer Jacobian Grams in Eq. (40) lead to the block-factorized form used in Section 4 and in Proposition 4.2.

Collecting the blocks $\{(\mathbf{J}_{e,\ell}^{\exp})^\top \mathbf{J}_{e',\ell}^{\exp}\}_{e,e'}$ gives the exact same-layer expert Gram $(\mathbf{J}_\ell^{\exp})^\top \mathbf{J}_\ell^{\exp}$ as an $E \times E$ block matrix whose $(e,e')$ block is a *sum* of Kronecker products between an activation factor $a_{e,\ell-1}^{\exp}(x_i)a_{e',\ell-1}^{\exp}(x_i)^\top$ and a gradient factor $g_{e,\ell}^{\exp}(x_i)g_{e',\ell}^{\exp}(x_i)^\top$, cf. Eq. (40). For later use it is convenient to introduce the corresponding empirical activation and gradient Grams at the expert layer. Define

$$\begin{aligned} \mathbf{A}_{\ell-1,(e,e')}^{\exp} &= \frac{1}{N} \sum_{i=1}^N h_{i,e}^{(K)} h_{i,e'}^{(K)}\, a_{e,\ell-1}^{\exp}(x_i)a_{e',\ell-1}^{\exp}(x_i)^\top, \\ \mathbf{G}_{\ell,(e,e')}^{\exp} &= \frac{1}{N} \sum_{i=1}^N g_{e,\ell}^{\exp}(x_i)g_{e',\ell}^{\exp}(x_i)^\top, \end{aligned} \tag{41}$$

and collect them into block matrices $\mathbf{A}_{\ell-1}^{\exp}=[\mathbf{A}_{\ell-1,(e,e')}^{\exp}]_{e,e'}$ and $\mathbf{G}_\ell^{\exp}=[\mathbf{G}_{\ell,(e,e')}^{\exp}]_{e,e'}$. In terms of these empirical Grams, the *normalized* same-layer expert Gram

$$\mathbf{G}_\ell^{\mathrm{GN},\exp} = \frac{1}{N}(\mathbf{J}_\ell^{\exp})^\top \mathbf{J}_\ell^{\exp}$$

has $(e, e')$ block

$$\mathbf{G}_{e,e',\ell}^{\mathrm{GN,exp}} = \tfrac{1}{N} \big(\mathbf{J}_{e,\ell}^{\mathrm{exp}}\big)^{\top} \mathbf{J}_{e',\ell}^{\mathrm{exp}}$$

$$= \tfrac{1}{N} \sum_{i=1}^{N} h_{i,e}^{(K)} h_{i,e'}^{(K)} \big(a_{e,\ell-1}^{\mathrm{exp}}(x_i) a_{e',\ell-1}^{\mathrm{exp}}(x_i)^{\top}\big) \otimes \big(g_{e,\ell}^{\mathrm{exp}}(x_i) g_{e',\ell}^{\mathrm{exp}}(x_i)^{\top}\big).$$

Viewing the minibatch index $i$ as a draw from the empirical data distribution, the sum above is an empirical estimate of a fourth-order moment in the activation and gradient factors, since each entry involves terms of the form $\mathbb{E}[a_u a_v g_p g_q]$. A within-layer independence approximation between activations and backprop gradients (see, e.g., Martens & Grosse, 2015; Grosse & Martens, 2016) factorizes this fourth-order moment into a product of second moments, yielding $\mathbb{E}[a_u a_v g_p g_q] \approx \mathbb{E}[a_u a_v]\mathbb{E}[g_p g_q]$; substituting empirical averages for expectations yields the block-factorized approximation

$$\mathbf{G}_{e,e',\ell}^{\mathrm{GN,exp}} \approx \mathbf{A}_{\ell-1,(e,e')}^{\mathrm{exp}} \otimes \mathbf{G}_{\ell,(e,e')}^{\mathrm{exp}}.$$

Collecting these approximated blocks over all $(e, e')$ pairs, the expert-layer Gram can be written compactly as a block Khatri–Rao product

$$\mathbf{G}_{\ell}^{\mathrm{GN,exp}} \approx \mathbf{A}_{\ell-1}^{\mathrm{exp}} * \mathbf{G}_{\ell}^{\mathrm{exp}},$$

$$\big[\mathbf{A}_{\ell-1}^{\mathrm{exp}} * \mathbf{G}_{\ell}^{\mathrm{exp}}\big]_{e,e'} = \mathbf{A}_{\ell-1,(e,e')}^{\mathrm{exp}} \otimes \mathbf{G}_{\ell,(e,e')}^{\mathrm{exp}},$$

where $*$ denotes the block Khatri–Rao product on corresponding blocks.

To make the activation blocks explicit, for an expert layer $\ell$ we define the *concatenated expert feature* for sample $i$ as

$$a_{\ell-1}^{\mathrm{ce}}(x_i) = \big[ h_{i,1}^{(K)} a_{1,\ell-1}^{\mathrm{exp}}(x_i)^{\top} \mid h_{i,2}^{(K)} a_{2,\ell-1}^{\mathrm{exp}}(x_i)^{\top} \mid \cdots$$
$$\mid h_{i,E}^{(K)} a_{E,\ell-1}^{\mathrm{exp}}(x_i)^{\top} \big]^{\top}. \tag{42}$$

This vector lies in $\mathbb{R}^{D_{\ell-1}}$, where $D_{\ell-1} = \sum_{e=1}^{E} d_{e,\ell-1}^{\mathrm{exp}}$. Stacking over the minibatch gives the concatenated expert feature matrix $\Phi_{\ell-1}^{\mathrm{ce}} \in \mathbb{R}^{N \times D_{\ell-1}}$ with rows $\big(a_{\ell-1}^{\mathrm{ce}}(x_i)\big)^{\top}$. The concatenated expert feature Gram is

$$\mathbf{A}_{\ell-1}^{\mathrm{exp}} = \tfrac{1}{N} \big(\Phi_{\ell-1}^{\mathrm{ce}}\big)^{\top} \Phi_{\ell-1}^{\mathrm{ce}} \in \mathbb{R}^{D_{\ell-1} \times D_{\ell-1}},$$

with $(e, e')$ block

$$\big[\mathbf{A}_{\ell-1}^{\mathrm{exp}}\big]_{e,e'} = \tfrac{1}{N} \sum_{i=1}^{N} h_{i,e}^{(K)} h_{i,e'}^{(K)} a_{e,\ell-1}^{\mathrm{exp}}(x_i) a_{e',\ell-1}^{\mathrm{exp}}(x_i)^{\top}.$$

This is exactly the empirical block activation Gram $\mathbf{A}_{\ell-1}^{\mathrm{exp}}$. For notational convenience, for expert layers we also define the concatenated expert gradients

$$g_{\ell}^{\mathrm{exp}}(x_i) := \big[ g_{1,\ell}^{\mathrm{exp}}(x_i)^{\top} \mid \cdots \mid g_{E,\ell}^{\mathrm{exp}}(x_i)^{\top} \big]^{\top},$$

so that $\mathbf{G}_{\ell}^{\mathrm{exp}} = (1/N) \sum_i g_{\ell}^{\mathrm{exp}}(x_i) g_{\ell}^{\mathrm{exp}}(x_i)^{\top}$ contains the within- and cross-expert gradient covariances at layer $\ell$.

# D. Surrogate for $r_e(\mathbf{G}_{\ell}^{\mathrm{GN,exp}})$

The expert-layer Gauss–Newton block $\mathbf{G}_{\ell}^{\mathrm{GN,exp}}$ is a block Khatri–Rao product whose spectral structure does not admit a closed-form decomposition. To circumvent this difficulty, we show that optimizing a simpler Kronecker proxy $\mathbf{A}_{\ell-1}^{\mathrm{exp}} \otimes \mathbf{G}_{\ell}^{\mathrm{exp}}$ suffices: its condition number controls a lower bound on the effective rank of $\mathbf{G}_{\ell}^{\mathrm{GN,exp}}$.

The proof proceeds in two steps. First, we embed the Khatri–Rao Gram as a principal submatrix of a Tracy–Singh product that shares the same eigenvalues as the Kronecker proxy (Proposition D.1). Second, Cauchy interlacing translates spectral isotropy on the proxy into an effective-rank guarantee on the target (Lemma D.3).

**Proposition D.1** (Expert block Khatri–Rao as a Tracy–Singh principal submatrix with Kronecker spectrum). *Let* $\mathbf{A} = \mathbf{A}_{\ell-1}^{\mathrm{exp}} \in \mathbb{R}^{D_{\ell-1} \times D_{\ell-1}}$ *and* $\mathbf{G} = \mathbf{G}_{\ell}^{\mathrm{exp}} \in \mathbb{R}^{d_\ell \times d_\ell}$ *be the concatenated expert feature and gradient Grams at expert layer $\ell$ defined in Section 4, with symmetric expert-wise block partitions* $\{\mathbf{A}_{e,e'}\}$ *and* $\{\mathbf{G}_{e,e'}\}$. *Denote by* $\mathbf{T}_\ell := \mathbf{A} \boxtimes \mathbf{G}$ *the Tracy–Singh product of* $\mathbf{A}$ *and* $\mathbf{G}$, *and by* $\mathbf{K}_\ell := \mathbf{A} \otimes \mathbf{G}$ *the Kronecker proxy. Then:*

1. $\mathbf{T}_\ell$ is SPSD and admits a Gram factorization $\mathbf{T}_\ell = Z_\ell Z_\ell^\top$ for a suitable matrix $Z_\ell$ constructed from blockwise Cholesky factors of $\mathbf{A}$ and $\mathbf{G}$.

2. The expert-layer Gram equals a Khatri–Rao block of $\mathbf{T}_\ell$ and can be written as a principal submatrix

$$\mathbf{G}_\ell^{\mathrm{GN,exp}} \;=\; \mathbf{A} * \mathbf{G} \;=\; S_\ell^\top \, \mathbf{T}_\ell \, S_\ell \,,$$

for some column-orthonormal selection matrix $S_\ell$ (so that $S_\ell^\top S_\ell = I$) that selects expert-aligned blocks.

3. $\mathbf{T}_\ell$ and the Kronecker proxy $\mathbf{K}_\ell$ share the same multiset of eigenvalues, and hence the same spectral-entropy effective rank:

$$\{\lambda(\mathbf{T}_\ell)\} \equiv \{\lambda(\mathbf{K}_\ell)\}, \qquad r_e(\mathbf{T}_\ell) = r_e(\mathbf{K}_\ell).$$

*Proof.* For (i), SPSD of $\mathbf{A}$ and $\mathbf{G}$ implies the existence of factorizations $\mathbf{A} = XX^\top$ and $\mathbf{G} = YY^\top$ with $X = [X_1^\top, \ldots, X_E^\top]^\top$ and $Y = [Y_1^\top, \ldots, Y_E^\top]^\top$ block-partitioned compatibly with the experts, so $\mathbf{A}_{e,e'} = X_e X_{e'}^\top$ and $\mathbf{G}_{e,e'} = Y_e Y_{e'}^\top$. By the definition of the Tracy–Singh product, the $(e, e')$ block of $\mathbf{T}_\ell = \mathbf{A} \boxtimes \mathbf{G}$ is

$$\mathbf{T}_\ell^{(e,e')} \;=\; \mathbf{A}_{e,e'} \otimes \mathbf{G}_{e,e'} \;=\; (X_e X_{e'}^\top) \otimes (Y_e Y_{e'}^\top) \;=\; (X_e \otimes Y_e)(X_{e'} \otimes Y_{e'})^\top.$$

Stacking the block rows $Z_e := X_e \otimes Y_e$ as $Z_\ell^\top := [\, Z_1^\top, \ldots, Z_E^\top \,]$ yields

$$\mathbf{T}_\ell \;=\; Z_\ell Z_\ell^\top,$$

which is SPSD by construction.

For (ii), Liu et al. (2008, Section 2.2) show that, under the same symmetric block partition, the (block) Khatri–Rao product can be obtained from the Tracy–Singh product by a single selection matrix $S_\ell$ satisfying $S_\ell^\top S_\ell = I$, namely

$$\mathbf{A} * \mathbf{G} \;=\; S_\ell^\top \, \mathbf{T}_\ell \, S_\ell.$$

By Eq. (40), $\mathbf{G}_\ell^{\mathrm{GN,exp}}$ has exactly this block Khatri–Rao structure, so $\mathbf{G}_\ell^{\mathrm{GN,exp}} = \mathbf{A} * \mathbf{G}$ and is therefore a principal submatrix of $\mathbf{T}_\ell$ selected on expert-aligned blocks.

For (iii), Lemma 3 in Liu et al. (2008) states that the Tracy–Singh product is related to the Kronecker product via orthogonal permutations:

$$\mathbf{T}_\ell \;=\; P_1^\top \, \mathbf{K}_\ell \, P_2, \qquad P_1^\top P_1 = P_2^\top P_2 = I.$$

Here $P_1, P_2$ are permutation matrices, hence orthogonal ($P^\top P = I$), so this transformation preserves singular values. Writing the singular values as $\sigma_i(\cdot)$ and using $\mathbf{K}_\ell^\top = \mathbf{K}_\ell$,

$$
\begin{aligned}
\mathbf{T}_\ell^\top \mathbf{T}_\ell \;&=\; (P_1^\top \mathbf{K}_\ell P_2)^\top (P_1^\top \mathbf{K}_\ell P_2) \\
&=\; P_2^\top \, \mathbf{K}_\ell^\top \, P_1 \, P_1^\top \mathbf{K}_\ell \, P_2 \qquad (\text{since } (ABC)^\top = C^\top B^\top A^\top) \\
&=\; P_2^\top \, \mathbf{K}_\ell^\top \, (P_1 P_1^\top) \, \mathbf{K}_\ell \, P_2 \\
&=\; P_2^\top \, (\mathbf{K}_\ell^\top \mathbf{K}_\ell) \, P_2 \qquad (\text{since } P_1 P_1^\top = I \text{ for permutation } P_1),
\end{aligned}
$$

so $\mathbf{T}_\ell^\top \mathbf{T}_\ell$ and $\mathbf{K}_\ell^\top \mathbf{K}_\ell$ are orthogonally similar and therefore have identical eigenvalues; equivalently, $\sigma_i(\mathbf{T}_\ell) = \sigma_i(\mathbf{K}_\ell)$ for all $i$. Finally, since $\mathbf{A}$ and $\mathbf{G}$ are Gram matrices, both $\mathbf{T}_\ell$ (by part (i)) and $\mathbf{K}_\ell = \mathbf{A} \otimes \mathbf{G}$ are symmetric SPSD, and for symmetric SPSD matrices the singular values equal the eigenvalues. Hence $\{\lambda(\mathbf{T}_\ell)\} \equiv \{\lambda(\mathbf{K}_\ell)\}$ and $r_e(\mathbf{T}_\ell) = r_e(\mathbf{K}_\ell)$. $\square$

Thus the Kronecker proxy $\mathbf{K}_\ell = \mathbf{A}_{\ell-1}^{\mathrm{exp}} \otimes \mathbf{G}_\ell^{\mathrm{exp}}$ provides an SPSD spectral envelope for the expert-layer Gram $\mathbf{G}_\ell^{\mathrm{GN,exp}}$ via the Tracy–Singh factorization $Z_\ell Z_\ell^\top$ and the principal-submatrix relation in Proposition D.1. In particular, any spectral isotropy (scaled-identity structure) imposed on $\mathbf{K}_\ell$ immediately transfers to $\mathbf{T}_\ell$, and Cauchy interlacing for principal submatrices (Proposition D.2 below) then implies that the eigenvalues of $\mathbf{G}_\ell^{\mathrm{GN,exp}}$ are driven toward equality as well. Since SPSD matrices attain maximal spectral-entropy effective rank exactly at scaled identities (Proposition E.2), regularizing $\mathbf{A}_{\ell-1}^{\mathrm{exp}} \otimes \mathbf{G}_\ell^{\mathrm{exp}}$ toward isotropy maximizes the effective rank of both the Tracy–Singh envelope and its expert-layer principal submatrix.

**Proposition D.2** (Isotropy transfer and effective rank). *Let $M \in \mathbb{R}^{n \times n}$ be SPSD with eigenvalues $\lambda_1 \geq \cdots \geq \lambda_n \geq 0$ and let $B \in \mathbb{R}^{k \times k}$ be a principal submatrix of $M$, i.e., $B = P^\top M P$ for a coordinate-selection matrix $P \in \mathbb{R}^{n \times k}$ formed by selecting $k$ columns of $I_n$ (so $P^\top P = I_k$). Denote the spectral spread $\Delta(M) = \lambda_1 - \lambda_n$ and $\Delta(B) = \mu_1 - \mu_k$ where $\mu_1 \geq \cdots \geq \mu_k$ are the eigenvalues of $B$. Then $\Delta(B) \leq \Delta(M)$. Moreover, if a sequence of SPSD matrices $\{M_t\}$ satisfies $\Delta(M_t) \to 0$ and we write $B_t := P^\top M_t P$ for a fixed coordinate-selection matrix $P$, then $\Delta(B_t) \to 0$ and $\max_i |\mu_{i,t} - \lambda_{1,t}| \to 0$.*

*Proof.* By Cauchy interlacing for eigenvalues of principal submatrices (see, e.g., Horn & Johnson, 2012; Golub & Van Loan, 2013), if $M$ has ordered eigenvalues $\lambda_1 \geq \cdots \geq \lambda_n$ and $B$ is any $k \times k$ principal submatrix with ordered eigenvalues $\mu_1 \geq \cdots \geq \mu_k$, then

$$\lambda_i \geq \mu_i \geq \lambda_{i+n-k} \qquad \text{for } i = 1, \ldots, k.$$

In particular, for $i=1$ and $i=k$ this gives

$$\lambda_1 \geq \mu_1, \qquad \mu_k \geq \lambda_n,$$

and hence $\Delta(B) = \mu_1 - \mu_k \leq \lambda_1 - \lambda_n = \Delta(M)$, which is the first claim.

For the second claim, let $\{M_t\}$ be a sequence of SPSD matrices with eigenvalues $\{\lambda_{i,t}\}_{i=1}^n$ such that $\Delta(M_t) = \lambda_{1,t} - \lambda_{n,t} \to 0$. Fix a coordinate-selection matrix $P$ and define $B_t := P^\top M_t P$ with eigenvalues $\{\mu_{i,t}\}_{i=1}^k$. By interlacing applied to each $t$,

$$\lambda_{1,t} \geq \mu_{i,t} \geq \lambda_{n,t} \qquad \text{for all } i = 1, \ldots, k.$$

It follows that, for each $i$ and $t$,

$$0 \leq \lambda_{1,t} - \mu_{i,t} \leq \Delta(M_t), \qquad 0 \leq \mu_{i,t} - \lambda_{n,t} \leq \Delta(M_t),$$

so $\max_i |\mu_{i,t} - \lambda_{1,t}| \leq \Delta(M_t) \to 0$ and $\max_i |\mu_{i,t} - \lambda_{n,t}| \leq \Delta(M_t) \to 0$. In particular, $\Delta(B_t) = \mu_{1,t} - \mu_{k,t} \leq \Delta(M_t) \to 0$. If additionally $\lambda_{1,t}$ (equivalently $\lambda_{n,t}$) converges along the sequence (e.g., under trace normalization), then every $\mu_{i,t}$ converges to the same limit. $\square$

**Lemma D.3** (Interlacing-to-effective-rank bound). *Let $M \in \mathbb{R}^{n \times n}$ be symmetric positive definite (SPD) with eigenvalues $\lambda_1 \geq \cdots \geq \lambda_n > 0$ and let $B \in \mathbb{R}^{k \times k}$ be any principal submatrix of $M$ with eigenvalues $\mu_1 \geq \cdots \geq \mu_k > 0$. Then:*

1. *(Condition-number inheritance.) $\mu_1 \leq \lambda_1$ and $\mu_k \geq \lambda_n$, hence $\kappa(B) := \mu_1/\mu_k \leq \lambda_1/\lambda_n := \kappa(M)$.*

2. *(Explicit effective-rank lower bound.) The spectral-entropy effective rank satisfies*

$$r_e(B) \geq \frac{\text{Tr}(B)}{\mu_1} \geq \frac{k \mu_k}{\mu_1} \geq \frac{k \lambda_n}{\lambda_1} = \frac{k}{\kappa(M)}.$$

   *Moreover, equality in the final bound $r_e(B) = k/\kappa(M)$ holds if and only if $M$ is a scaled identity (equivalently $\kappa(M) = 1$), in which case $B$ is also a scaled identity and $r_e(B) = k$.*

*Proof.* Item (i) is immediate from Cauchy interlacing for principal submatrices: $\lambda_1 \geq \mu_1$ and $\mu_k \geq \lambda_n$, so $\kappa(B) = \mu_1/\mu_k \leq \lambda_1/\lambda_n = \kappa(M)$.

For (ii), define the normalized eigenvalues

$$p_i := \frac{\mu_i}{\text{Tr}(B)}, \qquad i = 1, \ldots, k.$$

Since $B$ is SPD, each $\mu_i > 0$, so $p_i > 0$ and $\sum_i p_i = 1$. Moreover, $\mu_1 \geq \cdots \geq \mu_k$ implies $p_1 = \max_i p_i$.

Let $H(p) := -\sum_{i=1}^k p_i \log p_i$ denote Shannon entropy. Since $p_i \leq p_1$ for all $i$, we have $\log p_i \leq \log p_1$, hence $-\log p_i \geq -\log p_1$ and therefore

$$H(p) = \sum_{i=1}^k p_i (-\log p_i) \geq \sum_{i=1}^k p_i (-\log p_1) = -\log p_1 \sum_{i=1}^k p_i = -\log p_1.$$

Since $B$ is SPD, its singular values equal its eigenvalues, so by the definition of spectral-entropy effective rank (Eq. (7)),

$$r_e(B) \ = \ \exp(H(p)) \ \geq \ \exp(-\log p_1) \ = \ \frac{1}{p_1} \ = \ \frac{\mathrm{Tr}(B)}{\mu_1}.$$

Next, since every $\mu_i \geq \mu_k$, we have

$$\mathrm{Tr}(B) = \sum_{i=1}^{k} \mu_i \ \geq \ \sum_{i=1}^{k} \mu_k \ = \ k\,\mu_k,$$

so $\mathrm{Tr}(B)/\mu_1 \geq k\,\mu_k/\mu_1$. Finally, by item (i), $\mu_k \geq \lambda_n$ and $\mu_1 \leq \lambda_1$, so

$$\frac{k\,\mu_k}{\mu_1} \ \geq \ \frac{k\,\lambda_n}{\lambda_1} \ = \ \frac{k}{\kappa(M)}.$$

Chaining these inequalities yields the claim. For the equality condition, note that if $r_e(B) = k/\kappa(M)$ then every inequality in the chain must be tight. Tightness of $r_e(B) \geq \mathrm{Tr}(B)/\mu_1$ and $\mathrm{Tr}(B) \geq k\mu_k$ forces $\mu_1 = \cdots = \mu_k$, so $B$ is a scaled identity and $r_e(B) = k$. Tightness of the final step then implies $\kappa(M) = 1$, i.e., $\lambda_1 = \lambda_n$ and hence $M$ is a scaled identity. The converse is immediate: if $M = \alpha\mathbf{I}_n$ then $B = \alpha\mathbf{I}_k$ and the chain holds with equality. $\square$

**Corollary D.3.1** (Proxy-to-Khatri–Rao effective-rank guarantee). *Under the notation of Proposition D.1, assume the Tracy–Singh envelope $\mathbf{T}_\ell$ is SPD and let $\mathbf{G}_\ell^{\mathrm{GN,exp}}$ be its expert-aligned principal submatrix. Let $k_\ell := \dim(\mathbf{G}_\ell^{\mathrm{GN,exp}})$. Since $\mathbf{T}_\ell$ and the Kronecker proxy $\tilde{\mathbf{G}}_\ell^{\mathrm{GN,exp}}$ have the same eigenvalues (Proposition D.1), their spectral condition numbers coincide:*

$$\kappa(\mathbf{T}_\ell) \ = \ \kappa(\tilde{\mathbf{G}}_\ell^{\mathrm{GN,exp}}).$$

*Therefore,*

$$r_e(\mathbf{G}_\ell^{\mathrm{GN,exp}}) \ \geq \ \frac{k_\ell}{\kappa(\tilde{\mathbf{G}}_\ell^{\mathrm{GN,exp}})} \ = \ \mathrm{LB}_\ell^{\mathrm{GN,exp}} \ = \ \frac{k_\ell}{\kappa(\mathbf{T}_\ell)}.$$

*If additionally $\mathbf{A}$ and $\mathbf{G}$ are SPD, then $\kappa(\mathbf{K}_\ell) = \kappa(\mathbf{A})\,\kappa(\mathbf{G})$ and hence*

$$r_e(\mathbf{G}_\ell^{\mathrm{GN,exp}}) \ \geq \ \frac{k_\ell}{\kappa(\mathbf{A})\,\kappa(\mathbf{G})}.$$

*Proof.* Since $\mathbf{G}_\ell^{\mathrm{GN,exp}}$ is an expert-aligned principal submatrix of the Tracy–Singh envelope $\mathbf{T}_\ell$, Lemma D.3 gives

$$r_e(\mathbf{G}_\ell^{\mathrm{GN,exp}}) \ \geq \ \frac{k_\ell}{\kappa(\mathbf{T}_\ell)}.$$

Proposition D.1(iii) implies $\kappa(\mathbf{T}_\ell) = \kappa(\tilde{\mathbf{G}}_\ell^{\mathrm{GN,exp}})$, yielding $r_e(\mathbf{G}_\ell^{\mathrm{GN,exp}}) \geq k_\ell/\kappa(\tilde{\mathbf{G}}_\ell^{\mathrm{GN,exp}}) = \mathrm{LB}_\ell^{\mathrm{GN,exp}}$. If additionally $\mathbf{A}$ and $\mathbf{G}$ are SPD, then by the Kronecker-product spectrum, $\kappa(\mathbf{K}_\ell) = \kappa(\mathbf{A})\,\kappa(\mathbf{G})$, giving the final bound. $\square$

# E. Effective-Rank Identities and Proofs

This section collects standard identities for the spectral-entropy effective rank and proves the technical lemmas referenced in the main derivations.

**Lemma E.1** (Kronecker effective rank). *Let $A$ and $B$ have nonzero singular values $\{a_i\}_i$ and $\{b_j\}_j$ with $z = \sum_i a_i$ and $w = \sum_j b_j$. Then*

$$r_e(A \otimes B) \ = \ r_e(A)\,r_e(B).$$

*Proof.* Singular values of $A \otimes B$ are $\{a_i b_j\}_{i,j}$ (see, e.g., Horn & Johnson, 2012; Golub & Van Loan, 2013). Normalizing,

$$p_{ij} := \frac{a_i b_j}{\sum_{i',j'} a_{i'} b_{j'}} \ = \ \frac{a_i}{z} \cdot \frac{b_j}{w} \ = \ p_i^A p_j^B.$$

Hence $H(p^{A \otimes B}) = -\sum_{i,j} p_{ij} \log p_{ij} = -\sum_i p_i^A \log p_i^A - \sum_j p_j^B \log p_j^B = H(p^A) + H(p^B)$. Exponentiating gives the claim. $\square$

**Corollary E.1.1** (Kronecker block effective rank). *If $\tilde{\mathbf{G}}_\ell^{\mathrm{GN,exp}}=\mathbf{A}_\ell^{\mathrm{exp}} \otimes \mathbf{G}_\ell^{\mathrm{exp}}$, then $r_e(\tilde{\mathbf{G}}_\ell^{\mathrm{GN,exp}})=r_e(\mathbf{A}_\ell^{\mathrm{exp}})\, r_e(\mathbf{G}_\ell^{\mathrm{exp}})$.*

*Proof.* Lemma E.1 applied to $A = \mathbf{A}_\ell^{\mathrm{exp}}$ and $B = \mathbf{G}_\ell^{\mathrm{exp}}$ yields

$$r_e(\tilde{\mathbf{G}}_\ell^{\mathrm{GN,exp}}) \;=\; r_e(\mathbf{A}_\ell^{\mathrm{exp}} \otimes \mathbf{G}_\ell^{\mathrm{exp}}) \;=\; r_e(\mathbf{A}_\ell^{\mathrm{exp}})\, r_e(\mathbf{G}_\ell^{\mathrm{exp}}).$$

$\square$

**Corollary E.1.2** (Monotonicity under the Kronecker proxy). *Assume $\tilde{\mathbf{G}}_\ell^{\mathrm{GN,exp}}=\mathbf{A}_\ell^{\mathrm{exp}} \otimes \mathbf{G}_\ell^{\mathrm{exp}}$. Then, for fixed $\mathbf{G}_\ell^{\mathrm{exp}}$, $r_e(\tilde{\mathbf{G}}_\ell^{\mathrm{GN,exp}})$ strictly increases with $r_e(\mathbf{A}_\ell^{\mathrm{exp}})$.*

*Proof.* Lemma E.1 gives $r_e(\tilde{\mathbf{G}}_\ell^{\mathrm{GN,exp}}) = r_e(\mathbf{A}_\ell^{\mathrm{exp}})\, r_e(\mathbf{G}_\ell^{\mathrm{exp}})$. For fixed $\mathbf{G}_\ell^{\mathrm{exp}}$ with $r_e(\mathbf{G}_\ell^{\mathrm{exp}}) > 0$, this is a positive constant times $r_e(\mathbf{A}_\ell^{\mathrm{exp}})$, hence it strictly increases as $r_e(\mathbf{A}_\ell^{\mathrm{exp}})$ increases. $\square$

*Proof of Lemma 4.1.* By the direct-sum spectral property (e.g., Horn & Johnson, 2012; Golub & Van Loan, 2013), the multiset of singular values of $M$ equals the multiset union of those of the blocks:

$$\{\sigma(M)\} \;\equiv\; \biguplus_{b=1}^{B}\{\sigma_{b,i}\}_i.$$

Let $s_b=\sum_i \sigma_{b,i}$ and $s=\sum_m s_m$; by assumption $s>0$. For each block define $\alpha_b=s_b/s$ and, when $s_b>0$, the within-block normalized spectrum $q_{b,i}=\sigma_{b,i}/s_b$ (for $s_b=0$, set $q_{b,i}=0$). Then the globally normalized singular-value distribution is

$$p_{b,i} \;=\; \frac{\sigma_{b,i}}{\sum_{k,j}\sigma_{k,j}} \;=\; \frac{\sigma_{b,i}}{s} \;=\; \frac{s_b}{s}\frac{\sigma_{b,i}}{s_b} \;=\; \alpha_b\, q_{b,i} \quad (\text{for } s_b>0),$$

with $\sum_b \alpha_b=1$ and, for all $b$ with $s_b>0$, $\sum_i q_{b,i}=1$. The Shannon entropy of $p$ decomposes as

$$\begin{aligned}
H(p) &:= -\sum_{b,i} p_{b,i}\log p_{b,i} = -\sum_{\{b:\, s_b>0\},i} \alpha_b q_{b,i}\big(\log \alpha_b + \log q_{b,i}\big)\\
&= -\sum_{\{b:\, s_b>0\}} \alpha_b \log\alpha_b \sum_i q_{b,i} \;-\; \sum_{\{b:\, s_b>0\}} \alpha_b \sum_i q_{b,i}\log q_{b,i}\\
&= H(\alpha) \;+\; \sum_{\{b:\, s_b>0\}} \alpha_b\, H(q_b),
\end{aligned}$$

where $H(\alpha)=-\sum_b \alpha_b \log\alpha_b$ and $H(q_b) := -\sum_i q_{b,i}\log q_{b,i}$ is the entropy of the normalized spectrum in block $b$. By definition $r_e(M_b)=\exp(H(q_b))$ for blocks with $s_b>0$, so exponentiating $H(p)$ yields

$$r_e(M) \;=\; \exp\Big(H(\alpha) + \sum_{\{b:\, s_b>0\}} \alpha_b\, \log r_e(M_b)\Big),$$

which is the claimed expression. $\square$

**Proposition E.2.** *Let $M \in \mathbb{R}^{m\times m}$ be positive semidefinite. Then $r_e(M) \le m$, with equality iff $M=\alpha\mathbf{I}_m$ for some $\alpha>0$ (i.e., all eigenvalues are equal).*

*Proof.* Let $\{\sigma_i(M)\}_{i=1}^m$ be the singular values and set $r := |\{i : \sigma_i(M)>0\}| = \mathrm{rank}(M)$. For indices with $\sigma_i>0$, define $p_i := \sigma_i/\sum_{j:\,\sigma_j>0} \sigma_j$. Then

$$r_e(M) \;=\; \exp\Big(-\sum_{\{i:\,\sigma_i>0\}} p_i \log p_i\Big) \;\le\; r\,,$$

with equality iff $p_i \equiv 1/r$ for all $i$ (Schur-concavity of entropy; Marshall et al., 1979), i.e., the nonzero singular values are equal. In particular, $r_e(M)=m$ iff $r=m$ (full rank) and $\sigma_1=\cdots=\sigma_m$. Writing the symmetric SVD $M=U\Sigma U^\top$ (SPSD implies $U=V$), if $\Sigma=\alpha\mathbf{I}_m$ then $M=U\Sigma U^\top=\alpha\mathbf{I}_m$. $\square$

# F. SPHERE as Spectral Contraction

Having established that the effective rank of the expert-layer block is lower-bounded by the condition number of the Kronecker proxy (Appendix D), we now show that SPHERE monotonically decreases this condition number, thereby increasing spectral plasticity.

The argument proceeds as follows. We first show that the SPHERE penalty can be written as the squared Frobenius distance to a scaled identity (Proposition F.1), and that gradient descent on this penalty contracts each eigenvalue toward the mean (Lemma F.2). This contraction monotonically decreases the condition number of the feature Gram (Lemma 4.3), which propagates through the Kronecker product to the proxy (Lemma 4.4). Combining these results yields the main guarantee: optimizing the SPHERE penalty increases a lower bound on $r_e(\mathbf{K})$ (Theorem 4.7).

## F.1. Proof of Proposition Expanding the SPHERE penalty

**Proposition F.1** (Expanding the SPHERE penalty). *Let $A \in \mathbb{R}^{m \times m}$ and define $\bar{\lambda} := \mathrm{Tr}(A)/m$. Then*

$$\left\| A - \bar{\lambda} \, \mathbf{I}_m \right\|_F^2 \;=\; \|A\|_F^2 - \frac{\mathrm{Tr}(A)^2}{m}.$$

*In particular, setting $A = \mathbf{A}_{\text{last}}^{\exp}$ gives the second equality in Eq. (26).*

*Proof.* Using $\|X\|_F^2 = \mathrm{Tr}(X^\top X)$ and $\mathrm{Tr}(\mathbf{I}_m) = m$,

$$\begin{aligned}
\left\| A - \bar{\lambda} \, \mathbf{I}_m \right\|_F^2 &= \mathrm{Tr}\Big( (A - \bar{\lambda} \, \mathbf{I}_m)^\top (A - \bar{\lambda} \, \mathbf{I}_m) \Big) \\
&= \mathrm{Tr}(A^\top A) - \bar{\lambda} \, \mathrm{Tr}(A^\top \mathbf{I}_m) - \bar{\lambda} \, \mathrm{Tr}(\mathbf{I}_m^\top A) + \bar{\lambda}^2 \, \mathrm{Tr}(\mathbf{I}_m^\top \mathbf{I}_m) \\
&= \|A\|_F^2 - 2\bar{\lambda} \, \mathrm{Tr}(A) + m\bar{\lambda}^2,
\end{aligned}$$

where we used $\mathrm{Tr}(A^\top \mathbf{I}_m) = \mathrm{Tr}(A^\top) = \mathrm{Tr}(A)$. Finally, substituting $\bar{\lambda} = \mathrm{Tr}(A)/m$ gives $-2\bar{\lambda} \, \mathrm{Tr}(A) + m\bar{\lambda}^2 = -\mathrm{Tr}(A)^2/m$, proving the claim. $\square$

## F.2. Proof of Lemma Spectral form and gradient direction of SPHERE

**Lemma F.2** (Spectral form and gradient direction of SPHERE). *Let $M_t \succeq 0$ be an SPSD matrix in $\mathbb{R}^{m \times m}$ with eigenvalues $\{\lambda_{i,t}\}_{i=1}^m$ and mean eigenvalue $\bar{\lambda}_t := \mathrm{Tr}(M_t)/m$. Then the SPHERE penalty (Eq. (26)) satisfies*

$$\mathcal{L}_{\text{SPHERE}}(M_t) \;=\; \left\| M_t - \bar{\lambda}_t \, \mathbf{I}_m \right\|_F^2.$$

*Moreover, its Euclidean gradient with respect to $M_t$ is*

$$\nabla_{M_t} \mathcal{L}_{\text{SPHERE}}(M_t) \;=\; 2\big( M_t - \bar{\lambda}_t \, \mathbf{I}_m \big),$$

*and a single gradient-descent step on $M_t$ with step size $\eta$ yields*

$$M_{t+1} \;=\; M_t - \eta \nabla_{M_t} \mathcal{L}_{\text{SPHERE}}(M_t) \;=\; (1 - 2\eta)M_t + 2\eta\bar{\lambda}_t \, \mathbf{I}_m,$$

*which preserves trace and contracts each eigenvalue toward the mean:*

$$\mathrm{Tr}(M_{t+1}) = \mathrm{Tr}(M_t), \qquad \lambda_{i,t+1} = \lambda_{i,t} - 2\eta(\lambda_{i,t} - \bar{\lambda}_t), \qquad \lambda_{i,t+1} - \bar{\lambda}_t = (1 - 2\eta)(\lambda_{i,t} - \bar{\lambda}_t) \quad \forall i.$$

*Consequently, $\mathcal{L}_{\text{SPHERE}}(M_{t+1}) = (1 - 2\eta)^2 \, \mathcal{L}_{\text{SPHERE}}(M_t)$.*

*Proof.* Let $M_t = U \, \mathrm{diag}(\lambda_{1,t}, \ldots, \lambda_{m,t})U^\top$ be an eigendecomposition with $U$ orthogonal. By Proposition F.1 applied to $A = M_t$ and $c = \bar{\lambda}_t$, we have

$$\left\| M_t - \bar{\lambda}_t \, \mathbf{I}_m \right\|_F^2 \;=\; \|M_t\|_F^2 - \frac{\mathrm{Tr}(M_t)^2}{m},$$

which is Eq. (26).

To obtain the matrix gradient, differentiate $\mathcal{L}_{\text{SPHERE}}(M) = \text{Tr}(M^2) - \text{Tr}(M)^2/m$ under the Frobenius inner product $\langle X, Y \rangle = \text{Tr}(X^\top Y)$. We have $d\,\text{Tr}(M^2) = 2\,\text{Tr}(M\,dM)$ and $d\,\text{Tr}(M) = \text{Tr}(dM)$, hence

$$\nabla_M \mathcal{L}_{\text{SPHERE}}(M) \;=\; 2M - \frac{2\,\text{Tr}(M)}{m}\mathbf{I}_m \;=\; 2(M - \bar{\lambda}\,\mathbf{I}_m).$$

Evaluating at $M = M_t$ yields $\nabla_{M_t}\mathcal{L}_{\text{SPHERE}}(M_t) = 2(M_t - \bar{\lambda}_t\,\mathbf{I}_m)$. Substituting into $M_{t+1} = M_t - \eta\nabla_{M_t}\mathcal{L}_{\text{SPHERE}}(M_t)$ yields $M_{t+1} = (1 - 2\eta)M_t + 2\eta\bar{\lambda}_t\,\mathbf{I}_m$. Taking traces gives $\text{Tr}(M_{t+1}) = \text{Tr}(M_t)$ and hence $\bar{\lambda}_{t+1} = \bar{\lambda}_t$. Finally, since $\mathbf{I}_m = U\mathbf{I}_m U^\top$, we have

$$M_{t+1} = U \,\text{diag}\big((1 - 2\eta)\lambda_{i,t} + 2\eta\bar{\lambda}_t\big) U^\top,$$

so $\lambda_{i,t+1} = (1 - 2\eta)\lambda_{i,t} + 2\eta\bar{\lambda}_t$ and thus $\lambda_{i,t+1} - \bar{\lambda}_t = (1 - 2\eta)(\lambda_{i,t} - \bar{\lambda}_t)$ for all $i$. Since $\bar{\lambda}_{t+1} = \bar{\lambda}_t$, we equivalently have $M_{t+1} - \bar{\lambda}_t\,\mathbf{I}_m = (1 - 2\eta)(M_t - \bar{\lambda}_t\,\mathbf{I}_m)$, and therefore

$$\mathcal{L}_{\text{SPHERE}}(M_{t+1}) = \big\|M_{t+1} - \bar{\lambda}_t\,\mathbf{I}_m\big\|_F^2 = (1 - 2\eta)^2\big\|M_t - \bar{\lambda}_t\,\mathbf{I}_m\big\|_F^2 = (1 - 2\eta)^2\,\mathcal{L}_{\text{SPHERE}}(M_t).$$

Finally, iterating the one-step relation gives

$$\mathcal{L}_{\text{SPHERE}}(M_t) = (1 - 2\eta)^{2t}\,\mathcal{L}_{\text{SPHERE}}(M_0),$$

so $\mathcal{L}_{\text{SPHERE}}(M_t) \to 0$ if and only if $0 < \eta < 1$ (it is constant when $\eta \in \{0, 1\}$ and diverges when $\eta < 0$ or $\eta > 1$). To obtain a *monotone* contraction of every eigenvalue toward the mean (Definition 4.3), it suffices to take $0 \leq \eta \leq \frac{1}{2}$ so that $\beta := 2\eta \in [0, 1]$. Indeed, for each $i$,

$$\lambda_{i,t+1} \;=\; (1 - \beta)\lambda_{i,t} + \beta\bar{\lambda}_t,$$

so $\lambda_{i,t+1}$ lies between $\lambda_{i,t}$ and $\bar{\lambda}_t$, and

$$|\lambda_{i,t+1} - \bar{\lambda}_t| \;=\; (1 - \beta)\,|\lambda_{i,t} - \bar{\lambda}_t| \;\leq\; |\lambda_{i,t} - \bar{\lambda}_t|.$$

In particular, $0 < \eta < \frac{1}{2}$ yields geometric contraction, while $\eta = \frac{1}{2}$ yields $M_{t+1} = \bar{\lambda}_t\mathbf{I}_m$ and hence $\mathcal{L}_{\text{SPHERE}}(M_{t+1}) = 0$ in one step. Moreover, since $M_t \succeq 0$ implies $\bar{\lambda}_t \geq 0$, the update $M_{t+1} = (1 - \beta)M_t + \beta\bar{\lambda}_t\mathbf{I}_m$ also preserves $M_{t+1} \succeq 0$. $\quad\square$

### F.3. Proof of Lemma Monotonicity of spectral contraction

*Proof of Lemma 4.3.* Fix a nonzero SPSD matrix $M_t \in \mathbb{R}^{m \times m}$ and assume $M_{t+1}$ is obtained from $M_t$ by a spectral contraction update in the sense of Definition 4.3. Then there exists $\beta \in [0, 1]$ such that, letting $\bar{\lambda} = \text{Tr}(M_t)/m$,

$$M_{t+1} \;=\; (1 - \beta)\,M_t \;+\; \beta\,\bar{\lambda}\,\mathbf{I}_m.$$

Let $M_t = U\,\text{diag}(\lambda_i)\,U^\top$ be an eigendecomposition. Since $\mathbf{I}_m = UIU^\top$, we have

$$M_{t+1} = U\,\text{diag}\big((1 - \beta)\lambda_i + \beta\bar{\lambda}\big) U^\top,$$

so $\text{Tr}(M_{t+1}) = \text{Tr}(M_t)$. For the condition number, if $\lambda_{\min} = 0$ then $\kappa(M_t) = +\infty$ and the claim holds trivially. Otherwise $\lambda_{\min} > 0$ and

$$\kappa(M_{t+1}) = \frac{(1 - \beta)\lambda_{\max} + \beta\bar{\lambda}}{(1 - \beta)\lambda_{\min} + \beta\bar{\lambda}} \leq \frac{\lambda_{\max}}{\lambda_{\min}} = \kappa(M_t),$$

where the inequality follows by cross-multiplying:

$$\lambda_{\min}\big((1 - \beta)\lambda_{\max} + \beta\bar{\lambda}\big) \;\leq\; \lambda_{\max}\big((1 - \beta)\lambda_{\min} + \beta\bar{\lambda}\big) \;\iff\; \beta\bar{\lambda}(\lambda_{\min} - \lambda_{\max}) \leq 0,$$

which holds since $\beta \in [0, 1]$, $\bar{\lambda} \geq 0$ (as $M_t \succeq 0$), and $\lambda_{\min} \leq \lambda_{\max}$. $\quad\square$

### F.4. Proof of Proposition SPHERE is spectrally contractive

**Corollary F.2.1** (SPHERE is spectrally contractive). *Assume the setting of Lemma F.2 with* $\mathrm{Tr}(A_t) > 0$ *and stepsize* $0 \leq \eta \leq \frac{1}{2}$. *Then the update*

$$A_{t+1} = (1 - 2\eta)A_t + 2\eta \frac{\mathrm{Tr}(A_t)}{m} \mathbf{I}_m$$

*is a spectral contraction in the sense of Definition 4.3 with coefficient* $\beta = 2\eta$. *In particular, Lemma 4.3 implies* $\kappa(A_{t+1}) \leq \kappa(A_t)$.

*Proof.* Lemma F.2 gives

$$A_{t+1} = (1 - 2\eta)A_t + 2\eta \frac{\mathrm{Tr}(A_t)}{m} \mathbf{I}_m.$$

Setting $\beta := 2\eta \in [0, 1]$ and $\bar{\lambda}_t := \mathrm{Tr}(A_t)/m$, this matches Definition 4.3: $A_{t+1} = (1 - \beta)A_t + \beta \bar{\lambda}_t \mathbf{I}_m$. Lemma 4.3 then yields $\kappa(A_{t+1}) \leq \kappa(A_t)$. $\square$

### F.5. Proof of Lemma Kronecker preservation of monotone $\kappa$

**Lemma F.3** (Kronecker preservation of monotone $\kappa$). *Let* $A_t, A_{t+1} \in \mathbb{R}^{m \times m}$ *and* $B_t \in \mathbb{R}^{n \times n}$ *be nonzero SPSD. Assume* $\kappa(A_{t+1}) \leq \kappa(A_t)$. *Define* $K_t := A_t \otimes B_t$ *and* $K_{t+1} := A_{t+1} \otimes B_t$. *Then*

$$\kappa(K_{t+1}) \leq \kappa(K_t).$$

*Proof.* If $A_t$ or $B_t$ is singular then $K_t = A_t \otimes B_t$ is singular and $\kappa(K_t) = +\infty$, so $\kappa(K_{t+1}) \leq \kappa(K_t)$ holds trivially. Otherwise $A_t \succ 0$ and $B_t \succ 0$, and the Kronecker-product spectrum gives

$$\kappa(K_t) = \kappa(A_t \otimes B_t) = \kappa(A_t)\,\kappa(B_t), \qquad \kappa(K_{t+1}) = \kappa(A_{t+1})\,\kappa(B_t).$$

Using $\kappa(A_{t+1}) \leq \kappa(A_t)$ yields $\kappa(K_{t+1}) \leq \kappa(K_t)$. $\square$

### F.6. Proof of Proposition Spectral contraction improves the lower bound

*Proof of Proposition 4.5.* By Lemma 4.3, a spectrally contractive update on $\mathbf{A}_{\ell-1,t}^{\mathrm{exp}}$ yields $\kappa(\mathbf{A}_{\ell-1,t+1}^{\mathrm{exp}}) \leq \kappa(\mathbf{A}_{\ell-1,t}^{\mathrm{exp}})$. Applying Lemma F.3 with $A_t = \mathbf{A}_{\ell-1,t}^{\mathrm{exp}}$ and $B_t = \mathbf{G}_\ell^{\mathrm{exp}}$ gives $\kappa(\tilde{\mathbf{G}}_{\ell,t+1}^{\mathrm{GN,exp}}) \leq \kappa(\tilde{\mathbf{G}}_{\ell,t}^{\mathrm{GN,exp}})$. Dividing $k_\ell > 0$ by the condition-number inequality yields $\mathrm{LB}_{\ell,t+1}^{\mathrm{GN,exp}} \geq \mathrm{LB}_{\ell,t}^{\mathrm{GN,exp}}$. $\square$

**Corollary F.3.1** (Spectral contraction increases the proxy lower bound). *Under the notation of Corollary D.3.1, assume the Kronecker proxy* $\tilde{\mathbf{G}}_\ell^{\mathrm{GN,exp}}$ *is SPD and define* $\mathrm{LB}_\ell^{\mathrm{GN,exp}} := k_\ell/\kappa(\tilde{\mathbf{G}}_\ell^{\mathrm{GN,exp}})$. *If* $\tilde{\mathbf{G}}_\ell^{\mathrm{GN,exp}}$ *is updated by a spectrally contractive regularizer (Definition 4.3), then* $\mathrm{LB}_\ell^{\mathrm{GN,exp}}$ *is non-decreasing. Moreover,* $\mathrm{LB}_\ell^{\mathrm{GN,exp}} \leq k_\ell$ *with equality if and only if* $\tilde{\mathbf{G}}_\ell^{\mathrm{GN,exp}}$ *is a scaled identity, which is also the unique maximizer of* $r_e(\tilde{\mathbf{G}}_\ell^{\mathrm{GN,exp}})$.

*Proof.* Let $\tilde{\mathbf{G}}_t$ and $\tilde{\mathbf{G}}_{t+1}$ denote the proxy before and after a spectrally contractive update. Lemma 4.3 gives $\kappa(\tilde{\mathbf{G}}_{t+1}) \leq \kappa(\tilde{\mathbf{G}}_t)$, hence

$$\mathrm{LB}_{\ell,t+1}^{\mathrm{GN,exp}} = \frac{k_\ell}{\kappa(\tilde{\mathbf{G}}_{t+1})} \geq \frac{k_\ell}{\kappa(\tilde{\mathbf{G}}_t)} = \mathrm{LB}_{\ell,t}^{\mathrm{GN,exp}}.$$

For any SPD matrix, $\kappa(\cdot) \geq 1$ with equality if and only if the matrix is a scaled identity, so $\mathrm{LB}_\ell^{\mathrm{GN,exp}} = k_\ell/\kappa(\tilde{\mathbf{G}}_\ell^{\mathrm{GN,exp}}) \leq k_\ell$ with equality iff $\tilde{\mathbf{G}}_\ell^{\mathrm{GN,exp}}$ is a scaled identity. Proposition E.2 gives the same condition as the unique maximizer of $r_e(\tilde{\mathbf{G}}_\ell^{\mathrm{GN,exp}})$. $\square$

**Corollary F.3.2** (Spectral contraction on $\mathbf{A}_{\ell-1}^{\mathrm{exp}}$ increases the target lower bound). *Under the notation of Corollary D.3.1, assume the Kronecker proxy factorizes as* $\tilde{\mathbf{G}}_\ell^{\mathrm{GN,exp}} = \mathbf{A}_{\ell-1}^{\mathrm{exp}} \otimes \mathbf{G}_\ell^{\mathrm{exp}}$ *with* $\mathbf{A}_{\ell-1}^{\mathrm{exp}} \succ 0$ *and* $\mathbf{G}_\ell^{\mathrm{exp}} \succ 0$. *Define the factorized proxy-to-target lower bound*

$$\mathrm{LB}_\ell^A = \frac{k_\ell}{\kappa(\mathbf{A}_{\ell-1}^{\mathrm{exp}})\,\kappa(\mathbf{G}_\ell^{\mathrm{exp}})}.$$

*Then $r_e(\mathbf{G}_\ell^{\mathrm{GN,exp}}) \geq \mathrm{LB}_\ell^A$. Moreover, if $\mathbf{A}_{\ell-1}^{\mathrm{exp}}$ is updated by a spectrally contractive regularizer in the sense of Definition 4.3 while keeping $\mathbf{G}_\ell^{\mathrm{exp}}$ fixed, then $\mathrm{LB}_\ell^A$ is non-decreasing under the update. Since $\kappa(\mathbf{A}_{\ell-1}^{\mathrm{exp}}) \geq 1$ for SPD matrices, $\mathrm{LB}_\ell^A \leq k_\ell/\kappa(\mathbf{G}_\ell^{\mathrm{exp}})$, and this upper bound is achieved if and only if $\mathbf{A}_{\ell-1}^{\mathrm{exp}}$ is a scaled identity.*

*Proof.* Since $\mathbf{A}_{\ell-1}^{\mathrm{exp}} \succ 0$ and $\mathbf{G}_\ell^{\mathrm{exp}} \succ 0$, the Kronecker proxy is SPD and satisfies $\kappa(\tilde{\mathbf{G}}_\ell^{\mathrm{GN,exp}}) = \kappa(\mathbf{A}_{\ell-1}^{\mathrm{exp}})\,\kappa(\mathbf{G}_\ell^{\mathrm{exp}})$. Substituting this identity into Corollary D.3.1 yields $r_e(\mathbf{G}_\ell^{\mathrm{GN,exp}}) \geq k_\ell/\kappa(\tilde{\mathbf{G}}_\ell^{\mathrm{GN,exp}}) = \mathrm{LB}_\ell^A$.

If a spectrally contractive update produces $\mathbf{A}_{\ell-1,t}^{\mathrm{exp}} \mapsto \mathbf{A}_{\ell-1,t+1}^{\mathrm{exp}}$ and $\mathbf{G}_\ell^{\mathrm{exp}}$ is fixed, then Lemma 4.3 implies $\kappa(\mathbf{A}_{\ell-1,t+1}^{\mathrm{exp}}) \leq \kappa(\mathbf{A}_{\ell-1,t}^{\mathrm{exp}})$, hence

$$\frac{k_\ell}{\kappa(\mathbf{A}_{\ell-1,t+1}^{\mathrm{exp}})\,\kappa(\mathbf{G}_\ell^{\mathrm{exp}})} \;\geq\; \frac{k_\ell}{\kappa(\mathbf{A}_{\ell-1,t}^{\mathrm{exp}})\,\kappa(\mathbf{G}_\ell^{\mathrm{exp}})},$$

so the bound is non-decreasing. Finally, for any SPD matrix, $\kappa(\mathbf{A}_{\ell-1}^{\mathrm{exp}}) \geq 1$ with equality iff $\mathbf{A}_{\ell-1}^{\mathrm{exp}}$ is a scaled identity, yielding $\mathrm{LB}_\ell^A \leq k_\ell/\kappa(\mathbf{G}_\ell^{\mathrm{exp}})$. $\qquad\square$

**Theorem F.4** (Spectral contraction on $\mathbf{A}_{\ell-1}^{\mathrm{exp}}$ increases an eNTK lower bound)**.** *Assume the surrogate model of Section 4: (i) $r_e(\mathbf{K}) = r_e(G^{\mathrm{GN}})$ (Eq. (10)); (ii) the block-diagonal approximation (15) holds; and (iii) for each expert layer $\ell$ and iteration $t$, the expert block admits the Kronecker proxy $\tilde{\mathbf{G}}_{\ell,t}^{\mathrm{GN,exp}} = \mathbf{A}_{\ell-1,t}^{\mathrm{exp}} \otimes \mathbf{G}_\ell^{\mathrm{exp}}$ with $\mathbf{A}_{\ell-1,t}^{\mathrm{exp}} \succ 0$ and $\mathbf{G}_\ell^{\mathrm{exp}} \succ 0$.*

*Fix the mixture weights $\alpha$ of Proposition 4.1, the gate-block terms $\{r_e(\mathbf{G}_\ell^{\mathrm{GN,g}})\}$, and the gradient-factor condition numbers $\{\kappa(\mathbf{G}_\ell^{\mathrm{exp}})\}$. Fix an iteration index $t$ and define the per-layer lower bounds*

$$\mathrm{LB}_{\ell,t}^A \;=\; \frac{k_\ell}{\kappa(\mathbf{A}_{\ell-1,t}^{\mathrm{exp}})\,\kappa(\mathbf{G}_\ell^{\mathrm{exp}})} \qquad (\text{Corollary F.3.2}),$$

*and the induced global lower bound*

$$\mathrm{LB}_{r_e(\mathbf{K}),t} \;=\; \exp\!\Big(H(\alpha) + \sum_{\{\ell:\, s_\ell^{\mathrm{g}} > 0\}} \alpha_\ell^{\mathrm{g}} \log r_e(\mathbf{G}_\ell^{\mathrm{GN,g}}) + \sum_{\{\ell:\, s_\ell^{\mathrm{exp}} > 0\}} \alpha_\ell^{\mathrm{exp}} \log \mathrm{LB}_{\ell,t}^A\Big).$$

*Then, under this surrogate, $r_e(\mathbf{K}) \gtrsim \mathrm{LB}_{r_e(\mathbf{K}),t}$. Moreover, if a spectrally contractive regularizer (Definition 4.3) updates some feature Gram $\mathbf{A}_{\ell_0-1,t}^{\mathrm{exp}} \mapsto \mathbf{A}_{\ell_0-1,t+1}^{\mathrm{exp}}$ while keeping all other quantities fixed as above, then*

$$\mathrm{LB}_{r_e(\mathbf{K}),t+1} \;\geq\; \mathrm{LB}_{r_e(\mathbf{K}),t}.$$

*Proof.* Under the block-diagonal approximation, Proposition 4.1 gives

$$r_e(\mathbf{K}) \;\approx\; \exp\!\Big(H(\alpha) + \sum_{\{\ell:\, s_\ell^{\mathrm{g}} > 0\}} \alpha_\ell^{\mathrm{g}} \log r_e(\mathbf{G}_\ell^{\mathrm{GN,g}}) + \sum_{\{\ell:\, s_\ell^{\mathrm{exp}} > 0\}} \alpha_\ell^{\mathrm{exp}} \log r_e(\mathbf{G}_\ell^{\mathrm{GN,exp}})\Big).$$

For each expert layer, Corollary F.3.2 provides the lower bound $r_e(\mathbf{G}_\ell^{\mathrm{GN,exp}}) \geq \mathrm{LB}_{\ell,t}^A$, hence $\log r_e(\mathbf{G}_\ell^{\mathrm{GN,exp}}) \geq \log \mathrm{LB}_{\ell,t}^A$. Substituting these inequalities into the mixture expression yields $r_e(\mathbf{K}) \gtrsim \mathrm{LB}_{r_e(\mathbf{K}),t}$.

Finally, if a spectrally contractive update on $\mathbf{A}_{\ell_0-1,t}^{\mathrm{exp}}$ is applied while $\kappa(\mathbf{G}_{\ell_0}^{\mathrm{exp}})$ is fixed, then Lemma 4.3 implies $\kappa(\mathbf{A}_{\ell_0-1,t+1}^{\mathrm{exp}}) \leq \kappa(\mathbf{A}_{\ell_0-1,t}^{\mathrm{exp}})$, and therefore $\mathrm{LB}_{\ell_0,t+1}^A \geq \mathrm{LB}_{\ell_0,t}^A$ by Corollary F.3.2. With all mixture weights and other terms fixed, the only term that changes from $t$ to $t+1$ inside $\log \mathrm{LB}_{r_e(\mathbf{K}),\cdot}$ is $\alpha_{\ell_0}^{\mathrm{exp}} \log \mathrm{LB}_{\ell_0,\cdot}^A$. Since $\mathrm{LB}_{\ell_0,t+1}^A \geq \mathrm{LB}_{\ell_0,t}^A$, we have $\log \mathrm{LB}_{r_e(\mathbf{K}),t+1} \geq \log \mathrm{LB}_{r_e(\mathbf{K}),t}$ and hence $\mathrm{LB}_{r_e(\mathbf{K}),t+1} \geq \mathrm{LB}_{r_e(\mathbf{K}),t}$. $\qquad\square$

### F.7. Proof of Theorem SPHERE improves spectral plasticity

*Proof of Theorem 4.7.* This is an immediate corollary of Theorem F.4. Instantiate Theorem F.4 at the expert output block so that the updated feature factor is $\mathbf{A}_{\mathrm{last},t}^{\mathrm{exp}}$ (i.e., $\mathbf{A}_{\ell_0-1,t}^{\mathrm{exp}} = \mathbf{A}_{\mathrm{last},t}^{\mathrm{exp}}$ in the notation of Theorem F.4), and note that under the surrogate model the tractable objective is $\hat{r}_e(\mathbf{K})_t = \mathrm{LB}_{r_e(\mathbf{K}),t}$. Then applying SPHERE gives $\mathrm{LB}_{r_e(\mathbf{K}),t+1} \geq \mathrm{LB}_{r_e(\mathbf{K}),t}$, i.e., $\hat{r}_e(\mathbf{K})_{t+1} \geq \hat{r}_e(\mathbf{K})_t$. $\qquad\square$

## G. Dense MLP as a Degenerate Top-$K$ MoE

The preceding analysis focuses on Top-$K$ MoE policies. A natural question is whether SPHERE applies to standard dense networks. This section shows that a dense MLP is a degenerate case of Top-$K$ MoE with $E=1$ and $K=1$, and that the SPHERE penalty remains a valid representation regularizer in this setting. Empirical results confirm that SPHERE improves dense PPO backbones.

**Setup.** Consider the Top-$K$ MoE output (Eq. (3)) with a single expert ($E=1$) and $K=1$. Then the selected set is $\mathcal{S}(x)=\{1\}$ for every input (hence the routing is constant), and the subset-softmax weight (Eq. (2)) is

$$h_1^{(1)}(x;\psi) = \frac{\exp(s_1(x;\psi))}{\sum_{j\in\{1\}}\exp(s_j(x;\psi))} = 1.$$

Therefore the MoE reduces identically to a dense network,

$$y(x;\Theta) = m_1(x;\theta_1)\, h_1^{(1)}(x;\psi) = m_1(x;\theta_1),$$

and the gate parameters $\psi$ do not affect the output.

**No gate gradients and eNTK reduction.** Since $h_1^{(1)}(x;\psi) \equiv 1$, we have $\partial h_1^{(1)}/\partial s_1 = 0$ and hence $\nabla_\psi h_1^{(1)}(x;\psi) = 0$ by the chain rule. Therefore $\nabla_\psi y(x;\Theta) = m_1(x;\theta_1)\,\nabla_\psi h_1^{(1)}(x;\psi) = 0$ for all $x$. Stacking over a minibatch $X = \{x_i\}_{i=1}^N$, the Jacobian block with respect to gate parameters is identically zero, $\mathbf{J}^{\mathrm{g}} = 0$. Writing the full Jacobian as $\mathbf{J} = [\,\mathbf{J}^{\mathrm{g}} \mid \mathbf{J}^{\mathrm{exp}}\,]$, the empirical NTK becomes

$$\mathbf{K} = \mathbf{J}\mathbf{J}^\top = \mathbf{J}^{\mathrm{exp}}(\mathbf{J}^{\mathrm{exp}})^\top,$$

which is exactly the eNTK of the dense MLP $m_1(\cdot;\theta_1)$. Equivalently, the Gauss–Newton surrogate satisfies

$$G^{\mathrm{GN}} = \tfrac{1}{N}\mathbf{J}^\top\mathbf{J} = \begin{bmatrix} 0 & 0 \\ 0 & \tfrac{1}{N}(\mathbf{J}^{\mathrm{exp}})^\top\mathbf{J}^{\mathrm{exp}} \end{bmatrix},$$

so the gate contributes only zero eigenvalues; since $r_e(\cdot)$ depends only on the strictly positive spectrum (Eq. (7)), these zeros do not affect $r_e(\mathbf{K})$.

**SPHERE remains applicable.** In this degenerate setting, the gating-weighted feature construction collapses to MLP features. Specifically, in Eq. (18) we have $h_{i,1}^{(1)} = 1$, so $a_{\ell-1}(x_i) = a_{1,\ell-1}^{\mathrm{exp}}(x_i)$ and $\mathbf{A}_{\ell-1}^{\mathrm{exp}}$ is the usual activation Gram at layer $\ell-1$. Under the same within-layer independence approximation used throughout, the corresponding Gauss–Newton block reduces to the dense-layer Kronecker form $\tilde{\mathbf{G}}_\ell^{\mathrm{GN}} = \mathbf{A}_{\ell-1}^{\mathrm{exp}} \otimes \mathbf{G}_\ell^{\mathrm{exp}}$. Therefore the SPHERE penalty on $\mathbf{A}_{\mathrm{last}}^{\mathrm{exp}}$ is a well-defined representation regularizer for dense MLPs. Since SPHERE is spectrally contractive (Corollary F.2.1), Proposition 4.5 applies verbatim in this $E=1$ setting and implies that regularizing $\mathbf{A}_{\mathrm{last}}^{\mathrm{exp}}$ monotonically improves the corresponding surrogate lower bound on $r_e(\mathbf{K})$. This reduction is consistent with the empirical gains from applying SPHERE to dense PPO backbones in Table 3.

**Empirical evidence.** We additionally evaluate this reduction directly by applying the same feature-Gram isotropy penalty to dense MLP backbones under CRL. Table 3 shows that SPHERE improves both PPO and PPO(10x), consistent with SPHERE acting as a representation regularizer that preserves eNTK spectral plasticity beyond MoE architectures.

*Table 3.* **SPHERE improves dense MLP backbones.** SPHERE is applied to PPO and PPO(10x) by regularizing the last-layer feature Gram matrix.

| Backbone | w/o SPHERE | w/ SPHERE |
|---|---|---|
| PPO | $0.36 \pm 0.12$ | $\mathbf{0.43 \pm 0.09}$ |
| PPO(10x) | $0.41 \pm 0.13$ | $\mathbf{0.50 \pm 0.15}$ |

## H. Rethinking MoE Through an eNTK Spectral Perspective

**Overview.** Section 4 connects Top-$K$ MoE plasticity loss to decay of the spectral-entropy effective rank $r_e(\mathbf{K})$ and introduces a tractable surrogate expressed in terms of expert-layer feature Grams $\{\mathbf{A}_{\ell-1}^{\mathrm{exp}}\}$ and gradient Grams $\{\mathbf{G}_\ell^{\mathrm{exp}}\}$. This appendix section provides a complementary interpretation: two common MoE objectives—*expert load balancing* and *expert specialization*—can be viewed as mechanisms that shape the spectrum of $\mathbf{A}_{\ell-1}^{\mathrm{exp}}$, and consequently affect the proxy condition number $\kappa(\tilde{\mathbf{G}}_\ell^{\mathrm{GN,exp}})=\kappa(\mathbf{A}_{\ell-1}^{\mathrm{exp}})\kappa(\mathbf{G}_\ell^{\mathrm{exp}})$ as well as the lower bounds in Corollary F.3.2 and Theorem F.4.

## H.1. The expert-feature spectrum depends jointly on routing and representations

Fix an expert layer $\ell$ and recall the concatenated gating-weighted expert feature vector $a_{\ell-1}^{\mathrm{ce}}(x_i) \in \mathbb{R}^{D_{\ell-1}}$ and its Gram $\mathbf{A}_{\ell-1}^{\exp}$ (Appendix C.3). Writing the expert-wise block decomposition explicitly,

$$\left[\mathbf{A}_{\ell-1}^{\exp}\right]_{e,e'} = \frac{1}{N} \sum_{i=1}^{N} h_{i,e}^{(K)} h_{i,e'}^{(K)} \, a_{e,\ell-1}^{\exp}(x_i) \, a_{e',\ell-1}^{\exp}(x_i)^{\top}. \tag{43}$$

Equation (43) makes explicit that the spectrum of $\mathbf{A}_{\ell-1}^{\exp}$ depends on two coupled components: *(i)* the routing weights $\{h_{i,e}^{(K)}\}$ induced by the gate on the current state distribution, and *(ii)* the expert representations $\{a_{e,\ell-1}^{\exp}(x_i)\}$. Accordingly, load balancing and specialization can be interpreted as spectral constraints on $\mathbf{A}_{\ell-1}^{\exp}$: they control the *distribution of blockwise spectral mass* and the *magnitude of cross-expert coupling*, which in turn influence both $r_e(\mathbf{A}_{\ell-1}^{\exp})$ and $\kappa(\mathbf{A}_{\ell-1}^{\exp})$.

## H.2. Load balancing redistributes block-level spectral mass

Load balancing can be formalized by quantifying how much of the feature Gram trace is contributed by each expert block. Let $\mathbf{A}_{\ell-1,(e,e)}^{\exp} := [\mathbf{A}_{\ell-1}^{\exp}]_{e,e}$ and define

$$\beta_e := \frac{\mathrm{Tr}(\mathbf{A}_{\ell-1,(e,e)}^{\exp})}{\mathrm{Tr}(\mathbf{A}_{\ell-1}^{\exp})}, \qquad e \in \{1, \ldots, E\},$$

whenever $\mathrm{Tr}(\mathbf{A}_{\ell-1}^{\exp}) > 0$.

**Lemma H.1** (Trace weights induced by routing). *For each expert $e$,*

$$\mathrm{Tr}(\mathbf{A}_{\ell-1,(e,e)}^{\exp}) = \frac{1}{N} \sum_{i=1}^{N} \left(h_{i,e}^{(K)}\right)^2 \left\|a_{e,\ell-1}^{\exp}(x_i)\right\|_2^2.$$

*Moreover,* $\mathrm{Tr}(\mathbf{A}_{\ell-1}^{\exp}) = \sum_{e=1}^{E} \mathrm{Tr}(\mathbf{A}_{\ell-1,(e,e)}^{\exp})$, *so* $\beta \in \Delta^{E-1}$.

*Proof.* By (43) with $e' = e$,

$$\mathbf{A}_{\ell-1,(e,e)}^{\exp} = \frac{1}{N} \sum_{i=1}^{N} \left(h_{i,e}^{(K)}\right)^2 a_{e,\ell-1}^{\exp}(x_i) \, a_{e,\ell-1}^{\exp}(x_i)^{\top}.$$

Taking traces and using $\mathrm{Tr}(vv^{\top}) = \|v\|_2^2$ gives the first claim. For the second claim, note that $\mathrm{Tr}(\mathbf{A}_{\ell-1}^{\exp})$ equals the sum of the traces of the diagonal blocks, and the definition of $\beta$ then yields $\sum_e \beta_e = 1$. $\square$

**Corollary H.1.1** (Inactive experts reduce the effective-rank ceiling). *Let* $d_{e,\ell-1}^{\exp} := \dim\left(a_{e,\ell-1}^{\exp}(x)\right)$ *and* $D_{\ell-1} = \sum_{e=1}^{E} d_{e,\ell-1}^{\exp}$. *If an expert $e^\star$ is never selected on the batch, i.e.,* $h_{i,e^\star}^{(K)} = 0$ *for all* $i \in \{1, \ldots, N\}$, *then*

$$r_e(\mathbf{A}_{\ell-1}^{\exp}) \leq \mathrm{rank}(\mathbf{A}_{\ell-1}^{\exp}) \leq D_{\ell-1} - d_{e^\star,\ell-1}^{\exp}.$$

*Proof.* By Lemma H.1, $h_{i,e^\star}^{(K)} \equiv 0$ implies $\mathrm{Tr}(\mathbf{A}_{\ell-1,(e^\star,e^\star)}^{\exp}) = 0$. If additionally $\mathrm{Tr}(\mathbf{A}_{\ell-1}^{\exp}) > 0$ (so $\beta$ is well-defined), then $\beta_{e^\star} = 0$. Moreover, since $a_{\ell-1}^{\mathrm{ce}}(x_i)$ concatenates the blocks $h_{i,e}^{(K)} a_{e,\ell-1}^{\exp}(x_i)$, the $e^\star$ block is identically zero across all samples, so the corresponding $d_{e^\star,\ell-1}^{\exp}$ columns of $\Phi_{\ell-1}$ are all zeros. Thus $\mathbf{A}_{\ell-1}^{\exp} = (1/N)\Phi_{\ell-1}^{\top}\Phi_{\ell-1}$ has a zero row/column block of size $d_{e^\star,\ell-1}^{\exp}$ and therefore $\mathrm{rank}(\mathbf{A}_{\ell-1}^{\exp}) \leq D_{\ell-1} - d_{e^\star,\ell-1}^{\exp}$. Finally, $r_e(M) \leq \mathrm{rank}(M)$ for any $M$ (Section 3), giving the claimed bound. $\square$

This gives a complementary motivation for load balancing: beyond flattening the trace weights $\beta$, it also helps avoid *inactive* experts that impose a hard ceiling on $r_e(\mathbf{A}_{\ell-1}^{\exp})$ (and thus on the expert-layer contribution to spectral plasticity).

**Corollary H.1.2** (Expert-block mixture form under decoupling). *Assume that the cross-expert blocks vanish, i.e.,* $[\mathbf{A}_{\ell-1}^{\exp}]_{e,e'} = 0$ *for all* $e \neq e'$ *(so* $\mathbf{A}_{\ell-1}^{\exp} = \bigoplus_{e=1}^{E} \mathbf{A}_{\ell-1,(e,e)}^{\exp}$). *Then*

$$r_e(\mathbf{A}_{\ell-1}^{\exp}) = \exp\left(H(\beta) + \sum_{\{e:\,\beta_e>0\}} \beta_e \log r_e\left(\mathbf{A}_{\ell-1,(e,e)}^{\exp}\right)\right). \tag{44}$$

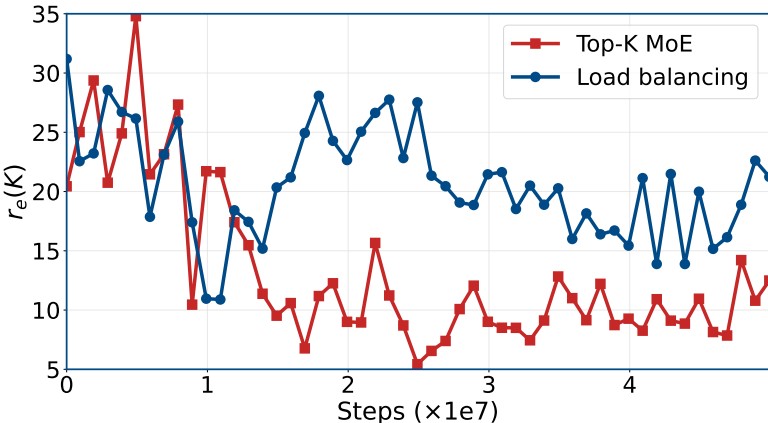

*Figure 8.* **Load balancing is positively associated with spectral plasticity.** The eNTK effective rank $r_e(\mathbf{K})$ is measured on Humanoid-Bench for a Top-$K$ MoE actor with and without a load-balancing objective. Load balancing maintains higher $r_e(\mathbf{K})$, consistent with the spectral view that redistributing routing-induced trace mass across experts helps prevent collapse of functional update directions.

*Table 4.* **Load balancing improves success alongside $r_e(\mathbf{K})$.** Under CRL on HumanoidBench, a load-balancing objective improves average final success and maintains a higher end-of-training $r_e(\mathbf{K})$. This supports the interpretation that load balancing benefits continual performance by preserving spectral plasticity.

| Method | Avg. final success under CRL | End-of-training $r_e(\mathbf{K})$ |
|---|---|---|
| Top-$K$ MoE | $0.36 \pm 0.08$ | 12.49 |
| + Load balancing | $\mathbf{0.45 \pm 0.23}$ | **21.24** |

*Proof.* Apply Lemma 4.1 to the block-diagonal matrix $\mathbf{A}_{\ell-1}^{\text{exp}} = \bigoplus_e \mathbf{A}_{\ell-1,(e,e)}^{\text{exp}}$. Lemma 4.1 uses mixture weights $\alpha_e = \|\mathbf{A}_{\ell-1,(e,e)}^{\text{exp}}\|_* / \|\mathbf{A}_{\ell-1}^{\text{exp}}\|_*$. Since each block is SPSD, $\|\mathbf{A}_{\ell-1,(e,e)}^{\text{exp}}\|_* = \text{Tr}(\mathbf{A}_{\ell-1,(e,e)}^{\text{exp}})$ and $\|\mathbf{A}_{\ell-1}^{\text{exp}}\|_* = \text{Tr}(\mathbf{A}_{\ell-1}^{\text{exp}})$, hence $\alpha_e = \beta_e$. Substituting $\alpha = \beta$ into Lemma 4.1 gives (44). $\square$

**Corollary H.1.3** (Balanced isotropic blocks maximize $r_e$ and minimize $\kappa$). *Assume the decoupling condition of Corollary H.1.2 and that all expert blocks have the same dimension d, with $\mathbf{A}_{\ell-1,(e,e)}^{\text{exp}} = \alpha_e \mathbf{I}_d$ for some $\alpha_e > 0$. Then*

$$r_e(\mathbf{A}_{\ell-1}^{\text{exp}}) = d \exp(H(\beta)), \qquad \kappa(\mathbf{A}_{\ell-1}^{\text{exp}}) = \frac{\max_e \alpha_e}{\min_e \alpha_e}, \qquad \beta_e = \frac{\alpha_e}{\sum_{m=1}^E \alpha_m}.$$

*In particular, $r_e(\mathbf{A}_{\ell-1}^{\text{exp}}) \leq dE$ with equality if and only if $\alpha_1 = \cdots = \alpha_E$, and $\kappa(\mathbf{A}_{\ell-1}^{\text{exp}}) \geq 1$ with equality if and only if $\alpha_1 = \cdots = \alpha_E$.*

*Proof.* If $\mathbf{A}_{\ell-1,(e,e)}^{\text{exp}} = \alpha_e \mathbf{I}_d$, then $r_e(\mathbf{A}_{\ell-1,(e,e)}^{\text{exp}}) = d$ and $\text{Tr}(\mathbf{A}_{\ell-1,(e,e)}^{\text{exp}}) = \alpha_e d$, hence $\beta_e = \alpha_e / \sum_m \alpha_m$. Substituting into (44) gives $r_e(\mathbf{A}_{\ell-1}^{\text{exp}}) = d \exp(H(\beta))$. Since $H(\beta) \leq \log E$ with equality iff $\beta$ is uniform, this implies $r_e(\mathbf{A}_{\ell-1}^{\text{exp}}) \leq dE$ with equality iff $\alpha_1 = \cdots = \alpha_E$. Finally, the eigenvalues of $\mathbf{A}_{\ell-1}^{\text{exp}}$ are $\{\alpha_e\}_{e=1}^E$ each with multiplicity $d$, giving $\kappa(\mathbf{A}_{\ell-1}^{\text{exp}}) = \max_e \alpha_e / \min_e \alpha_e$. $\square$

**Empirical link to $r_e(\mathbf{K})$.** Figure 8 empirically corroborates the above mechanism. A central observation is that load balancing acts *spectrally*: by preventing a few experts from dominating the trace of the gated feature Gram, it makes the trace weights $\beta$ less peaked, increasing $H(\beta)$ and thus increasing the block-mixture contribution to $r_e(\mathbf{A}_{\ell-1}^{\text{exp}})$ (cf. (44)). Through the Kronecker proxy in Section 4, this translates into a better-conditioned expert-layer surrogate and a larger expert-layer contribution to $r_e(\mathbf{K})$. Table 4 reports the corresponding improvement in final success alongside the higher end-of-training $r_e(\mathbf{K})$.

### H.3. Specialization attenuates cross-expert coupling

In practice, Top-$K$ routing induces nonzero off-diagonal blocks in (43) whenever two experts co-activate on the same samples. These cross blocks couple expert subspaces and can induce a more non-uniform spectrum, decreasing spectral

entropy (hence $r_e$) and typically worsening conditioning.

To formalize this, decompose $\mathbf{A}_{\ell-1}^{\exp}$ into its block-diagonal and off-diagonal parts:

$$\mathbf{A}_{\ell-1}^{\exp} \;=\; \mathbf{A}_{\ell-1,\mathrm{bd}}^{\exp} \;+\; \Delta_{\ell-1}, \qquad \mathbf{A}_{\ell-1,\mathrm{bd}}^{\exp} := \bigoplus_{e=1}^{E} [\mathbf{A}_{\ell-1}^{\exp}]_{e,e},$$

where $\Delta_{\ell-1}$ collects the cross-expert blocks.

**Proposition H.2** (Effective-rank deviation induced by cross-expert coupling). *Let $D := D_{\ell-1}$ and assume $\mathrm{Tr}(\mathbf{A}_{\ell-1}^{\exp}) > 0$. Define $\tau := \mathrm{Tr}(\mathbf{A}_{\ell-1}^{\exp})$ and*

$$\varepsilon_{\Delta} \;:=\; \frac{\sqrt{D}\,\|\Delta_{\ell-1}\|_F}{\tau}.$$

*If $\varepsilon_{\Delta} \le 1 - 1/D$, then*

$$\left| \log r_e(\mathbf{A}_{\ell-1}^{\exp}) - \log r_e(\mathbf{A}_{\ell-1,\mathrm{bd}}^{\exp}) \right| \;\le\; \varepsilon_{\Delta} \log(D-1) + h(\varepsilon_{\Delta}),$$

*where $h(\varepsilon) := -\varepsilon \log \varepsilon - (1-\varepsilon) \log(1-\varepsilon)$.*

*Proof.* Apply Lemma I.1 with $K = \mathbf{A}_{\ell-1}^{\exp}$ and $\widetilde{K} = \mathbf{A}_{\ell-1,\mathrm{bd}}^{\exp}$ (both are SPSD). Since $\Delta_{\ell-1}$ contains only off-diagonal blocks, $\mathrm{Tr}(\mathbf{A}_{\ell-1}^{\exp}) = \mathrm{Tr}(\mathbf{A}_{\ell-1,\mathrm{bd}}^{\exp}) = \tau$, and $\Delta = K - \widetilde{K} = \Delta_{\ell-1}$. In Lemma I.1, take $n = D$, so $\varepsilon = \sqrt{D}\|\Delta\|_F/\tau = \varepsilon_{\Delta}$, which yields the stated bound. $\qquad\square$

**Proposition H.3** (Condition-number control under cross-expert coupling). *Assume $\mathbf{A}_{\ell-1,\mathrm{bd}}^{\exp} \succ 0$ and $\|\Delta_{\ell-1}\|_2 < \lambda_{\min}(\mathbf{A}_{\ell-1,\mathrm{bd}}^{\exp})$. Then $\mathbf{A}_{\ell-1}^{\exp} \succ 0$ and*

$$\kappa(\mathbf{A}_{\ell-1}^{\exp}) \;\le\; \frac{\lambda_{\max}(\mathbf{A}_{\ell-1,\mathrm{bd}}^{\exp}) + \|\Delta_{\ell-1}\|_2}{\lambda_{\min}(\mathbf{A}_{\ell-1,\mathrm{bd}}^{\exp}) - \|\Delta_{\ell-1}\|_2} \,.$$

*Proof.* Weyl's inequalities give $\lambda_{\max}(\mathbf{A}_{\ell-1}^{\exp}) \le \lambda_{\max}(\mathbf{A}_{\ell-1,\mathrm{bd}}^{\exp}) + \|\Delta_{\ell-1}\|_2$ and $\lambda_{\min}(\mathbf{A}_{\ell-1}^{\exp}) \ge \lambda_{\min}(\mathbf{A}_{\ell-1,\mathrm{bd}}^{\exp}) - \|\Delta_{\ell-1}\|_2$. Under the stated assumption, the lower bound is positive, so $\mathbf{A}_{\ell-1}^{\exp} \succ 0$ and dividing yields the condition-number bound. $\quad\square$

### H.4. Summary and implications

This analysis yields a unified interpretation: load balancing and specialization act as complementary spectral constraints on $\mathbf{A}_{\ell-1}^{\exp}$. Balancing increases the block-level entropy term $H(\beta)$ by distributing trace mass across expert blocks, while specialization reduces cross-expert coupling $\Delta_{\ell-1}$ so that $r_e(\mathbf{A}_{\ell-1}^{\exp})$ is close to the block-diagonal approximation (Proposition H.2) and $\kappa(\mathbf{A}_{\ell-1}^{\exp})$ remains controlled (Proposition H.3). SPHERE operates directly on $\mathbf{A}_{\mathrm{last}}^{\exp}$ (Section 4.4), encouraging isotropy of the *gating-weighted* expert features; empirically, this is consistent with the strong correlation between $r_e(\mathbf{A}_{\mathrm{last}}^{\exp})$ and $r_e(\mathbf{K})$ (Figure 6).

## I. Perturbation Bounds for Approximation Errors

The surrogate analysis in Appendices D and F relies on block-diagonal and within-layer independence approximations. This section establishes perturbation bounds that quantify how approximation errors propagate to $r_e(\mathbf{K})$.

The development proceeds in two parts. First, Lemma I.1 proves that the effective rank is stable under Frobenius-norm perturbations: if the perturbation is small relative to the trace scale, then $r_e(\cdot)$ changes by at most a controlled multiplicative factor. Since $r_e(\mathbf{K}) = r_e(G^{\mathrm{GN}})$ (Eq. (10)), this directly controls changes in spectral plasticity. Second, we instantiate this general bound for the specific approximations used in our analysis: gate–expert decoupling (Corollary I.1.1), ignoring the gating contribution (Corollary I.1.2), and K-FAC factorization (Corollary I.1.3). The empirical diagnostics in Figures 13 and 15 verify that the coupling terms are small in practice.

**Effective-rank stability under matrix perturbations.** We control approximation error by bounding the total-variation distance between the normalized spectra of two SPSD matrices, which directly upper bounds the change in spectral entropy (hence $\log r_e$). *Remark.* If the perturbation is small in Frobenius norm relative to the trace scale, then $r_e(\cdot)$ is stable up to a multiplicative factor that grows smoothly with this ratio. These worst-case continuity bounds can be conservative; we use

them to make explicit which coupling terms must be small, and rely on the empirical diagnostics in Figures 13 and 15 to assess their magnitude in practice.

**Lemma I.1** (Stability of $r_e$ under Frobenius perturbations). *Let $K, \widetilde{K} \in \mathbb{R}^{n \times n}$ be SPSD with $\mathrm{Tr}(K) > 0$ and $\mathrm{Tr}(\widetilde{K}) > 0$. Let $\lambda(K) \in \mathbb{R}_+^n$ denote the vector of eigenvalues (with multiplicity) and define normalized spectra*

$$p := \frac{\lambda(K)}{\mathrm{Tr}(K)}, \qquad \widetilde{p} := \frac{\lambda(\widetilde{K})}{\mathrm{Tr}(\widetilde{K})}.$$

*Let $\tau := \min\{\mathrm{Tr}(K), \mathrm{Tr}(\widetilde{K})\}$ and $\Delta := K - \widetilde{K}$. Define*

$$\varepsilon := \frac{\sqrt{n}\, \|\Delta\|_F}{\tau}, \qquad h(\varepsilon) := -\varepsilon \log \varepsilon - (1 - \varepsilon) \log(1 - \varepsilon).$$

*If $\varepsilon \leq 1 - 1/n$, then*

$$\left| \log r_e(K) - \log r_e(\widetilde{K}) \right| = \left| H(p) - H(\widetilde{p}) \right| \leq \varepsilon \log(n - 1) + h(\varepsilon),$$

*and hence*

$$e^{-(\varepsilon \log(n-1) + h(\varepsilon))} \leq \frac{r_e(K)}{r_e(\widetilde{K})} \leq e^{\varepsilon \log(n-1) + h(\varepsilon)}.$$

*Proof.* Since $K \succeq 0$, all eigenvalues are nonnegative and $\sum_{i=1}^n \lambda_i(K) = \mathrm{Tr}(K) > 0$, so $p \in \Delta^{n-1}$ is a probability vector. The same holds for $\widetilde{p}$.

Let $\lambda := \lambda(K)$ and $\widetilde{\lambda} := \lambda(\widetilde{K})$, ordered non-increasingly. Since $K$ and $\widetilde{K}$ are symmetric, the Hoffman–Wielandt inequality gives

$$\|\lambda - \widetilde{\lambda}\|_2 \leq \|K - \widetilde{K}\|_F = \|\Delta\|_F.$$

By $\|v\|_1 \leq \sqrt{n}\|v\|_2$, we obtain

$$\|\lambda - \widetilde{\lambda}\|_1 \leq \sqrt{n}\, \|\Delta\|_F. \tag{45}$$

Next, since $\mathrm{Tr}(\Delta) = \langle \Delta, \mathbf{I}_n \rangle_F$ under the Frobenius inner product, Cauchy–Schwarz yields

$$\left| \mathrm{Tr}(K) - \mathrm{Tr}(\widetilde{K}) \right| = |\mathrm{Tr}(\Delta)| \leq \|\Delta\|_F \|\mathbf{I}_n\|_F = \sqrt{n}\, \|\Delta\|_F. \tag{46}$$

We now bound the total-variation distance between the normalized spectra. Using the triangle inequality and $\|\widetilde{\lambda}\|_1 = \mathrm{Tr}(\widetilde{K})$,

$$\begin{aligned}
\|p - \widetilde{p}\|_1 &= \left\| \frac{\lambda}{\mathrm{Tr}(K)} - \frac{\widetilde{\lambda}}{\mathrm{Tr}(\widetilde{K})} \right\|_1 \\
&\leq \left\| \frac{\lambda}{\mathrm{Tr}(K)} - \frac{\widetilde{\lambda}}{\mathrm{Tr}(K)} \right\|_1 + \left\| \frac{\widetilde{\lambda}}{\mathrm{Tr}(K)} - \frac{\widetilde{\lambda}}{\mathrm{Tr}(\widetilde{K})} \right\|_1 \\
&= \frac{\|\lambda - \widetilde{\lambda}\|_1}{\mathrm{Tr}(K)} + \|\widetilde{\lambda}\|_1 \cdot \left| \frac{1}{\mathrm{Tr}(K)} - \frac{1}{\mathrm{Tr}(\widetilde{K})} \right| \\
&= \frac{\|\lambda - \widetilde{\lambda}\|_1}{\mathrm{Tr}(K)} + \mathrm{Tr}(\widetilde{K}) \cdot \frac{\left| \mathrm{Tr}(K) - \mathrm{Tr}(\widetilde{K}) \right|}{\mathrm{Tr}(K)\,\mathrm{Tr}(\widetilde{K})} \\
&= \frac{\|\lambda - \widetilde{\lambda}\|_1}{\mathrm{Tr}(K)} + \frac{\left| \mathrm{Tr}(K) - \mathrm{Tr}(\widetilde{K}) \right|}{\mathrm{Tr}(K)} \\
&\leq \frac{\|\lambda - \widetilde{\lambda}\|_1}{\tau} + \frac{\left| \mathrm{Tr}(K) - \mathrm{Tr}(\widetilde{K}) \right|}{\tau}.
\end{aligned}$$

Substituting (45) and (46) gives

$$\|p - \widetilde{p}\|_1 \leq \frac{2\sqrt{n}\, \|\Delta\|_F}{\tau} = 2\varepsilon.$$

Define $\delta := \frac{1}{2}\|p - \widetilde{p}\|_1$. Then $\delta \leq \varepsilon$.

The Fannes–Audenaert inequality states that for probability vectors in dimension $n$, if $\delta \leq 1 - 1/n$ then

$$|H(p) - H(\widetilde{p})| \leq \delta \log(n - 1) + h(\delta).$$

Since $\varepsilon \leq 1 - 1/n$ by assumption and $\delta \leq \varepsilon$, the condition holds. Moreover, the function $g(x) := x \log(n - 1) + h(x)$ is non-decreasing on $[0, 1 - 1/n]$ because $g'(x) = \log\left(\frac{(n-1)(1-x)}{x}\right) \geq 0$ iff $x \leq 1 - 1/n$. Therefore,

$$|H(p) - H(\widetilde{p})| \leq \delta \log(n - 1) + h(\delta) \leq \varepsilon \log(n - 1) + h(\varepsilon).$$

Finally, since $K$ and $\widetilde{K}$ are SPSD, $r_e(K) = \exp(H(p))$ and $r_e(\widetilde{K}) = \exp(H(\widetilde{p}))$, hence $\left| \log r_e(K) - \log r_e(\widetilde{K}) \right| = \left| H(p) - H(\widetilde{p}) \right|$ and exponentiating yields the ratio bound. $\qquad \square$

**Instantiation for Top-$K$ MoE approximations.** Recall that $r_e(\mathbf{K}) = r_e(G^{\text{GN}})$ (Eq. (10)), where $G^{\text{GN}} := (1/N)\mathbf{J}^\top \mathbf{J}$ and $\mathbf{J} = [\,\mathbf{J}^{\text{g}} \mid \mathbf{J}^{\text{exp}}\,]$ partitions gate and expert parameters (Section 4.2). The next two corollaries upper bound the perturbation size $\|\Delta\|_F$ in terms of (i) the gate–expert cross block $\mathbf{G}^{\text{GN,g,exp}}$ and (ii) the gate block $\mathbf{G}^{\text{GN,g}}$.

**Corollary I.1.1** (Gate–expert decoupling error bound). *Let $G^{\text{GN}}$ be partitioned into gate/expert blocks as in Section 4.2:*

$$G^{\text{GN}} = \begin{bmatrix} \mathbf{G}^{\text{GN,g}} & \mathbf{G}^{\text{GN,g,exp}} \\ \mathbf{G}^{\text{GN,exp,g}} & \mathbf{G}^{\text{GN,exp}} \end{bmatrix}.$$

*Define the gate–expert block-diagonal approximation*

$$G^{\text{GN}}_{\text{bd}} := \begin{bmatrix} \mathbf{G}^{\text{GN,g}} & 0 \\ 0 & \mathbf{G}^{\text{GN,exp}} \end{bmatrix}.$$

*Then $\Delta_{\text{bd}} := G^{\text{GN}} - G^{\text{GN}}_{\text{bd}}$ satisfies*

$$\Delta_{\text{bd}} = \begin{bmatrix} 0 & \mathbf{G}^{\text{GN,g,exp}} \\ \mathbf{G}^{\text{GN,exp,g}} & 0 \end{bmatrix}, \qquad \|\Delta_{\text{bd}}\|_F = \sqrt{2}\,\|\mathbf{G}^{\text{GN,g,exp}}\|_F.$$

*Consequently, letting $P := P^{\text{g}} + P^{\text{exp}}$, $\tau_{\text{bd}} := \text{Tr}(G^{\text{GN}}) = \text{Tr}(G^{\text{GN}}_{\text{bd}})$ and $\varepsilon_{\text{bd}} := \sqrt{P}\|\Delta_{\text{bd}}\|_F/\tau_{\text{bd}}$, Lemma I.1 with $(K, \widetilde{K}) = (G^{\text{GN}}, G^{\text{GN}}_{\text{bd}})$ and $n = P$ gives, for $\varepsilon_{\text{bd}} \leq 1 - 1/P$,*

$$\left| \log r_e(G^{\text{GN}}) - \log r_e(G^{\text{GN}}_{\text{bd}}) \right| \leq \varepsilon_{\text{bd}} \log(P - 1) + h(\varepsilon_{\text{bd}}).$$

*Proof.* Subtracting the block-diagonal matrix $G^{\text{GN}}_{\text{bd}}$ from $G^{\text{GN}}$ yields the stated $\Delta_{\text{bd}}$. Since $\mathbf{G}^{\text{GN,exp,g}} = (\mathbf{G}^{\text{GN,g,exp}})^\top$, we have $\|\Delta_{\text{bd}}\|_F^2 = \|\mathbf{G}^{\text{GN,g,exp}}\|_F^2 + \|\mathbf{G}^{\text{GN,exp,g}}\|_F^2 = 2\|\mathbf{G}^{\text{GN,g,exp}}\|_F^2$. $\qquad \square$

**Corollary I.1.2** (Ignoring gating contribution). *Define the expert-only approximation*

$$G^{\text{GN}}_{\text{exp}} := \begin{bmatrix} 0 & 0 \\ 0 & \mathbf{G}^{\text{GN,exp}} \end{bmatrix}.$$

*Then $\Delta_{\text{exp}} := G^{\text{GN}} - G^{\text{GN}}_{\text{exp}}$ satisfies*

$$\Delta_{\text{exp}} = \begin{bmatrix} \mathbf{G}^{\text{GN,g}} & \mathbf{G}^{\text{GN,g,exp}} \\ \mathbf{G}^{\text{GN,exp,g}} & 0 \end{bmatrix}, \qquad \|\Delta_{\text{exp}}\|_F^2 = \|\mathbf{G}^{\text{GN,g}}\|_F^2 + 2\|\mathbf{G}^{\text{GN,g,exp}}\|_F^2.$$

*Consequently, letting $P := P^{\text{g}} + P^{\text{exp}}$, $\tau_{\text{exp}} := \min\{\text{Tr}(G^{\text{GN}}), \text{Tr}(G^{\text{GN}}_{\text{exp}})\}$ and $\varepsilon_{\text{exp}} := \sqrt{P}\|\Delta_{\text{exp}}\|_F/\tau_{\text{exp}}$, Lemma I.1 with $(K, \widetilde{K}) = (G^{\text{GN}}, G^{\text{GN}}_{\text{exp}})$ and $n = P$ gives, for $\varepsilon_{\text{exp}} \leq 1 - 1/P$,*

$$\left| \log r_e(G^{\text{GN}}) - \log r_e(G^{\text{GN}}_{\text{exp}}) \right| \leq \varepsilon_{\text{exp}} \log(P - 1) + h(\varepsilon_{\text{exp}}).$$

*Proof.* Subtracting $G_{\exp}^{\mathrm{GN}}$ from $G^{\mathrm{GN}}$ yields the stated $\Delta_{\exp}$. Summing the squared Frobenius norms of the disjoint blocks gives

$$\|\Delta_{\exp}\|_F^2 \;=\; \|\mathbf{G}^{\mathrm{GN,g}}\|_F^2 + \|\mathbf{G}^{\mathrm{GN,g,exp}}\|_F^2 + \|\mathbf{G}^{\mathrm{GN,exp,g}}\|_F^2.$$

Using $\mathbf{G}^{\mathrm{GN,exp,g}} = (\mathbf{G}^{\mathrm{GN,g,exp}})^\top$ yields $\|\mathbf{G}^{\mathrm{GN,exp,g}}\|_F = \|\mathbf{G}^{\mathrm{GN,g,exp}}\|_F$ and hence the stated identity. $\qquad\square$

**Corollary I.1.3** (K-FAC factorization error bound). *Let* $\mathbf{G}^{\mathrm{G\hat{N},exp}}_{\mathrm{out}} = \frac{1}{T}\sum_t (g_t g_t^\top) \otimes (a_t a_t^\top)$ *and* $\mathbf{G}^{\hat{\mathrm{exp}}}_{\mathrm{out}} = \frac{1}{T}\sum_t g_t g_t^\top$, $\mathbf{A}^{\hat{\mathrm{exp}}}_{\mathrm{last}} = \frac{1}{T}\sum_t a_t a_t^\top$ *(as in Figure 15). Then*

$$\left\|\mathbf{G}^{\mathrm{G\hat{N},exp}}_{\mathrm{out}} - \mathbf{G}^{\hat{\mathrm{exp}}}_{\mathrm{out}} \otimes \mathbf{A}^{\hat{\mathrm{exp}}}_{\mathrm{last}}\right\|_F \;\le\; \sqrt{\frac{1}{T}\sum_t \left\|g_t g_t^\top - \mathbf{G}^{\hat{\mathrm{exp}}}_{\mathrm{out}}\right\|_F^2} \cdot \sqrt{\frac{1}{T}\sum_t \left\|a_t a_t^\top - \mathbf{A}^{\hat{\mathrm{exp}}}_{\mathrm{last}}\right\|_F^2}.$$

*Remark. K-FAC is accurate when token-wise activation and gradient second-moment factors exhibit weak co-fluctuation across tokens. In particular, the factorization is exact when the centered factors $(g_t g_t^\top - \mathbf{G}^{\hat{\mathrm{exp}}}_{\mathrm{out}})$ and $(a_t a_t^\top - \mathbf{A}^{\hat{\mathrm{exp}}}_{\mathrm{last}})$ are uncorrelated in the Hilbert–Schmidt inner product. Moreover, for any perturbation $\Delta W$,*

$$\left|\langle \Delta W, \, (\mathbf{G}^{\mathrm{G\hat{N},exp}}_{\mathrm{out}} - \mathbf{G}^{\hat{\mathrm{exp}}}_{\mathrm{out}} \otimes \mathbf{A}^{\hat{\mathrm{exp}}}_{\mathrm{last}})\Delta W\rangle_F\right| \;\le\; \left\|\mathbf{G}^{\mathrm{G\hat{N},exp}}_{\mathrm{out}} - \mathbf{G}^{\hat{\mathrm{exp}}}_{\mathrm{out}} \otimes \mathbf{A}^{\hat{\mathrm{exp}}}_{\mathrm{last}}\right\|_F \|\Delta W\|_F^2,$$

*so the same bound controls the deviation between the GN and K-FAC directional curvatures used in Figure 15 (up to normalization by trace).*

*Proof.* Let $X_t = g_t g_t^\top$, $Y_t = a_t a_t^\top$, $\bar{X} := \mathbf{G}^{\hat{\mathrm{exp}}}_{\mathrm{out}} = \frac{1}{T}\sum_t X_t$, and $\bar{Y} := \mathbf{A}^{\hat{\mathrm{exp}}}_{\mathrm{last}} = \frac{1}{T}\sum_t Y_t$. Then

$$\mathbf{G}^{\mathrm{G\hat{N},exp}}_{\mathrm{out}} - \mathbf{G}^{\hat{\mathrm{exp}}}_{\mathrm{out}} \otimes \mathbf{A}^{\hat{\mathrm{exp}}}_{\mathrm{last}} \;=\; \frac{1}{T}\sum_t X_t \otimes Y_t - \bar{X} \otimes \bar{Y} \;=\; \frac{1}{T}\sum_t (X_t - \bar{X}) \otimes (Y_t - \bar{Y}),$$

where the last step follows by expanding $(X_t - \bar{X}) \otimes (Y_t - \bar{Y})$ and using $\sum_t (X_t - \bar{X}) = 0$ and $\sum_t (Y_t - \bar{Y}) = 0$. Then $\|(X_t - \bar{X}) \otimes (Y_t - \bar{Y})\|_F = \|X_t - \bar{X}\|_F \|Y_t - \bar{Y}\|_F$ and

$$\left\|\frac{1}{T}\sum_t (X_t - \bar{X}) \otimes (Y_t - \bar{Y})\right\|_F \le \frac{1}{T}\sum_t \|X_t - \bar{X}\|_F \|Y_t - \bar{Y}\|_F \le \sqrt{\frac{1}{T}\sum_t \|X_t - \bar{X}\|_F^2} \sqrt{\frac{1}{T}\sum_t \|Y_t - \bar{Y}\|_F^2},$$

by Cauchy–Schwarz. The quadratic-form bound follows from $|\langle U, MV\rangle_F| \le \|M\|_F \|U\|_F \|V\|_F$ with $U = V = \Delta W$. $\qquad\square$

**Connection to the Gauss–Newton blocks.** Since $\mathbf{G}^{\mathrm{GN,g,exp}} = (1/N)\mathbf{J}^{\mathrm{g}\top}\mathbf{J}^{\exp}$ and $\mathbf{G}^{\mathrm{GN,g}} = (1/N)\mathbf{J}^{\mathrm{g}\top}\mathbf{J}^{\mathrm{g}}$, the perturbation magnitudes in Corollaries I.1.1–I.1.2 are directly controlled by the gate–expert cross-block and the gate-block scale in $G^{\mathrm{GN}}$. In particular, the gate–expert block cosine used in Figure 13 implies

$$\|\mathbf{G}^{\mathrm{GN,g,exp}}\|_F \;=\; \cos(\mathrm{gate}, \mathrm{expert}) \cdot \sqrt{\|\mathbf{G}^{\mathrm{GN,g}}\|_F \|\mathbf{G}^{\mathrm{GN,exp}}\|_F},$$

so a small off-diagonal cosine provides a direct quantitative link between the empirical diagnostic and the perturbation terms that govern the induced change in $\log r_e(G^{\mathrm{GN}})$, hence in $\log r_e(\mathbf{K})$ by Eq. (10), via Lemma I.1.

# J. Additional Experimental Results

This section provides additional experimental results that complement the main paper: per-task success rates underlying the main-paper averages, additional ablations and robustness analyses, runtime overhead at large expert counts, and supervised continual-learning experiments.

## J.1. Additional Robustness Analyses

This subsection reports robust statistics for the main CRL comparisons.

### J.1.1. ROBUST STATISTICS FOR THE MAIN CRL COMPARISONS

Table 5 reports 10-seed robust statistics for the main CRL comparisons. SPHERE has higher mean success and IQM than the Top-$K$ MoE baseline in both benchmarks.

*Table 5.* **10-seed robust statistics for the main CRL comparisons.** We report mean $\pm$ std and robust aggregate statistics.

| Setting | Method | Mean $\pm$ std | IQM / IQR |
|---|---|---|---|
| HumanoidBench CRL | Top-$K$ MoE | $0.36 \pm 0.05$ | $0.36/0.05$ |
| HumanoidBench CRL | SPHERE | $\mathbf{0.53 \pm 0.07}$ | $\mathbf{0.52}/0.11$ |
| MetaWorld CRL | Top-$K$ MoE | $0.16 \pm 0.08$ | $0.14/0.14$ |
| MetaWorld CRL | SPHERE | $\mathbf{0.41 \pm 0.07}$ | $\mathbf{0.41}/0.07$ |

## J.2. Additional Baseline Results

### J.2.1. SPECTRAL AND PLASTICITY BASELINES

Table 6 compares SPHERE with additional spectral and plasticity baselines on HumanoidBench under CRL. Spectral normalization (Miyato et al., 2018; Bjorck et al., 2021), ReDo (Sokar et al., 2023), and spectral regularization (Lewandowski et al., 2025) improve over Top-$K$ MoE, while remaining below SPHERE.

*Table 6.* **Additional baselines on HumanoidBench under CRL.** All methods use the matched Top-$K$ MoE PPO setting.

| Method | Average success |
|---|---|
| Top-$K$ MoE | $0.36 \pm 0.08$ |
| Spectral Normalization | $0.41 \pm 0.08$ |
| ReDo | $0.41 \pm 0.04$ |
| Spectral Regularization | $0.48 \pm 0.06$ |
| SPHERE | $\mathbf{0.54 \pm 0.12}$ |

### J.2.2. EXPERT-CHOICE ROUTING

The main Top-$K$ MoE policy uses token-choice routing, with the control-step state representation serving as the routed token. Table 7 evaluates expert-choice routing (Zhou et al., 2022) on HumanoidBench under CRL. SPHERE improves the expert-choice baseline from $0.38 \pm 0.03$ to $0.44 \pm 0.05$.

*Table 7.* **SPHERE improves expert-choice routing on HumanoidBench under CRL.** We report average success across the five-task sequence.

| Method | Average success |
|---|---|
| Expert-choice routing | $0.38 \pm 0.03$ |
| Expert-choice routing + SPHERE | $\mathbf{0.44 \pm 0.05}$ |

## J.3. Plasticity Diagnostics

### J.3.1. PLASTICITY DIAGNOSTICS BEYOND $r_e(\mathbf{K})$

Beyond $r_e(\mathbf{K})$, Table 8 reports dormant ratio, feature norm, and gradient norm on HumanoidBench under CRL. SPHERE reduces dormant ratio and gradient norm while keeping feature norms comparable to the baseline.

*Table 8.* **Plasticity diagnostics beyond $r_e(\mathbf{K})$.** We report dormant ratio, feature L2 norm, and gradient L2 norm on HumanoidBench under CRL.

| Diagnostic | Top-$K$ MoE | SPHERE |
|---|---|---|
| Dormant ratio | 0.010 | 0.005 |
| Feature L2 norm | 269.60 | 269.80 |
| Gradient L2 norm | 312.70 | 255.40 |

## J.4. Architecture Robustness

### J.4.1. EXPERT-COUNT SWEEP

We vary the number of experts on HumanoidBench under CRL while keeping $K = 2$. Table 9 shows that SPHERE improves over the baseline across the tested expert counts.

*Table 9.* **Effect of the number of experts on HumanoidBench under CRL.** We keep $K = 2$ and vary the number of experts $E$.

| Number of experts | Top-$K$ MoE | SPHERE | Difference |
|:---:|:---:|:---:|:---:|
| 4 | $0.32 \pm 0.07$ | $0.45 \pm 0.08$ | $+0.13$ |
| 10 | $0.36 \pm 0.05$ | $0.53 \pm 0.07$ | $+0.17$ |
| 16 | $0.40 \pm 0.09$ | $0.59 \pm 0.10$ | $+0.19$ |

## J.5. Actor–Critic Placement

### J.5.1. ACTOR/CRITIC PLACEMENT

Table 10 compares actor-only, critic-only, and both-network application in PPO on HumanoidBench under CRL. Actor-only already outperforms critic-only, and applying SPHERE to both actor and critic performs best in this ablation. We use actor-only application as the default to isolate the policy-side effect and match the theoretical analysis.

*Table 10.* **Actor/critic placement in PPO on HumanoidBench under CRL.** Both-network application is strongest in this ablation.

| PPO placement | Average success |
|:---|:---:|
| Actor only | $0.54 \pm 0.12$ |
| Critic only | $0.41 \pm 0.06$ |
| Actor and critic | $\mathbf{0.57 \pm 0.07}$ |

## J.6. Interaction with Other Regularization

### J.6.1. LAYERNORM INTERACTION

Table 11 reports a small interaction check with layer normalization. In HumanoidBench CRL with PPO, adding LN leaves the mean unchanged and reduces variance; in MetaWorld CRL with TD3, adding LN yields a further gain.

*Table 11.* **Interaction with layer normalization.** We compare SPHERE alone with SPHERE plus LN.

| Setting | SPHERE only | SPHERE with LN |
|:---|:---:|:---:|
| HumanoidBench CRL with PPO | $0.54 \pm 0.12$ | $0.54 \pm 0.03$ |
| MetaWorld CRL with TD3 | $0.55 \pm 0.06$ | $0.65 \pm 0.09$ |

## J.7. Setting Robustness

### J.7.1. STATIONARY CONTROL AND LONGER-HORIZON RL

Table 12 reports a fixed-data stationary TD3+BC control and a longer-horizon MetaWorld RL chain, where SPHERE improves average success in both cases.

*Table 12.* **Stationary-control and longer-horizon checks.** We report average success in each setting.

| Setting | Baseline | SPHERE |
|:---|:---:|:---:|
| Stationary TD3+BC control | $0.46 \pm 0.09$ | $\mathbf{0.55 \pm 0.06}$ |
| Longer-horizon MetaWorld RL | $0.09 \pm 0.03$ | $\mathbf{0.21 \pm 0.04}$ |

## J.8. Per-Task Success Rates

We report full per-task success rates underlying the average summaries in the main paper, along with additional success-rate comparisons between RL and CRL for alternative baselines. For CRL, we report *per-task final success* evaluated on each task immediately after its own training segment (rather than evaluating the final policy on all tasks after completing the sequence) to eliminate the influence of catastrophic forgetting and isolate plasticity loss.

Throughout this appendix, HumanoidBench refers to five H1 locomotion tasks (stand, walk, pole, slide, and run) and MetaWorld refers to the CW10 subset from Continual World (Wołczyk et al., 2021) (hammer, push-wall, faucet-close, push-back, stick-pull, handle-press-side, push, shelf-place, window-close, and peg-unplug-side).

*Table 13.* **On MetaWorld-10 under RL, SPHERE yields the highest average success among Top-$K$ MoE variants.** Full per-task final success rates on MetaWorld-10 under RL for dense PPO, PPO(10x), and a Top-$K$ MoE actor with mitigation baselines. LN denotes applying LayerNorm to the Top-$K$ MoE actor. Figure 4 summarizes the corresponding averages.

| Task | PPO | PPO(10x) | Top-$K$ MoE | LN | SPHERE | C-CHAIN | CBP | PW |
|---|---|---|---|---|---|---|---|---|
| hammer | $0.76 \pm 0.39$ | $1.00 \pm 0.00$ | $0.98 \pm 0.04$ | $0.40 \pm 0.42$ | $0.96 \pm 0.08$ | $0.90 \pm 0.16$ | $1.00 \pm 0.00$ | $1.00 \pm 0.00$ |
| push-wall | $0.96 \pm 0.05$ | $0.80 \pm 0.28$ | $1.00 \pm 0.00$ | $0.25 \pm 0.38$ | $1.00 \pm 0.00$ | $0.78 \pm 0.35$ | $1.00 \pm 0.00$ | $0.90 \pm 0.13$ |
| faucet-close | $1.00 \pm 0.00$ | $1.00 \pm 0.00$ | $1.00 \pm 0.00$ | $1.00 \pm 0.00$ | $1.00 \pm 0.00$ | $1.00 \pm 0.00$ | $0.92 \pm 0.16$ | $1.00 \pm 0.00$ |
| push-back | $0.18 \pm 0.27$ | $0.33 \pm 0.47$ | $0.00 \pm 0.00$ | $0.23 \pm 0.39$ | $0.02 \pm 0.04$ | $0.20 \pm 0.40$ | $0.16 \pm 0.32$ | $0.00 \pm 0.00$ |
| stick-pull | $0.00 \pm 0.00$ | $0.00 \pm 0.00$ | $0.00 \pm 0.00$ | $0.00 \pm 0.00$ | $0.16 \pm 0.32$ | $0.00 \pm 0.00$ | $0.00 \pm 0.00$ | $0.00 \pm 0.00$ |
| handle-press-side | $1.00 \pm 0.00$ | $1.00 \pm 0.00$ | $1.00 \pm 0.00$ | $1.00 \pm 0.00$ | $1.00 \pm 0.00$ | $1.00 \pm 0.00$ | $1.00 \pm 0.00$ | $1.00 \pm 0.00$ |
| push | $0.68 \pm 0.34$ | $0.83 \pm 0.17$ | $0.82 \pm 0.27$ | $0.57 \pm 0.12$ | $1.00 \pm 0.00$ | $0.94 \pm 0.08$ | $0.96 \pm 0.05$ | $0.96 \pm 0.05$ |
| shelf-place | $0.00 \pm 0.00$ | $0.00 \pm 0.00$ | $0.00 \pm 0.00$ | $0.00 \pm 0.00$ | $0.00 \pm 0.00$ | $0.00 \pm 0.00$ | $0.00 \pm 0.00$ | $0.00 \pm 0.00$ |
| window-close | $1.00 \pm 0.00$ | $1.00 \pm 0.00$ | $1.00 \pm 0.00$ | $1.00 \pm 0.00$ | $1.00 \pm 0.00$ | $1.00 \pm 0.00$ | $1.00 \pm 0.00$ | $1.00 \pm 0.00$ |
| peg-unplug-side | $0.92 \pm 0.12$ | $0.65 \pm 0.35$ | $0.90 \pm 0.13$ | $0.83 \pm 0.24$ | $1.00 \pm 0.00$ | $0.84 \pm 0.23$ | $0.84 \pm 0.08$ | $0.86 \pm 0.15$ |
| average | $0.65 \pm 0.12$ | $0.66 \pm 0.13$ | $0.67 \pm 0.04$ | $0.53 \pm 0.16$ | $\mathbf{0.71 \pm 0.04}$ | $0.67 \pm 0.12$ | $0.69 \pm 0.06$ | $0.67 \pm 0.03$ |

*Table 14.* **On MetaWorld-10 under CRL, SPHERE yields the highest average success among the evaluated methods.** Full per-task final success rates on MetaWorld-10 under CRL for dense PPO, PPO(10x), and a Top-$K$ MoE actor with mitigation baselines. LN denotes applying LayerNorm to the Top-$K$ MoE actor. Figure 4 summarizes the corresponding averages.

| Task | PPO | PPO(10x) | Top-$K$ MoE | LN | SPHERE | PW | C-CHAIN | CBP |
|---|---|---|---|---|---|---|---|---|
| hammer | $1.00 \pm 0.00$ | $0.97 \pm 0.05$ | $0.98 \pm 0.04$ | $0.48 \pm 0.43$ | $0.98 \pm 0.04$ | $1.00 \pm 0.00$ | $0.80 \pm 0.40$ | $0.60 \pm 0.45$ |
| push-wall | $0.20 \pm 0.40$ | $0.00 \pm 0.00$ | $0.00 \pm 0.00$ | $0.24 \pm 0.39$ | $0.70 \pm 0.25$ | $0.00 \pm 0.00$ | $0.04 \pm 0.08$ | $0.18 \pm 0.36$ |
| faucet-close | $0.14 \pm 0.23$ | $0.67 \pm 0.47$ | $0.38 \pm 0.47$ | $1.00 \pm 0.00$ | $1.00 \pm 0.00$ | $0.70 \pm 0.38$ | $0.38 \pm 0.38$ | $1.00 \pm 0.00$ |
| push-back | $0.00 \pm 0.00$ | $0.00 \pm 0.00$ | $0.00 \pm 0.00$ | $0.14 \pm 0.28$ | $0.00 \pm 0.00$ | $0.00 \pm 0.00$ | $0.00 \pm 0.00$ | $0.18 \pm 0.36$ |
| stick-pull | $0.00 \pm 0.00$ | $0.00 \pm 0.00$ | $0.00 \pm 0.00$ | $0.00 \pm 0.00$ | $0.00 \pm 0.00$ | $0.00 \pm 0.00$ | $0.00 \pm 0.00$ | $0.00 \pm 0.00$ |
| handle-press-side | $0.50 \pm 0.42$ | $0.45 \pm 0.45$ | $0.50 \pm 0.45$ | $0.82 \pm 0.36$ | $0.98 \pm 0.04$ | $0.62 \pm 0.39$ | $0.60 \pm 0.45$ | $0.70 \pm 0.40$ |
| push | $0.00 \pm 0.00$ | $0.05 \pm 0.05$ | $0.00 \pm 0.00$ | $0.46 \pm 0.41$ | $0.22 \pm 0.17$ | $0.00 \pm 0.00$ | $0.00 \pm 0.00$ | $0.06 \pm 0.08$ |
| shelf-place | $0.00 \pm 0.00$ | $0.00 \pm 0.00$ | $0.00 \pm 0.00$ | $0.00 \pm 0.00$ | $0.00 \pm 0.00$ | $0.00 \pm 0.00$ | $0.00 \pm 0.00$ | $0.00 \pm 0.00$ |
| window-close | $0.00 \pm 0.00$ | $0.00 \pm 0.00$ | $0.20 \pm 0.40$ | $0.62 \pm 0.43$ | $1.00 \pm 0.00$ | $0.03 \pm 0.04$ | $0.40 \pm 0.49$ | $0.20 \pm 0.40$ |
| peg-unplug-side | $0.00 \pm 0.00$ | $0.00 \pm 0.00$ | $0.00 \pm 0.00$ | $0.00 \pm 0.00$ | $0.02 \pm 0.04$ | $0.00 \pm 0.00$ | $0.02 \pm 0.04$ | $0.00 \pm 0.00$ |
| average | $0.18 \pm 0.11$ | $0.21 \pm 0.10$ | $0.21 \pm 0.14$ | $0.38 \pm 0.23$ | $\mathbf{0.49 \pm 0.06}$ | $0.23 \pm 0.08$ | $0.22 \pm 0.18$ | $0.29 \pm 0.21$ |

*Table 15.* **Relative to Top-$K$ MoE, SPHERE improves average success by 36% under RL.** Full per-task final success rates on HumanoidBench under RL with ReLU activations. Methods include dense PPO, PPO(10x), and MoE baselines. LN denotes applying LayerNorm to the Top-$K$ MoE actor. Mitigation baselines include SPHERE, PW, C-CHAIN, and CBP, applied to the Top-$K$ MoE actor.

| Task | PPO | PPO(10x) | Top-$K$ MoE | LN | Dense-MoE | DS-MoE | SPHERE | PW | C-CHAIN | CBP |
|---|---|---|---|---|---|---|---|---|---|---|
| stand | $0.74 \pm 0.20$ | $0.88 \pm 0.12$ | $0.84 \pm 0.22$ | $0.85 \pm 0.05$ | $0.87 \pm 0.05$ | $0.77 \pm 0.17$ | $0.95 \pm 0.05$ | $0.68 \pm 0.15$ | $0.90 \pm 0.08$ | $0.83 \pm 0.13$ |
| walk | $0.62 \pm 0.12$ | $0.76 \pm 0.39$ | $0.46 \pm 0.41$ | $1.00 \pm 0.00$ | $0.57 \pm 0.40$ | $0.73 \pm 0.13$ | $0.75 \pm 0.13$ | $0.83 \pm 0.15$ | $0.53 \pm 0.41$ | $0.50 \pm 0.41$ |
| pole | $0.60 \pm 0.09$ | $0.66 \pm 0.14$ | $0.58 \pm 0.20$ | $0.75 \pm 0.15$ | $0.60 \pm 0.40$ | $0.67 \pm 0.13$ | $0.83 \pm 0.13$ | $0.73 \pm 0.12$ | $0.87 \pm 0.05$ | $0.63 \pm 0.45$ |
| slide | $0.00 \pm 0.00$ | $0.00 \pm 0.00$ | $0.00 \pm 0.00$ | $0.00 \pm 0.00$ | $0.00 \pm 0.00$ | $0.00 \pm 0.00$ | $0.33 \pm 0.17$ | $0.23 \pm 0.19$ | $0.00 \pm 0.00$ | $0.00 \pm 0.00$ |
| run | $0.76 \pm 0.38$ | $0.78 \pm 0.15$ | $0.88 \pm 0.13$ | $0.45 \pm 0.45$ | $0.85 \pm 0.05$ | $0.85 \pm 0.15$ | $0.90 \pm 0.08$ | $0.65 \pm 0.19$ | $0.57 \pm 0.40$ | $0.50 \pm 0.50$ |
| average | $0.54 \pm 0.16$ | $0.62 \pm 0.16$ | $0.55 \pm 0.19$ | $0.61 \pm 0.13$ | $0.58 \pm 0.18$ | $0.60 \pm 0.11$ | $\mathbf{0.75 \pm 0.11}$ | $0.62 \pm 0.16$ | $0.57 \pm 0.19$ | $0.49 \pm 0.30$ |

*Table 16.* **Relative to Top-$K$ MoE, SPHERE improves average success by 50% under CRL.** Full per-task final success rates on HumanoidBench under CRL with ReLU activations. Methods include dense PPO, PPO(10x), and MoE baselines. LN denotes applying LayerNorm to the Top-$K$ MoE actor. Mitigation baselines include SPHERE, PW, C-CHAIN, and CBP, applied to the Top-$K$ MoE actor.

| Task | PPO | PPO(10x) | Top-$K$ MoE | LN | Dense-MoE | DS-MoE | SPHERE | PW | C-CHAIN | CBP |
|---|---|---|---|---|---|---|---|---|---|---|
| stand | $0.74 \pm 0.20$ | $0.88 \pm 0.12$ | $0.84 \pm 0.22$ | $0.82 \pm 0.08$ | $0.90 \pm 0.00$ | $0.77 \pm 0.17$ | $0.85 \pm 0.18$ | $0.70 \pm 0.16$ | $0.90 \pm 0.08$ | $0.83 \pm 0.13$ |
| walk | $0.78 \pm 0.13$ | $0.74 \pm 0.22$ | $0.92 \pm 0.08$ | $1.00 \pm 0.00$ | $0.90 \pm 0.00$ | $0.83 \pm 0.09$ | $0.98 \pm 0.04$ | $0.83 \pm 0.09$ | $0.73 \pm 0.19$ | $0.90 \pm 0.00$ |
| pole | $0.26 \pm 0.26$ | $0.42 \pm 0.29$ | $0.06 \pm 0.08$ | $0.71 \pm 0.17$ | $0.05 \pm 0.05$ | $0.33 \pm 0.34$ | $0.38 \pm 0.18$ | $0.40 \pm 0.22$ | $0.23 \pm 0.26$ | $0.53 \pm 0.41$ |
| slide | $0.00 \pm 0.00$ | $0.00 \pm 0.00$ | $0.00 \pm 0.00$ | $0.00 \pm 0.00$ | $0.00 \pm 0.00$ | $0.00 \pm 0.00$ | $0.03 \pm 0.04$ | $0.07 \pm 0.09$ | $0.00 \pm 0.00$ | $0.00 \pm 0.00$ |
| run | $0.00 \pm 0.00$ | $0.00 \pm 0.00$ | $0.00 \pm 0.00$ | $0.00 \pm 0.00$ | $0.00 \pm 0.00$ | $0.00 \pm 0.00$ | $0.47 \pm 0.17$ | $0.00 \pm 0.00$ | $0.20 \pm 0.20$ | $0.00 \pm 0.00$ |
| average | $0.36 \pm 0.12$ | $0.41 \pm 0.13$ | $0.36 \pm 0.08$ | $0.50 \pm 0.05$ | $0.37 \pm 0.01$ | $0.39 \pm 0.12$ | $\mathbf{0.54 \pm 0.12}$ | $0.40 \pm 0.11$ | $0.41 \pm 0.15$ | $0.45 \pm 0.11$ |

### J.8.1. A CONTINUAL-LEARNING SUMMARY METRIC ON HUMANOIDBENCH (FIRST FIVE TASKS)

**Average success of the final policy.** Per-task final success reveals which tasks benefit most, but it can obscure the overall plasticity–retention trade-off in a continual-learning sequence. We therefore report a complementary single-number metric: after sequentially training on the first five HumanoidBench tasks, we evaluate the final checkpoint on Stand, Walk, Pole, Slide, and Run and average success across tasks. This five-task average directly reflects both adaptation to later tasks and retention on earlier tasks in one summary score. Importantly, the improvement does not come at the cost of increased forgetting: SPHERE improves the five-task average while maintaining comparable and often higher success on earlier tasks, as shown in Table 16.

*Table 17.* **Accounting for both adaptation and retention, SPHERE improves the final five-task average success.** The final checkpoint after completing the five-task sequence is evaluated on Stand, Walk, Pole, Slide, and Run and averaged across tasks.

| Method | Avg. over five tasks |
|---|---|
| Top-$K$ MoE | $0.03 \pm 0.04$ |
| SPHERE | $\mathbf{0.25 \pm 0.05}$ |

### J.8.2. ENTK EFFECTIVE RANK ON ONLINE STATES

The main paper tracks spectral plasticity using $r_e(\mathbf{K})$ computed on a fixed held-out state batch. As a robustness check, we also compute $r_e(\mathbf{K})$ on *online* states sampled from the rollout buffer at each logging step. Figure 9 shows that the same qualitative trend holds: Top-$K$ MoE exhibits a decay in $r_e(\mathbf{K})$ over training, while SPHERE maintains higher effective rank throughout.

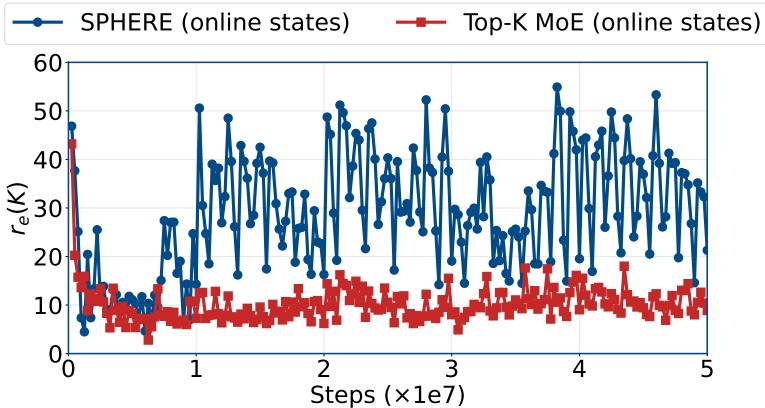

*Figure 9.* **The spectral-plasticity trend persists when $r_e(\mathbf{K})$ is computed on online rollout states.** Top-$K$ MoE exhibits effective-rank decay, while SPHERE maintains a higher effective rank throughout training.

## J.9. Additional Ablation Experiments

### J.9.1. SPHERE Design Ablations

**Adaptive Target Scale.** SPHERE uses the batch-dependent isotropic reference scale $\alpha = \mathrm{Tr}(\mathbf{A}_{\text{last}}^{\text{exp}})/m$ (Eq. (26)), so the penalty depends only on anisotropy and does not introduce an additional target-scale hyperparameter. Fixing $\alpha$ to a constant degrades performance relative to adaptive scaling, suggesting that treating the target scale as a fixed knob is not beneficial in this setting. Results are reported in Table 18.

**Expert-Only Regularization.** By design, SPHERE regularizes only the expert representations in the actor and leaves the gate unregularized, to isolate effects on policy plasticity without directly shaping routing. Concretely, it applies the penalty to the routing-weighted expert feature Gram constructed from the weighted concatenation in Eq. (18). We ablate an unweighted variant that drops the routing weights and directly concatenates expert features in Table 18.

**Gate-Only Regularization.** To isolate whether the gains come from regularizing the expert representations versus directly regularizing routing, we remove the expert penalty and apply the regularizer only to the gate outputs. This improves over the baseline ($0.43 \pm 0.08$ vs. $0.36 \pm 0.08$) but remains below SPHERE ($0.54 \pm 0.12$), suggesting that expert-feature mode separation is the main driver of the gains.

**Block-Diagonal Approximation.** Our analysis uses a block-diagonal approximation across layers to obtain a tractable layerwise surrogate objective. As a stress test, we remove this layerwise separation by concatenating the weighted expert features from all expert layers into a single vector and regularizing the resulting Gram. This variant attains $0.47 \pm 0.12$, improving over the baseline but providing no additional gain over SPHERE, supporting the adequacy of the block-diagonal layerwise proxy.

**Layer-Mixture Entropy Regularization.** Proposition 4.1 expresses $r_e(\mathbf{K})$ as a mixture over per-layer terms with weights $\{\alpha_\ell^{\text{exp}}\}$. We ablate adding an auxiliary entropy regularizer $-\sum_{\ell:\, s_\ell^{\text{exp}}>0} \alpha_\ell^{\text{exp}} \log \alpha_\ell^{\text{exp}}$ on top of SPHERE, which yields $0.41 \pm 0.03$. This underperforms SPHERE, suggesting that explicitly shaping the layer-mixture weights is not beneficial in this setting. One possible explanation is that this term imposes a global cross-layer constraint through $\{\alpha_\ell^{\text{exp}}\}$ and can overconstrain optimization, reducing the flexibility of different layers to adapt.

**Regularizing Both Feature and Gradient Factors.** Our method regularizes the weighted expert feature Gram $\mathbf{A}_{\text{last}}^{\text{exp}}$ (Section 4.4), treating the backprop-gradient Gram $\mathbf{G}_{\text{last}}^{\text{exp}}$ as fixed in the feature–gradient effective-rank factorization (Section 4). As an additional ablation, we apply the same SPHERE penalty to both $\mathbf{A}_{\text{last}}^{\text{exp}}$ and $\mathbf{G}_{\text{last}}^{\text{exp}}$, yielding $0.77 \pm 0.23$. This suggests that regularizing gradient factors can further help, although the higher variance indicates less stable optimization. However, $\mathbf{G}_{\text{last}}^{\text{exp}}$ is not directly available from forward passes and requires additional backpropagation to extract, increasing implementation complexity and computational overhead. We therefore do not adopt joint $\mathbf{A}_{\text{last}}^{\text{exp}}$–$\mathbf{G}_{\text{last}}^{\text{exp}}$ regularization as the main SPHERE instantiation and leave a more systematic study of jointly regularizing $\mathbf{A}_{\text{last}}^{\text{exp}}$ and $\mathbf{G}_{\text{last}}^{\text{exp}}$ to future work.

**Regularizing the Kronecker Proxy $\mathbf{A}_{\text{last}}^{\text{exp}} \otimes \mathbf{G}_{\text{last}}^{\text{exp}}$.** Section 4 introduces a Kronecker-product proxy $\tilde{\mathbf{G}}_\ell^{\text{GN,exp}} = \mathbf{A}_{\ell-1}^{\text{exp}} \otimes \mathbf{G}_\ell^{\text{exp}}$ for the expert-layer Gauss–Newton blocks. As an additional ablation, we directly apply the SPHERE penalty to the proxy matrix $\mathbf{A}_{\text{last}}^{\text{exp}} \otimes \mathbf{G}_{\text{last}}^{\text{exp}}$, which yields $0.66 \pm 0.20$ average success. This improves substantially over the baseline and even exceeds SPHERE in mean, suggesting that explicitly shaping a feature–gradient proxy of the expert Gauss–Newton geometry can be beneficial. However, the high variance indicates less stable optimization, consistent with coupling two stochastic factors and requiring extraction of $\mathbf{G}_{\text{last}}^{\text{exp}}$ via additional backpropagation.

**Regularizing the Khatri–Rao Expert Block $\mathbf{A}_{\text{last}}^{\text{exp}} * \mathbf{G}_{\text{last}}^{\text{exp}}$.** Section 4 models the same-layer expert Gauss–Newton blocks in block Khatri–Rao form $\mathbf{G}_\ell^{\text{GN,exp}} = \mathbf{A}_{\ell-1}^{\text{exp}} * \mathbf{G}_\ell^{\text{exp}}$. As an additional ablation, we directly apply the SPHERE penalty to $\mathbf{A}_{\text{last}}^{\text{exp}} * \mathbf{G}_{\text{last}}^{\text{exp}}$, which yields $0.50 \pm 0.25$ average success. This variant underperforms SPHERE and is similarly high-variance, suggesting that directly regularizing the full expert-block geometry is either too strict for this setting or introduces additional optimization noise relative to regularizing the feature Gram alone.

*Table 18.* **SPHERE design ablations reveal key components and trade-offs.** Fixing the target scale $\alpha$ degrades performance, while several alternative penalties underperform the default setting or trade higher mean success for substantially higher variance.

| Variant | Average success |
|---|---|
| w/o SPHERE | $0.36 \pm 0.08$ |
| w/ SPHERE | $0.54 \pm 0.12$ |
| Fixed $\alpha{=}1$ | $0.27 \pm 0.09$ |
| Fixed $\alpha{=}2$ | $0.40 \pm 0.18$ |
| Fixed $\alpha{=}5$ | $0.41 \pm 0.09$ |
| Unweighted feature Gram | $0.50 \pm 0.03$ |
| Joint feature Gram across layers | $0.47 \pm 0.12$ |
| Gate-only regularization | $0.43 \pm 0.08$ |
| Layer-mixture entropy term | $0.41 \pm 0.03$ |
| On $\mathbf{A}_{\text{last}}^{\text{exp}}$ and $\mathbf{G}_{\text{last}}^{\text{exp}}$ | $\mathbf{0.77 \pm 0.23}$ |
| On $\mathbf{A}_{\text{last}}^{\text{exp}} \otimes \mathbf{G}_{\text{last}}^{\text{exp}}$ | $0.66 \pm 0.20$ |
| On $\mathbf{A}_{\text{last}}^{\text{exp}} * \mathbf{G}_{\text{last}}^{\text{exp}}$ | $0.50 \pm 0.25$ |

## J.10. Additional Robustness Analysis Experiments

### J.10.1. ACTIVATION-FUNCTION ABLATION

**Activation Agnosticism.** We ablate the policy activation function on HumanoidBench under CRL. Table 19 compares Top-$K$ MoE and SPHERE under each activation, keeping all other settings fixed; the Tanh setting differs from the ReLU baseline only in the activation. SPHERE improves over the baseline under both ReLU and Tanh, suggesting that the gains do not rely on a particular activation choice.

*Table 19.* **SPHERE is robust to activation choice on HumanoidBench under CRL.** SPHERE improves five-task average success over Top-$K$ MoE under both ReLU and Tanh.

| Activation | Top-$K$ MoE | SPHERE |
|---|---|---|
| ReLU | $0.36 \pm 0.08$ | $\mathbf{0.54 \pm 0.12}$ |
| Tanh | $0.36 \pm 0.10$ | $\mathbf{0.58 \pm 0.17}$ |

### J.10.2. STRUCTURE AGNOSTICISM

**Structure Agnosticism.** To test whether SPHERE depends on a particular MoE structure, we apply the same penalty to Dense-MoE and DS-MoE actors under CRL. Results are summarized in Table 20. SPHERE attains $0.55 \pm 0.12$ on Dense-MoE and $0.51 \pm 0.08$ on DS-MoE, suggesting that the method transfers across MoE variants beyond Top-$K$. In the same HumanoidBench CRL setting, SPHERE also improves SoftMoE (Sokar et al., 2025) from $0.55 \pm 0.09$ to $0.60 \pm 0.05$.

*Table 20.* **SPHERE transfers across MoE architectures on HumanoidBench under CRL.** SPHERE improves both Dense-MoE and DS-MoE, indicating benefits beyond Top-$K$ MoE.

| MoE variant | w/o SPHERE | w/ SPHERE |
|---|---|---|
| Dense-MoE | $0.37 \pm 0.01$ | $\mathbf{0.55 \pm 0.12}$ |
| DS-MoE | $0.39 \pm 0.12$ | $\mathbf{0.51 \pm 0.08}$ |
| SoftMoE | $0.55 \pm 0.09$ | $\mathbf{0.60 \pm 0.05}$ |

### J.10.3. PENALTY RATIO ROBUSTNESS

**Penalty-ratio robustness.** SPHERE sets the adaptive regularization coefficient as $\lambda^{\text{e}} = \rho \cdot \text{stopgrad}\left( \frac{\|\nabla_\theta \mathcal{L}_{\text{actor}}\|_2}{\|\nabla_\theta \mathcal{L}_{\text{SPHERE}}\|_2 + \varepsilon} \right)$. Figure 10 reports a sweep over $\rho$ on HumanoidBench under CRL. Performance is best at $\rho = 1\text{e}{-}3$, while $\rho = 0$ removes the regularizer and recovers the baseline. Importantly, all nonzero $\rho$ values in this sweep outperform $\rho = 0$ by a clear margin ($+0.09$ to $+0.17$ average success), indicating that SPHERE is robust to the choice of $\rho$ and provides consistent gains once the regularizer is enabled. Overall, performance exhibits a broad optimum around $10^{-3}$. When $\rho$ is increased (e.g., $5\text{e}{-}2$), performance degrades relative to the best setting, consistent with over-regularization that can overconstrain representation

learning. We additionally find that this robustness carries over across benchmarks: on MetaWorld under CRL, using the same $\rho = 1e-3$ as HumanoidBench (two orders of magnitude smaller than our default MetaWorld setting in Table 2) yields $0.369 \pm 0.130$ average success, still improving over the unregularized Top-$K$ MoE baseline ($0.21 \pm 0.14$; Table 14).

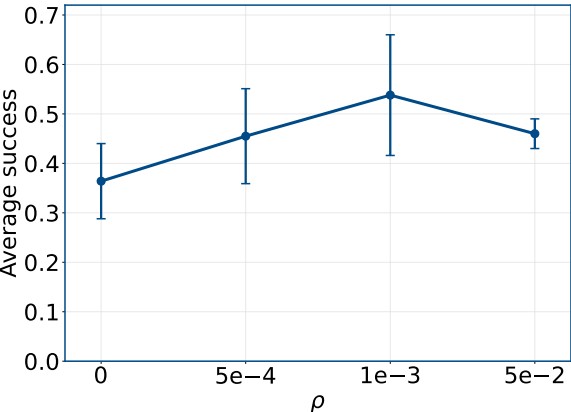

*Figure 10.* **SPHERE is robust to the ratio hyperparameter $\rho$ on HumanoidBench under CRL.** Any $\rho > 0$ improves five-task average success over $\rho = 0$ by $+0.09$ to $+0.17$, with a broad optimum around $\rho = 10^{-3}$.

### J.10.4. CAUSAL ANALYSIS

**Negative (sign-flip) intervention.** To probe whether SPHERE's effect on performance is mediated by spectral plasticity, we run a counterfactual intervention that flips the sign of the SPHERE ratio, $\rho = -1e-3$, while keeping the rest of the PPO and architecture settings fixed. This reverses the direction of the Gram-isotropy pressure in Eq. (26), encouraging *anisotropy* rather than isotropy. Table 21 summarizes average final success for $\rho = 0$ and $\rho = 1e-3$ from Figure 10, and reports the final success of the negative-intervention run. Compared to the SPHERE setting in Figure 3, the negative intervention exhibits substantially lower $r_e(\mathbf{K})$ throughout training, along with a sharp drop in success, as shown in Figure 11.

*Table 21.* **Sign-flipping the SPHERE ratio substantially reduces success on HumanoidBench under CRL.** Average final success for $\rho \in \{0, 10^{-3}, -10^{-3}\}$.

| Setting | Average final success |
|---|---|
| Top-$K$ MoE, $\rho = 0$ | $0.36 \pm 0.08$ |
| SPHERE, $\rho = 1e-3$ | $\mathbf{0.54 \pm 0.12}$ |
| Negative-SPHERE, $\rho = -1e-3$ | $0.20 \pm 0.09$ |

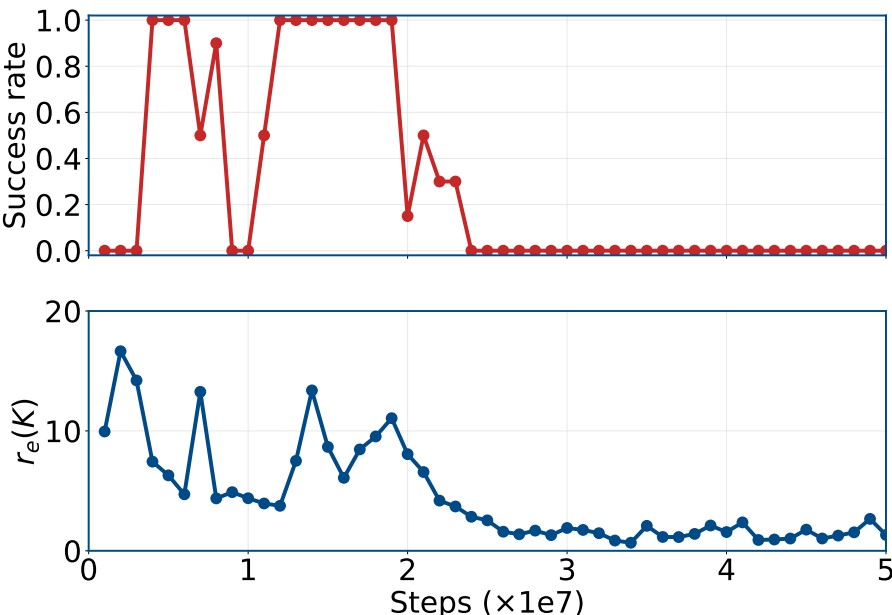

*Figure 11.* **Under the sign-flip intervention, both $r_e(\mathbf{K})$ and success collapse.** Evaluation success rate and $r_e(\mathbf{K})$ over training steps for $\rho = -10^{-3}$.

**Lead–lag evidence (negative intervention).** We quantify whether changes in $r_e(\mathbf{K})$ precede changes in success by computing the lagged correlation $\mathrm{corr}(\mathrm{success}_t, r_e(\mathbf{K})_{t-\ell})$ over evaluation checkpoints, with $\ell$ measured in evaluation ticks. As shown in Figures 11 and 12, $r_e(\mathbf{K})$ tends to lead success: lagged correlations are larger for $\ell > 0$ and peak at $\ell = 10$ (Pearson 0.52). Consistently, in the one-step predictive model $\mathrm{success}_t \sim \mathrm{success}_{t-1} + \mathrm{trend}(t) + r_e(\mathbf{K})_{t-1}$, the coefficient on $r_e(\mathbf{K})_{t-1}$ is positive (standardized 0.12) and remains positive under a moving-block bootstrap (block size 25; $p(\beta \leq 0) \approx 0.007$). These results are consistent with $r_e(\mathbf{K})$ being a leading indicator of success in this setting. The sign-flip intervention identifies the causal effect of $\rho$ on both $r_e(\mathbf{K})$ and success; the lead–lag analysis is used only to support the mechanism that changes in $r_e(\mathbf{K})$ precede changes in success, rather than to identify a causal effect of $r_e(\mathbf{K})$ on success.

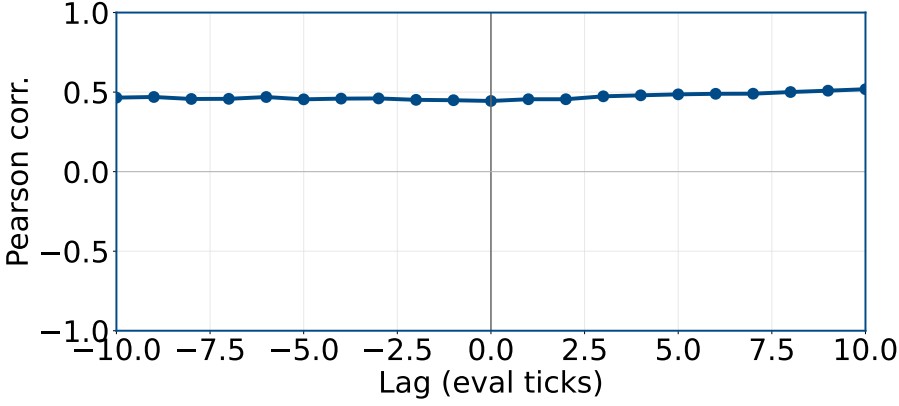

*Figure 12.* $r_e(\mathbf{K})$ **tends to lead success under the negative intervention.** We report the correlation between $\mathrm{success}_t$ and $r_e(\mathbf{K})_{t-\ell}$ across evaluation checkpoints. Positive lag $\ell > 0$ means $r_e(\mathbf{K})$ is shifted earlier.

### J.10.5. ALGORITHM AGNOSTICISM

**Algorithm Agnosticism.** To test whether SPHERE is algorithm-agnostic, we apply the same penalty to a Top-$K$ MoE critic trained with TD3 under CRL. Table 22 reports final success averaged over seeds. SPHERE improves average success from $0.50 \pm 0.08$ to $0.65 \pm 0.09$, suggesting that the penalty can also benefit off-policy training. This further suggests that

SPHERE is not limited to MoE actors and can also benefit MoE critics.

*Table 22.* **SPHERE improves TD3 with a Top-$K$ MoE critic by 30% on MetaWorld under CRL.** This demonstrates that SPHERE benefits off-policy training and generalizes beyond PPO actors to MoE critics.

| Method | Average success |
|---|---|
| Top-$K$ MoE critic (TD3) | $0.50 \pm 0.08$ |
| Regularized MoE critic (TD3) | $\mathbf{0.65 \pm 0.09}$ |

## J.11. SPHERE Promotes Expert Specialization

We study whether SPHERE increases expert specialization on HumanoidBench by measuring (i) representation-level diversity among the selected experts and (ii) geometric coherence of the gate's routing in its embedding space.

**Representation-level specialization.** Let $h_{i,e} \in \mathbb{R}^H$ denote the last-hidden feature of expert $e$ at state $s_i$, and let $S_i$ denote the set of selected experts at $s_i$. We measure redundancy among selected experts via the mean cosine overlap over pairs $(e, e') \in S_i$ with $e \neq e'$. We also report a complementary orthogonality score defined as the mean of $1 - |\cos|$ over the same pairs.

**Gate geometry and local consistency.** Let $g_i$ be the gate embedding at $s_i$ and let $y_i$ be the Top-1 expert label. We compute (i) the silhouette score of $\{g_i\}$ clustered by $\{y_i\}$, and (ii) a $k_{\mathrm{nn}}$-nearest-neighbor consistency score that averages the Jaccard overlap between the selected-expert sets of neighboring states in the gate-embedding space.

**Expert load balancing.** Let $c_e$ denote the number of times expert $e$ is selected across the batch, i.e., $c_e := |\{(i, e') : e' \in S_i, \ e'=e\}|$, so that $\sum_e c_e = \sum_i |S_i|$. We quantify load imbalance by the maximum relative deviation from the uniform expected count,

$$\mathrm{MaxVio} := \max_e \frac{|c_e - \bar{c}|}{\bar{c}}, \qquad \bar{c} := \frac{1}{E} \sum_{e=1}^{E} c_e,$$

We also report the active-expert ratio $|\{e : c_e > 0\}|/E$ as a simple coverage measure, and the normalized routing entropy

$$H_{\mathrm{route}} := \frac{-\sum_{e=1}^{E} \hat{p}_e \log \hat{p}_e}{\log E}, \qquad \hat{p}_e := \frac{c_e}{\sum_{e'=1}^{E} c_{e'}}.$$

**Results.** Table 23 reports these metrics on the HumanoidBench run task using $N$=512 deterministic on-policy rollout states across independent runs. SPHERE reduces overlap and increases orthogonality, indicating less redundant expert representations for the same state. In parallel, SPHERE improves gate-embedding coherence and local routing consistency. Overall, these results indicate that SPHERE improves representation-level specialization and routing geometry in MoE actors.

*Table 23.* **SPHERE promotes expert specialization and routing coherence.** On the HumanoidBench run task, SPHERE reduces expert overlap from 0.34 to 0.13 and increases orthogonality from 0.66 to 0.87, indicating less redundant representations among selected experts. Gate-geometry metrics such as silhouette and kNN Jaccard also improve, reflecting more coherent routing. Arrows in column headers indicate the direction of improvement.

| Method | Overlap ↓ | Orthogonality ↑ | Silhouette ↑ | kNN Jaccard ↑ | MaxVio ↓ | Active ratio ↑ | Routing entropy ↑ |
|---|---|---|---|---|---|---|---|
| Top-$K$ MoE (baseline) | $0.34 \pm 0.01$ | $0.66 \pm 0.01$ | $0.02 \pm 0.15$ | $0.82 \pm 0.11$ | $2.73 \pm 0.85$ | $0.67 \pm 0.21$ | $0.66 \pm 0.11$ |
| SPHERE (ours) | $\mathbf{0.13 \pm 0.01}$ | $\mathbf{0.87 \pm 0.01}$ | $\mathbf{0.15 \pm 0.06}$ | $\mathbf{0.96 \pm 0.02}$ | $\mathbf{2.70 \pm 0.75}$ | $\mathbf{0.69 \pm 0.10}$ | $0.66 \pm 0.08$ |

## J.12. Runtime Overhead Scaling to $E$=1000 Experts

Table 24 reports wall-clock training time per task under the same training setup while scaling the expert count up to $E$=1000. Although total training time increases with $E$, the relative overhead of SPHERE remains small (only 3.4–6.3% across $E \in \{10, 20, 40, 80, 160, 1000\}$), indicating that SPHERE is not a computational bottleneck.[2]

---

[2]To rule out confounding from gradient-norm scaling, for the large-$E$ runs ($E \in \{80, 160, 1000\}$) we directly optimize $\mathcal{L}_{\mathrm{PPO}} + \mathcal{L}_{\mathrm{SPHERE}}$ (i.e., without adaptive scaling).

**Efficient Gram computation.** Recall that the SPHERE penalty depends only on $\mathrm{Tr}(\mathbf{A}_{\text{last}}^{\text{exp}})$ and $\|\mathbf{A}_{\text{last}}^{\text{exp}}\|_F^2$. We therefore evaluate it using whichever is cheaper to form: $\mathbf{A}_{\text{last}}^{\text{exp}}$ or the sample Gram $(1/N)\Phi\Phi^\top$. By cyclicity of trace and the fact that $\Phi^\top\Phi$ and $\Phi\Phi^\top$ share the same non-zero eigenvalues, we have $\mathrm{Tr}(\mathbf{A}_{\text{last}}^{\text{exp}}) = \mathrm{Tr}((1/N)\Phi\Phi^\top)$ and $\|\mathbf{A}_{\text{last}}^{\text{exp}}\|_F^2 = \|(1/N)\Phi\Phi^\top\|_F^2$. In our experiments, $m$ scales linearly with the expert count for fixed per-expert width, so this Gram computation scales linearly in $E$. Let $\Phi \in \mathbb{R}^{N \times m}$ denote the gating-weighted expert feature matrix at the regularized layer (Eq. (18)), where $N$ is the PPO minibatch size and $m$ is the concatenated feature dimension. Forming the smaller Gram and evaluating the statistics costs

$$\text{Forward cost} \;=\; \Theta\big(\min\{N,m\}^2 \max\{N,m\}\big),$$

dominated by a single matrix multiplication computing either $\Phi^\top\Phi$ or $\Phi\Phi^\top$. Moreover, when $m \geq N$ and we work with the sample Gram $G := \Phi\Phi^\top/N$, the SPHERE penalty takes the form

$$\mathcal{L}_{\text{SPHERE}}(\Phi) \;=\; \|G\|_F^2 \;-\; \frac{\mathrm{Tr}(G)^2}{m}, \qquad G = \frac{1}{N}\Phi\Phi^\top,$$

and a direct differentiation yields

$$\nabla_\Phi \mathcal{L}_{\text{SPHERE}}(\Phi) \;=\; \frac{4}{N}\Big(G - \frac{\mathrm{Tr}(G)}{m}\mathbf{I}_N\Big)\Phi.$$

Thus the backward pass is dominated by the matrix multiplication $G\Phi$ and has the same asymptotic cost as forming $G$. In our regime $m \gg N$, this gives an additional $\Theta(N^2 m)$ flops per PPO minibatch for SPHERE (forward and backward each contributing $\Theta(N^2 m)$).

**Potential block-sparse acceleration.** Since the penalty is defined on the *gating-weighted* expert features, Top-$K$ routing induces a block-sparse structure in $\Phi$ (and hence in the expert-layer Gram), which could potentially be exploited in future work to accelerate the computation with custom block-sparse kernels. In the experiments reported in Table 24, we do not implement such kernels and use the default dense computation described above.

*Table 24.* **SPHERE adds 3.4–6.3% wall-clock overhead when scaling to $E{=}1000$ experts.** Hours per task on NVIDIA A100 for the HumanoidBench five-task suite with 10M environment steps per task.

| Method | $E{=}10$ | $E{=}20$ | $E{=}40$ | $E{=}80$ | $E{=}160$ | $E{=}1000$ |
|---|---|---|---|---|---|---|
| Top-$K$ MoE (baseline) | 2.23 | 2.23 | 2.24 | 2.05 | 2.33 | 4.37 |
| SPHERE (ours) | 2.37 (+6.28%) | 2.34 (+4.93%) | 2.37 (+5.80%) | 2.11 (+3.35%) | 2.45 (+5.27%) | 4.61 (+5.39%) |

## J.13. Validation of Theoretical Approximation Assumptions

Our analysis adopts a block-diagonal approximation of the Gauss–Newton matrix that separates gate and expert parameters (Eq. (15)). For dense MLPs, block-diagonal and within-layer independence approximations are widely adopted in practical curvature modeling and natural-gradient methods (e.g., KFAC and its variants; Martens & Grosse, 2015; Grosse & Martens, 2016; Botev et al., 2017). In MoE actors, however, the validity of an analogous *gate–expert* block structure is *not* obvious: Top-$K$ routing multiplies gate probabilities with expert activations, which can in principle induce substantial cross-block curvature between gate and expert parameters. Complementing the empirical diagnostics below, Appendix I provides a general perturbation bound that translates small gate–expert coupling into a controlled change in $r_e(\mathbf{K})$.

**Validation of the gate–expert block-diagonal assumption.** To empirically assess this MoE-specific assumption, we measure the relative magnitude of the gate–expert cross-block in the Gauss–Newton matrix. Let $G_{ab}^{\text{GN}}$ denote the Gauss–Newton sub-block between parameter groups $a$ and $b$, where $a, b \in \{\text{gate}, \text{expert}\}$. We report the normalized block cosine similarity, defined as

$$\cos(a,b) \;:=\; \frac{\|G_{ab}^{\text{GN}}\|_F}{\sqrt{\|G_{aa}^{\text{GN}}\|_F \|G_{bb}^{\text{GN}}\|_F}} \in [0,1],$$

which is scale-invariant and directly compares cross-block curvature to the geometric mean of the corresponding within-block curvature. If gate and expert parameters were strongly coupled in $G^{\text{GN}}$, we would expect a large off-diagonal value comparable to the diagonal blocks; conversely, values near zero indicate a near block-diagonal structure. Figure 13 shows that the off-diagonal gate–expert coupling is substantially smaller than the within-block terms in a representative late-stage checkpoint on the HumanoidBench run task, supporting the gate–expert block-diagonal surrogate used throughout our analysis.

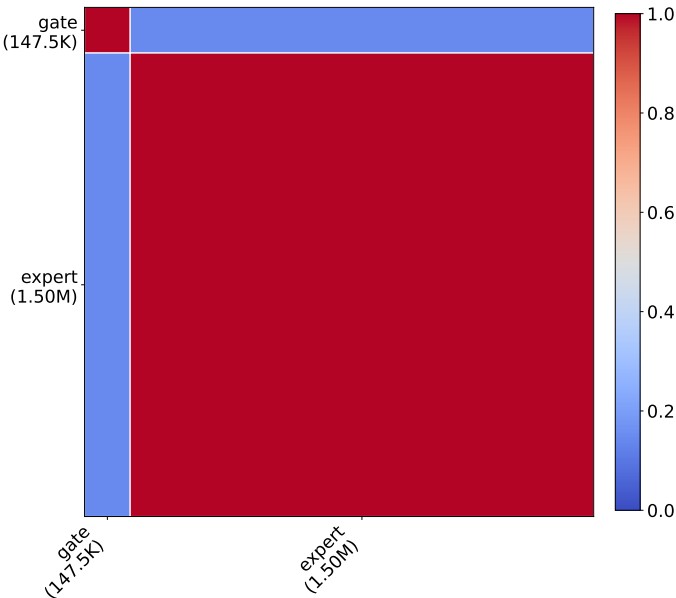

*Figure 13.* **Gate–expert coupling is weak in a Top-$K$ MoE actor.** We visualize the Gauss–Newton block cosine similarity $\cos(a, b)$ between the gate and expert parameter groups on the HumanoidBench run task. Blocks are ordered as $\{\text{gate}, \text{expert}\}$ to match Eq. (15), and block widths and heights are proportional to parameter counts. Small off-diagonal values indicate weak gate–expert coupling relative to within-block curvature.

**Validation of Proposition 4.2.** Proposition 4.2 replaces the intractable expert-layer block $\mathbf{G}_\ell^{\text{GN,exp}}=\mathbf{A}_\ell^{\text{exp}} * \mathbf{G}_\ell^{\text{exp}}$ (block Khatri–Rao form) with its Kronecker proxy $\tilde{\mathbf{G}}_\ell^{\text{GN,exp}}=\mathbf{A}_\ell^{\text{exp}} \otimes \mathbf{G}_\ell^{\text{exp}}$, and predicts that optimizing the proxy effective rank is sufficient for improving the expert-layer effective rank under the model. We empirically verify this prediction by checking that the proxy effective rank tracks the target effective rank across rollout batches.

Concretely, we roll out a fixed checkpoint on the HumanoidBench run task, repeatedly sample batches of $N{=}64$ states, and instantiate Proposition 4.2 at the actor's output block. For each batch, we compute the proxy effective rank $r_e(\tilde{\mathbf{G}}_{\text{out}}^{\text{GN,exp}})$ and the target effective rank $r_e(\mathbf{G}_{\text{out}}^{\text{GN,exp}})$, where $\tilde{\mathbf{G}}_{\text{out}}^{\text{GN,exp}} = \mathbf{A}_{\text{last}}^{\text{exp}} \otimes \mathbf{G}_{\text{out}}^{\text{exp}}$ and $\mathbf{G}_{\text{out}}^{\text{GN,exp}} = \mathbf{A}_{\text{last}}^{\text{exp}} * \mathbf{G}_{\text{out}}^{\text{exp}}$ under the model. Figure 14 shows a strong positive association, supporting the Kronecker proxy as a faithful surrogate for analyzing and optimizing expert-layer spectral properties.

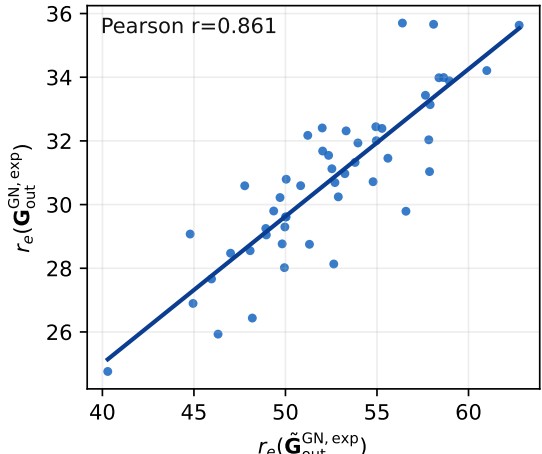

*Figure 14.* **The Kronecker proxy provides a reliable surrogate for expert-block effective rank.** The proxy effective rank $r_e(\tilde{\mathbf{G}}_{\text{out}}^{\text{GN,exp}})$ is strongly correlated with the target effective rank $r_e(\mathbf{G}_{\text{out}}^{\text{GN,exp}})$, with Pearson $r{=}0.861$. This supports Proposition 4.2 at the expert output block in a Top-$K$ MoE actor.

**Validation of the K-FAC factorization assumption.** Beyond the Kronecker proxy at the level of effective ranks, K-FAC additionally assumes that, within an expert layer, the activation and backprop factors are weakly coupled across tokens (tokens are (state, expert) pairs), so that the empirical GN block $\mathbf{G}^{\hat{\mathrm{GN}},\exp}_{\mathrm{out}} = \frac{1}{T}\sum_t (g_t g_t^\top) \otimes (a_t a_t^\top)$ is well-approximated by $\mathbf{G}^{\hat{\exp}}_{\mathrm{out}} \otimes \mathbf{A}^{\hat{\exp}}_{\mathrm{last}}$. To test this without forming either matrix, we sample many random rank-one parameter perturbations $\Delta W$ over rollout batches and compare the resulting normalized directional curvatures. Figure 15 compares the K-FAC proxy curvature $q_{\mathrm{KFAC}}$ against the empirical GN curvature $q_{\mathrm{GN}}$; if the factorization holds, the points concentrate along the diagonal.

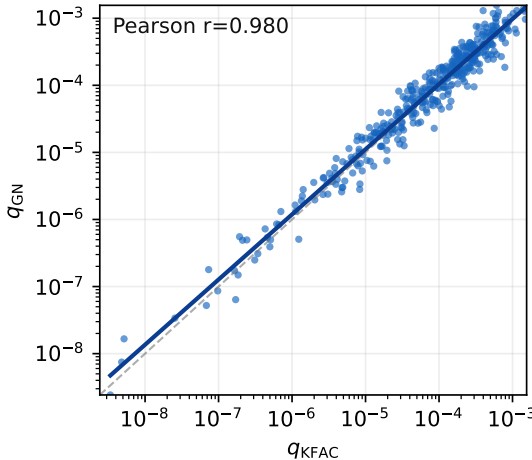

*Figure 15.* **K-FAC proxy curvatures closely match empirical Gauss–Newton curvatures at the expert output block.** The K-FAC proxy curvature $q_{\mathrm{KFAC}}$ concentrates near the diagonal when compared to the empirical Gauss–Newton curvature $q_{\mathrm{GN}}$. Pearson correlation between $\log q_{\mathrm{KFAC}}$ and $\log q_{\mathrm{GN}}$ is $r{=}0.980$.

### J.14. Applicability Beyond RL and MLP Settings

#### J.14.1. VISUAL METAWORLD WITH A RESNET BACKBONE

We further evaluate SPHERE in a visual MetaWorld CRL setting with a ResNet encoder, as shown in Table 25. SPHERE improves average success from $0.18 {\pm} 0.04$ to $0.32 {\pm} 0.07$, suggesting that the gain extends beyond an MLP policy backbone.

*Table 25.* **SPHERE improves visual MetaWorld CRL with a ResNet backbone.** We report average success across the continual task sequence.

| Method | Average success |
|---|---|
| ResNet policy | $0.18 \pm 0.04$ |
| ResNet policy + SPHERE | $\mathbf{0.32 \pm 0.07}$ |

#### J.14.2. COMPUTER VISION WITH MLP AND VIT BACKBONES

To probe whether SPHERE provides similar spectral regularization benefits outside reinforcement learning, we evaluate SPHERE in a supervised continual-learning setting on SplitCIFAR-100.

**Setup.** We consider SplitCIFAR-100 with 20 tasks (5 classes per task) and sequentially fine-tune a Top-$K$ MoE classifier with $E{=}10$ experts and $K{=}2$. We train under task-incremental supervision via task-masked cross entropy (restricting the softmax to the 5 classes of the current task), AdamW learning rate $5 \times 10^{-4}$, 5 epochs per task, and CIFAR augmentation. SPHERE uses the feature-Gram penalty in Eq. (26) with gradient-norm scaling (Appendix A.2) and ratio $\rho = 0.1$.

**Metrics.** Following our per-task evaluation protocol for continual training, after finishing task $t$ we evaluate that checkpoint on the *current* task-$t$ test set and then average across tasks. We report class-incremental accuracy (argmax over all 100 classes), i.e., without task masking at evaluation. As an auxiliary spectral diagnostic, we compute $r_e(\mathbf{K})$ of the classifier empirical NTK on a fixed held-out batch of CIFAR test images reused across checkpoints and methods.

**Results.** Table 26 reports the results. SPHERE improves class-incremental in-task success, indicating improved ability to fit new tasks under the sequential protocol.

*Table 26.* **SPHERE improves SplitCIFAR-100 continual-learning performance and increases final-task $r_e(\mathbf{K})$.** We report average class-incremental accuracy across tasks and the final-task empirical NTK effective rank $r_e(\mathbf{K})$.

| Method | Average success rate | Final-task $r_e(\mathbf{K})$ |
|---|---|---|
| Fine-tune | $0.29 \pm 0.01$ | 6.55 |
| SPHERE | $\mathbf{0.48 \pm 0.01}$ | **10.03** |
| Gain | $+69\% \pm 6\%$ | $+53\%$ |

**ViT backbone.** We additionally evaluate a small vision transformer where we replace the feed-forward network (FFN) sublayer in each block with a Top-$K$ MoE (8 layers, 8 heads, patch size 4, embed dim 256, $E{=}128$, $K{=}2$). For this variant, we use the same SPHERE penalty with ratio $\rho = 0.1$. Table 27 shows that SPHERE continues to improve class-incremental in-task success under this architecture.

*Table 27.* **SPHERE continues to improve SplitCIFAR-100 with a transformer MoE-FFN backbone and increases final-task $r_e(\mathbf{K})$.** We report average class-incremental accuracy across tasks and the final-task empirical NTK effective rank $r_e(\mathbf{K})$.

| Method | Average success rate | Final-task $r_e(\mathbf{K})$ |
|---|---|---|
| Fine-tune | $0.38 \pm 0.03$ | 4.54 |
| SPHERE | $\mathbf{0.52 \pm 0.03}$ | **7.46** |
| Gain | $+37\% \pm 13\%$ | $+64\%$ |

### J.14.3. NATURAL LANGUAGE PROCESSING WITH A TRANSFORMER BACKBONE

To probe whether the supervised gains observed above extend to text, SPHERE is evaluated on a continual 20 Newsgroups classification stream.

**Setup.** A long-horizon stream of $T{=}400$ tasks is constructed, where each task is a 10-way classification problem. For each task, ten classes are sampled without replacement from the 20 labels, and classes may repeat across tasks. Per task, 5 training examples per class and 50 test examples per class are sampled from the standard train/test split; when a class re-appears later in the stream, the per-class training subset is resampled to avoid repeatedly training on an identical few-shot set. The model is a 4-layer Transformer classifier with 8 attention heads and embedding dimension 128, and uses a Top-$K$ MoE FFN in every layer with $E{=}8$, $K{=}1$, temperature 0.01, and FFN hidden dimension 1024. Training uses task-masked cross entropy, AdamW learning rate $10^{-3}$, weight decay 0.3, batch size 64, and 1 epoch per task. SPHERE uses the feature-Gram penalty in Eq. (26) with gradient-norm scaling as described in Appendix A.2, and uses ratio $\rho{=}0.1$ applied to all MoE layers.

**Metric.** Following our per-task evaluation protocol, after finishing task $t$ we evaluate that checkpoint on the *current* task-$t$ test set and then average across tasks. We report task-masked in-task success, i.e., argmax restricted to the 10 classes of task $t$. The corresponding chance level is 0.1. As an auxiliary spectral diagnostic, we compute the empirical NTK effective rank $r_e(\mathbf{K})$ on a fixed held-out batch of $N{=}32$ test examples reused across methods.

**Results.** Table 28 reports the results. SPHERE substantially improves the raw average per-task success in this long-horizon text stream.

*Table 28.* **SPHERE substantially improves long-horizon continual learning on 20 Newsgroups and increases final-task $r_e(\mathbf{K})$.** We report average task-masked accuracy across tasks and the final-task empirical NTK effective rank $r_e(\mathbf{K})$.

| Method | Average success rate | Final-task $r_e(\mathbf{K})$ |
|---|---|---|
| Fine-tune | $0.17 \pm 0.02$ | 2.71 |
| SPHERE | $\mathbf{0.27 \pm 0.01}$ | **14.55** |
| Gain | $+62\% \pm 12\%$ | $+438\%$ |

