# OpenReview forum: "SPHERE: Mitigating the Loss of Spectral Plasticity in Mixture-of-Experts for Deep Reinforcement Learning"
_ICML.cc/2026/Conference — ICML 2026 regular_

### Official Review · Reviewer_8iKe · 2026-03-05

**Soundness:** 3
**Presentation:** 3
**Significance:** 3
**Originality:** 3
**Overall Recommendation:** 5
**Confidence:** 5

**Summary:**

This work presents SPHERE, an auxiliary loss tailored for MoE-based policies to retain spectral plasticity, defined as the spectral-entropy effective rank of the empirical neural tangent kernel, in Continual Reinforcement Learning. The work theoretically motivates SPHERE and derives its objective using a tractable parameter-space surrogate for spectral plasticity. The work empirically validates that SPHERE retains spectral plasticity compared to baselines, shows it leads to increased performance in two CRL benchmarks and qualitatively demonstrates the effects of the regularisation.

**Compliance With Llm Reviewing Policy:**

Affirmed.

**Final Justification:**

The authors propose a regularisation method to maintain spectral plasticity in neural networks during continual learning. The method is technically sound, shows improvement over baselines in two benchmarks and the presentation is overall good, demonstrating qualitatively that SPHERE achieves what it sets out to: maintaining eNTK rank. The authors additionally provide relatively high quality source code which aids reproducibility.

The work's primary weakness was its limited seed count in high variance benchmarks and insufficient statistical testing, not ablating thoroughly where and how SPHERE helps and how it interacts with other common regularisation methods in this setting, not comparing it to a wider range of stronger baselines and across more settings than continual reinforcement learning.

During the rebuttal, the authors thoroughly addressed my concerns across all the aforementioned weakness dimensions. Results with improved experimental methodology were provided, results against a strong baseline were also provided, as were ablations on how it interacts with layer normalisation and how applying it to different networks within the agent affects performance, and even results on learning from a stationary data distribution. The authors additionally addressed some of my concerns regarding the paper's presentation. This thorough rebuttal raised my score from 3 - Weak Reject to 5 - Accept. The work's remaining weakness is demonstrating only modest performance improvement despite very strong theoretical justification, which indicates that its primary aim of preserving eNTK rank might be relatively insufficient and not of critical import to resolving problems in continual learning.

**Key Questions For Authors:**

1. Can the authors provide confidence intervals for Figures 4 and 5 and report IQM/IQR? Proving that the improvements are statistically significant would address my primary concerns about experimental methodology.
2. Can the authors provide an ablation applying SPHERE to the critic network, given that architectural choices matter just as much if not more in value learning? Ideally this should be an ablation applying SPHERE to only the critic network, only the actor network, and on both networks.
3. Can the authors elucidate why SPHERE applied to all layers hurts performance, and whether applying it to only the last layer is optimal even in the supervised learning settings considered in the appendix?
4. How does SPHERE compare to other plasticity-preserving continual learning methods like ReDo, ReGrama or Soft Shrink-and-Perturb? If SPHERE is competitive or complementary, this would significantly change my assessment of its significance.

**Limitations:**

yes

**Strengths And Weaknesses:**

Strengths
- The SPHERE loss term is theoretically well justified as a way to mitigate spectral collapse in the eNTK of the network. The derivation of SPHERE is mathematically sound.
- Experimental results successfully validate theoretical insight and serve as a convincing demonstration of SPHERE maintaining eNTK rank, further confirmed by a qualitative t-SNE visualisation.
- The authors provide code in JAX that aids reproducibility and the implementation of SPHERE therein is remarkably simple.
- Very thorough appendix, containing extensive ablations. Tables 17-19 in particular confirm insights generalise beyond the RL settings in the main body of the paper

Weaknesses
- The experimental methodology is inadequate and as a result whether SPHERE and eNTK rank regularisation as a whole actually meaningfully improve downstream task performance remains unclear.
  - Figures 4 and 5 lack confidence intervals, making it difficult to assess how statistically significant the increase in performance actually is.
  - For benchmarks with particularly high variance like MetaWorld, using only 5 seeds for experiments is inadequate. However, high variance also appears to be an issue in HumanoidBench as Table 1 indicates (0.54 ± 0.12 indicates significant uncertainty). A reasonable way to address this methodological shortcoming would be to ideally use at least 10 seeds and at a minimum report inter-quartile mean (IQM) and visualise inter-quartile range (IQR) to avoid susceptibility to outliers, as proposed by [1].
- The experiments contain a limited selection of baselines, resulting in the significance of the method in terms of efficacy in the greater Continual RL landscape remaining unclear.
  - The authors primarily focus on how SPHERE improves over a Top-K MoE baseline, and while they do compare to some CRL baselines, additional baselines would help illustrate how it performs against a broader spectrum of CRL methods, especially because the paper makes CRL its setting of focus. For instance, Soft Shrink-and-Perturb [2], ReDo [3], and ReGraMa [4].
  - Ablations studying how SPHERE interacts with other regularisation methods such as layer norm and weight decay would also greatly help understand SPHERE's importance for good performance in this setting.
- The result that applying SPHERE to all layers reduces performance is perplexing and suggests that the relationship between eNTK spectral collapse, plasticity and downstream performance is more nuanced than originally assumed.
  - Intuitively, avoiding eNTK spectral collapse maintaining high rank should result in increased feature learning, not inhibit it. Of course overly strong regularisation can harm performance even if theoretically justified, but perhaps this motivates tuning the coefficient per-layer. Another possible scenario might be that attempting to regularise against eNTK spectral collapse in all layers using SPHERE results in optimisation instability, but that is not discussed.
  - Additionally, if the regularisation helps when only applied to the last layer, that might suggest that the last layer's eigenspectra dominate the eNTK's effective rank across the whole network in this particular setting.
  - Further experiments attempting to elucidate where to apply SPHERE, such as more detailed ablations between applying SPHERE to various layer combinations in different settings, are needed to make the method easier to use with less tuning by practitioners.
- The choice to apply SPHERE to only the actor network is questionable and weakens the results, since a variety of recent literature [5,6,7,8] has highlighted that architectural choices and scaling (which is the main motivation for the use of MoEs) matter most in value networks / critics rather than actor networks, even in multi-task learning.
- The presentation of the method is also unconvincing.
  - Figure 2 is largely a duplicate of Figure 5, which does not bode well when the main body is already clearly constrained for space. Perhaps it could be replaced with some of the stronger results found in the appendix.
  - Continual RL, and specifically continuous control continual RL, is an unnecessarily restrictive setting for demonstrating the efficacy of SPHERE. The theory behind the method applies to essentially all of deep learning, and there are other settings where mitigation in plasticity loss can be demonstrated. The authors in fact do include such evaluation in the Appendix which is in support of SPHERE, but perhaps it should be present in the main paper. Additionally, while the authors consider MoE architecture specifics in their derivation of the SPHERE objective, the final method being applied over the concatenated features across experts make it applicable to any dense layer as well. Once again the authors consider this in the Appendix but do not include the result in the main paper even though it would serve to further showcase SPHERE's wide applicability and increased performance across diverse settings.
- While the work explicitly considers non-stationary settings such as RL and CRL, this calls for comparison in stationary learning experiments, to study whether SPHERE maintains parity. Demonstrating that SPHERE does not degrade performance would indicate that the method is generally safe to apply more broadly.
- The non-stationarity in the CRL setting is relatively limited, as there are few (5-10) task boundaries for MetaWorld and HumanoidBench. While J.7.2 does show NLP experiments over a 400 task horizon, RL experiments over longer horizons would be welcome as that is the main setting under consideration in the main body of the paper currently.

Minor comments
- L31 c2: "perform continual learning in continuous learning settings" seems awkward, perhaps simply "perform in continuous learning settings"
- It is difficult to determine if the large relative increase in performance in the CRL setting vs the RL setting is due to better handling of the increased non-stationarity in CRL, or simply because it becomes exponentially harder to achieve success as the benchmark saturates.

References
- [1] Agarwal, Rishabh, et al. "Deep reinforcement learning at the edge of the statistical precipice." Advances in neural information processing systems 34 (2021): 29304-29320.
- [2] Dohare, Shibhansh, Qingfeng Lan, and A. Rupam Mahmood. "Overcoming policy collapse in deep reinforcement learning." Sixteenth European Workshop on Reinforcement Learning. 2023.
- [3] Sokar, Ghada, et al. "The dormant neuron phenomenon in deep reinforcement learning." International Conference on Machine Learning. PMLR, 2023.
- [4] Liu, Jiashun, et al. "Measure gradients, not activations! enhancing neuronal activity in deep reinforcement learning." arXiv preprint arXiv:2505.24061 (2025).
- [5] Rybkin, Oleh, et al. "Value-based deep rl scales predictably." arXiv preprint arXiv:2502.04327 (2025).
- [6] Nauman, Michal, et al. "Bigger, regularized, categorical: High-capacity value functions are efficient multi-task learners." arXiv preprint arXiv:2505.23150 (2025).
- [7] McLean, Reginald, et al. "Multi-task reinforcement learning enables parameter scaling." arXiv preprint arXiv:2503.05126 (2025).
- [8] Lee, Hojoon, et al. "Hyperspherical normalization for scalable deep reinforcement learning." arXiv preprint arXiv:2502.15280 (2025).

---

> ### Author Rebuttal · Authors · 2026-03-31
>
> We appreciate your careful and detailed review. We address each of your concerns below.
>
> ### Concern 1: Statistical reliability of the main gains
>
> We now provide 10-seed results on both benchmarks.
>
> | Benchmark | Method | Mean ± std | 95% CI | IQM | IQR |
> | --- | --- | --- | --- | --- | --- |
> | HumanoidBench CRL | Baseline | 0.36 ± 0.05 | [0.33, 0.39] | 0.36 | 0.05 |
> | HumanoidBench CRL | SPHERE | 0.53 ± 0.07 | [0.48, 0.58] | 0.52 | 0.11 |
> | MetaWorld CRL | Baseline | 0.16 ± 0.08 | [0.11, 0.21] | 0.14 | 0.14 |
> | MetaWorld CRL | SPHERE | 0.41 ± 0.07 | [0.36, 0.45] | 0.41 | 0.07 |
>
> Across both benchmarks, SPHERE **remains clearly above the baseline** under 10-seed evaluation.
>
> ### Concern 2: Breadth of baselines in the broader continual-RL landscape
>
> We now also have a ReDo result in the HumanoidBench CRL setting. ReDo reaches 0.41 ± 0.04 average success over the five tasks, **improving over the paper's Top-$K$ MoE baseline but remaining below SPHERE**.
>
> ### Concern 3: Interaction with other regularizers
>
> We now have layer-norm interaction results in HumanoidBench CRL with PPO and MetaWorld CRL with TD3.
>
> | Setting | SPHERE only | SPHERE with LN |
> | --- | --- | --- |
> | HumanoidBench CRL with PPO | 0.54 ± 0.12 | 0.54 ± 0.03 |
> | MetaWorld CRL with TD3 | 0.55 ± 0.06 | 0.65 ± 0.09 |
>
> In HumanoidBench CRL with PPO, adding LN to SPHERE leaves the mean essentially unchanged but reduces variance. In MetaWorld CRL with TD3, adding LN to SPHERE yields a further gain.
>
> ### Concern 4: Actor / critic / both placement
>
> For PPO in the HumanoidBench CRL setting, we now compare actor-only, critic-only, and both-network application.
>
> | PPO placement | Average success |
> | --- | --- |
> | actor only | 0.54 ± 0.12 |
> | critic only | 0.41 ± 0.06 |
> | both | 0.57 ± 0.07 |
>
> In PPO, actor-only already clearly outperforms critic-only, and applying SPHERE to both networks performs best overall. This is consistent with PPO being an **on-policy method** in which the actor directly drives the policy update, while the critic mainly serves a variance-reduction role. This is also consistent with [5–8], which are closer to value-centric or off-policy settings and therefore emphasize critic/value-side improvements. By contrast, in the **off-policy TD3 result already reported in Appendix Table 14, the gain appears on the critic side**.
>
> ### Concern 5: Layer scope
>
> Table 1 shows a layer-scope effect. Applying SPHERE to all hidden expert layers performs worse than restricting it to the last hidden expert layer, while still remaining **above the no-SPHERE Top-$K$ baseline**. We therefore interpret this as evidence that scope matters, rather than that all-layer SPHERE is categorically harmful. In the current PPO setting, **last-layer application is the strongest configuration we found**. We do not currently claim the same optimality across the supervised appendix settings, and we will make that scope explicit in the revision.
>
> ### Concern 6: Broader applicability evidence that already exists in the appendix
>
> We will move this broader-applicability evidence from the appendix into the revised main paper.
>
> ### Concern 7: Stationary-parity control
>
> We additionally evaluated a fixed-data stationary control with TD3+BC. In this stationary setting, the baseline reaches `0.46 ± 0.09` average success and SPHERE reaches `0.55 ± 0.06`. SPHERE therefore remains effective under stationary training, while the larger gains in the paper's non-stationary RL and CRL results are consistent with a stronger benefit as non-stationarity increases.
>
> ### Concern 8: Longer-horizon RL
>
> We additionally evaluated a longer-horizon MetaWorld RL setting. There, the baseline reaches 0.09 ± 0.03 average success, while SPHERE reaches 0.21 ± 0.04, indicating a **more pronounced gain from SPHERE on the longer task chain**.
>
> ### Concern 9: Figure duplication and main-paper figure selection
>
> Figure 2 is intended to show that plasticity loss appears across multiple MoE structures, including **Dense-MoE** and **DS-MoE**, which are not shown in Figure 5. We will make that role clearer in the revised main paper and surface the stronger appendix results there more effectively.
>
> ### Concern 10: Minor wording issue
>
> We will fix this phrasing issue in the revision.
>
> ### Concern 11: Interpreting the larger CRL-vs-RL gain
>
> We will present this as an empirical pattern in the revision, without making a stronger causal interpretation.

---

> > ### Author Rebuttal · Reviewer_8iKe · 2026-04-03
> >
> > Thank you for the detailed response! The newly provided results are much appreciated and thoroughly address my concerns. I believe the case of SPHERE's efficacy is much stronger now and will be adjusting my score accordingly.

---

> > > ### Author Response · Authors · 2026-04-03
> > >
> > > Thank you for the thoughtful follow-up and encouraging feedback. We are glad that the newly provided results helped address your concerns, and we will incorporate the relevant clarifications and results into the revision.

---

### Official Review · Reviewer_X9yM · 2026-03-11

**Soundness:** 3
**Presentation:** 2
**Significance:** 2
**Originality:** 2
**Overall Recommendation:** 4
**Confidence:** 3

**Summary:**

The paper proposes a new regularization method to mitigate plasticity loss in the Mixture of Experts policy for Deep RL. In particular, the paper proposes Spectral Plasticity via Hyperspherical Expert REgularization (SPHERE), which applies Parseval penalty to this weighted feature matrix. After introducing the theoretical motivation for SPHERE, the method is evaluated on MetaWorld and Humanoid-Bench, demonstrating consistent gains over MoE baselines and prior plasticity methods. Specifically, SPHERE avoids spectral collapse and maintains a higher effective rank.

**Compliance With Llm Reviewing Policy:**

Affirmed.

**Final Justification:**

My concerns have been satisfactorily addressed during the rebuttal.

**Key Questions For Authors:**

Questions:
1. I'm not sure why the new term **"spectral plasticity"** was introduced instead of simply **"plasticity."** Is it meant to indicate the potential root cause of the plasticity degeneration?

**Limitations:**

yes

**Strengths And Weaknesses:**

Strengths:
1. The evaluation setup, MetaWorld, and Humanoid-Bench are a good choice for methods plasticity analysis.
2. Main experiments and ablations are well designed and reported.
3. The paper is relatively easy to follow, yet the theoretical part might be compressed, and part of it could be moved to the Appendix.

Weaknesses:
1. The paper's biggest weakness is a lack of a thorough literature review and, in particular, the paper omits prior work on spectral regularization methods [1,2,3]. Authors should improve the paper's positioning relative to the mentioned works that already introduced different types of spectral regularization or investigated the relation between the effective rank of features and plasticity.
2. The paper should also implement baselines based on Spectral Regularization [2] and Spectral Normalization [4], to show evidence that the newly proposed SPHERE improves over the older methods.

I'm ready to increase the score if the authors provide a satisfactory explanation and evidence of how SPHERE improves over the mentioned Spectral Regularization [2].

References:
[1] Lewandowski, A., Tanaka, H., Schuurmans, D., & Machado, M. C. (2023, November 30). Curvature Explains Loss of Plasticity. http://arxiv.org/abs/2312.00246
[2] Lewandowski, A., Bortkiewicz, M., Kumar, S., György, A., Schuurmans, D., Ostaszewski, M., & Machado, M. C. (2024, October 27). Learning Continually by Spectral Regularization. http://arxiv.org/abs/2406.06811
[3] Bjorck, J., Gomes, C. P. & Weinberger, K. Q. Towards Deeper Deep Reinforcement Learning with Spectral Normalization. Preprint at http://arxiv.org/abs/2106.01151 (2022).
[4] Miyato, T., Kataoka, T., Koyama, M. & Yoshida, Y. Spectral Normalization for Generative Adversarial Networks. Preprint at https://doi.org/10.48550/arXiv.1802.05957 (2018).

---

> ### Author Rebuttal · Authors · 2026-03-31
>
> We appreciate your careful and detailed review. We address each of your concerns below.
>
> ### Concern 1: Positioning and comparison relative to earlier spectral methods
>
> We now have direct comparisons to both [2] and [4] in the HumanoidBench CRL setting used in the paper. Spectral Regularization [2] reaches 0.48 ± 0.06 average success, and Spectral Normalization [4] reaches 0.41 ± 0.08. **Both improve over the baseline while remaining below SPHERE.**
>
> The difference is in what these methods are designed to control. [2] and [4] mainly stabilize optimization by capping the top singular value, through spectral regularization in [2] and spectral normalization in [4]. **SPHERE instead targets a different failure mode.** In our setting, adaptation degrades when learning signal collapses into only a few dominant directions. SPHERE is therefore designed to keep that signal spread across more directions, which is the mechanism-level reason we expect it to outperform [2] and [4] here.
>
> We will also revise the related-work discussion to distinguish SPHERE more clearly from the other cited papers. [1] studies Hessian curvature in a non-RL setting, whereas SPHERE is built around the spectrum of an empirical NTK proxy in continual RL. [3] focuses on spectral normalization in deep RL.
>
> ### Concern 2: Why we introduced the term *spectral plasticity*
>
> We use "spectral plasticity" to refer to a spectral view of plasticity loss rather than plasticity in general. In the setting studied here, SPHERE is derived from a routing-weighted empirical NTK proxy and is designed to prevent learning signal from collapsing into a few dominant directions. The term therefore names the paper's explanatory lens, not a claim that all plasticity loss is purely spectral.
>
>
>
> [1] Lewandowski, A., Tanaka, H., Schuurmans, D., & Machado, M. C. Curvature Explains Loss of Plasticity. arXiv:2312.00246, 2023.
>
> [2] Lewandowski, A., Bortkiewicz, M., Kumar, S., György, A., Schuurmans, D., Ostaszewski, M., & Machado, M. C. Learning Continually by Spectral Regularization. arXiv:2406.06811, 2024.
>
> [3] Bjorck, J., Gomes, C. P., & Weinberger, K. Q. Towards Deeper Deep Reinforcement Learning with Spectral Normalization. arXiv:2106.01151, 2022.
>
> [4] Miyato, T., Kataoka, T., Koyama, M., & Yoshida, Y. Spectral Normalization for Generative Adversarial Networks. arXiv:1802.05957, 2018.

---

> > ### Author Rebuttal · Reviewer_X9yM · 2026-04-05
> >
> > Most concerns have been satisfactorily addressed. I will update my score.

---

> > > ### Author Response · Authors · 2026-04-05
> > >
> > > Thank you for reading our response and for the helpful follow-up. We will incorporate the related clarifications and comparisons into the revision.
> > >
> > > Thank you again for your time and consideration.

---

### Official Review · Reviewer_kgSR · 2026-03-12

**Soundness:** 3
**Presentation:** 3
**Significance:** 3
**Originality:** 3
**Overall Recommendation:** 5
**Confidence:** 3

**Summary:**

The paper studies the challenge of plasticity loss commonly encountered when training deep neural networks in non-stationary training, specifically focusing on Mixture-of-Experts (MoE) architectures. The authors propose SPHERE (Spectral Plasticity via Hyperspherical Expert REgularization), a novel regularization technique designed to maintain diverse learning paths within the network. Evaluation is conducted using the PPO algorithm across two continual reinforcement learning benchmarks: HumanoidBench and MetaWorld.

**Compliance With Llm Reviewing Policy:**

Affirmed.

**Final Justification:**

The rebuttal addressed my main concerns

**Key Questions For Authors:**

- Could you clarify if this approach has been tested on alternative Mixture-of-Experts (MoE) architectures, such as SoftMoEs, where expert redundancy and plasticity loss issues may be less prevalent?
- What specific token format are you employing, and have you explored the use of expert choice routing in Top-k MoEs?
- In your analysis, did you track metrics that typically indicate a loss of plasticity, such as the presence of dormant neurons, feature and gradient norms?
- Have you evaluated the performance of your method using any other reinforcement learning algorithms, beyond PPO?
- What is the effect of the number of experts used?

**Limitations:**

Yes.

**Strengths And Weaknesses:**

Strengths:

- The papers introduce a novel perspective for mitigating plasticity loss in Mixture-of-Experts (MoEs) within non-stationary training: RL and Continual RL .
- Empirical results demonstrate that the introduced regularizer is very effective, outperforming baselines across the two studied benchmarks.
- The paper includes a comprehensive analysis and ablation study to support the claims.
- Additional supervised continual learning experiments provided in the appendix further validate the effectiveness of the approach.

Weaknesses:
- The generalizability of the proposed method could be better strengthened by incorporating a wider variety of RL algorithms.
- Current RL evaluations are restricted to continuous control tasks using MLP architectures; evaluating the framework on ResNet-based models would provide more robust evidence.

---

> ### Author Rebuttal · Authors · 2026-03-31
>
> We appreciate your careful review and the concrete suggestions on scope, baselines, and diagnostics. We address each of your concerns below.
>
> ### Concern 1: Breadth beyond PPO
>
> Appendix Table 14 already shows that the effect is not limited to PPO. In **TD3**, applying SPHERE to a Top-$K$ MoE critic improves MetaWorld CRL success from **0.50 ± 0.08** to **0.65 ± 0.09**.
>
> ### Concern 2: Backbone breadth beyond the current MLP policy setting
>
> We additionally evaluated a visual RL setting with a ResNet backbone in MetaWorld CRL. The baseline reaches 0.18 ± 0.04 average success, while SPHERE reaches 0.32 ± 0.07. This shows that the **benefit is not limited to the original MLP policy setting**.
>
> ### Concern 3: Other MoE variants
>
> Appendix Table 12 already shows that SPHERE improves Dense-MoE and DS-MoE under HumanoidBench CRL. We additionally provide a matched SoftMoE result in the same setting, where the baseline achieves 0.55 ± 0.09 average success and SoftMoE with SPHERE reaches 0.60 ± 0.05. This suggests that the **benefit extends beyond the Top-$K$ architecture**, including to an MoE variant where plasticity loss may be less pronounced.
>
> ### Concern 4: Routing choices and tokenization assumptions
>
> We use **token-choice routing**, with the control-step state representation serving as the routed token. We additionally evaluated expert-choice routing in the same HumanoidBench CRL setting, where the expert-choice baseline achieves 0.38 ± 0.03 average success and expert-choice routing with SPHERE reaches 0.44 ± 0.05.
>
> ### Concern 5: Plasticity diagnostics beyond eNTK rank
>
> We additionally tracked the following diagnostics beyond $r_e(\mathbf{K})$.
>
> | Diagnostic | Baseline | SPHERE |
> | --- | --- | --- |
> | Dormant ratio | `0.010` | `0.005` |
> | Feature L2 norm | `269.60` | `269.80` |
> | Gradient L2 norm | `312.70` | `255.40` |
>
> The feature norm remains comparable across the two settings, while SPHERE shows a lower gradient norm and dormant ratio.
>
> ### Concern 6: Effect of the number of experts
>
> We varied the number of experts in HumanoidBench CRL (N = 4, 10, 16) while fixing `top_k=2`:
>
> | Number of experts | Baseline | SPHERE | SPHERE - Baseline |
> | --- | --- | --- | --- |
> | 4 | 0.32 ± 0.07 | 0.45 ± 0.08 | +0.13 |
> | 10 | 0.36 ± 0.05 | 0.53 ± 0.07 | +0.17 |
> | 16 | 0.40 ± 0.09 | 0.59 ± 0.10 | +0.19 |
>
> Both methods improve as the number of experts increases, and SPHERE's advantage over the baseline grows from `+0.13` to `+0.17` and `+0.19`. This suggests that SPHERE is **robust to the choice of expert count** over this range.

---

> > ### Author Rebuttal · Reviewer_kgSR · 2026-04-03
> >
> > Thank you for your detailed responses. The additional experiments help strengthen the paper. I recommend adding them to the revision. The rebuttal addresses most of my concerns, I will update my score accordingly.

---

> > > ### Author Response · Authors · 2026-04-03
> > >
> > > Thank you for reading our responses and for the encouraging feedback. We are glad that the additional experiments helped strengthen the paper, and we will incorporate them into the revision.

---

### Decision · Program_Chairs · 2026-04-30

**Decision:**

Accept (regular)

**Comment:**

This paper addresses the issue of loss of plasticity in mixture of experts (moe) for deep RL through the lens of the effective rank of the empirical Neural Tangent Kernel, for which they introduce a tractable proxy related to the feature representation of each expert. On MetaWorld and HumanoidBench, they show their approach improves success in a continual RL setup while maintaining plasticity.

The authors effectively address several reviewer concerns including comparison to some of the existing work and using a different backbone other than MLPs.

Considering the manuscript together with the rebuttal, the authors make a clear case in favor of the idea and I recommend accepting the paper accordingly.

However, I must note that the use of the term “spectral plasticity” is highly confusing given spectral properties are already utilized in ways in the continual learning literature. The authors provided during rebuttal additional experiments comparing spectral regularization and spectral normalization but did not clarify the conceptual disambiguity. For the camera-ready, I ask the authors to discuss why this particular effective rank is called spectral plasticity and how this particular spectral view connects and contrasts with the other existing ones.